# Anoxic chlorophyll maximum enhances local organic matter remineralization and nitrogen loss in Lake Tanganyika

Cameron M. Callbeck [1,3✉], Benedikt Ehrenfels [1,2,3], Kathrin B. L. Baumann[1,2], Bernhard Wehrli[1,2] & Carsten J. Schubert [1,2]

In marine and freshwater oxygen-deficient zones, the remineralization of sinking organic matter from the photic zone is central to driving nitrogen loss. Deep blooms of photosynthetic bacteria, which form the suboxic/anoxic chlorophyll maximum (ACM), widespread in aquatic ecosystems, may also contribute to the local input of organic matter. Yet, the influence of the ACM on nitrogen and carbon cycling remains poorly understood. Using a suite of stable isotope tracer experiments, we examined the transformation of nitrogen and carbon under an ACM (comprising of Chlorobiaceae and Synechococcales) and a non-ACM scenario in the anoxic zone of Lake Tanganyika. We find that the ACM hosts a tight coupling of photo/litho-autotrophic and heterotrophic processes. In particular, the ACM was a hotspot of organic matter remineralization that controlled an important supply of ammonium driving a nitrification-anammox coupling, and thereby played a key role in regulating nitrogen loss in the oxygen-deficient zone.

[1] Eawag, Swiss Federal Institute of Aquatic Science and Technology, Surface Waters—Research and Management, Kastanienbaum, Switzerland. [2] Institute of Biogeochemistry and Pollutant Dynamics, ETH Zurich, Zurich, Switzerland. [3] These authors contributed equally: Cameron M. Callbeck, Benedikt Ehrenfels.
✉email: cameron.callbeck@eawag.ch

In the global N-budget, oxygen-deficient zones that are near anoxic ($O_2 < 1–2$ μM) in freshwater and marine environments, play a disproportionate role in converting bioavailable nitrogen ($NO_x$ and $NH_4^+$) to inert $N_2$ gas, thus removing it from the active biosphere[1]. Marine eutrophic upwelling regions, where large oxygen-deficient zones form, contribute to nearly one-fifth of global $N_2$ production at 81 Tg N per year, and freshwater lakes contribute up to 31 Tg N per year[1]. These freshwater and marine waterscapes together exert significant control over the nitrogen budget, despite covering only a small fraction of the Earth's surface.

Fluxes of organic matter are essential to driving nitrogen loss in these aquatic environments. Organic matter fluxes sustain low-oxygen concentrations, and hence, the use of alternative electron acceptors such as nitrate. In addition, the remineralization of sinking organic matter releases ammonium that, in turn, drives a tightly coupled nitrogen cycle producing $N_2$ via nitrification, nitrate reduction to nitrite, dissimilatory nitrate reduction to ammonium (DNRA), denitrification, and anammox. In both marine oxygen deficient zones[2–6], and in highly stratified basins and freshwater lakes[7–11], active nitrogen turnover to $N_2$ is generally greatest near the top of the anoxic zone. While primary productivity in the euphotic region sustains a flux of sinking particles to the anoxic zone[3], high rates of organic matter remineralization in the water column strongly attenuate these fluxes with depth. Particle trap data and dissolved organic matter profiles in water column studies additionally highlight the presence of deep organic matter peaks that form in the oxygen-deficient zone[12–17]. These local peaks are often found in association with the suboxic/anoxic chlorophyll maximum (or ACM) located at the base of the photic zone, also known as the secondary chlorophyll maximum[12–17]. The phototrophic organisms that thrive in the ACM are believed to offer an important local source of the labile organic matter driving nitrogen cycling in the oxygen-deficient zone[12,18,19].

In marine oxygen-deficient zones, the ACM tends to be dominated by the low-light adapted cyanobacteria genera *Prochlorococcus* and to a lesser extent by *Synechococcus*[12,18,20–22]. The marine ACM, which can occur at an irradiance of 5 μmol photons $m^{-2} s^{-1}$[18], is considered a prevalent feature of most marine oxygen- deficient zones[12,18,20–22] and can persist year-round within a stable isopycnal layer[15,23]. Oxygenic photosynthesis by cyanobacteria has been shown to generate molecular oxygen ($O_2$) possibly supporting microaerobic nitrification in the ACM[18]. In euxinic basins and in stratified freshwater lakes, the ACM can comprise of Cyanobacteria (such as *Plankthotrix*, *Phormidium*, and *Synechoccocus*)[13,19,24–26], that has also been shown to generate $O_2$ in situ[13]. In addition, the reduced sulfur fluxes in these systems can sustain anoxygenic photosynthesis by green sulfur bacteria. Unlike cyanobacteria, green sulfur bacteria such as *Chlorobium*, produce sulfur instead of oxygen, and either assimilate dissolved inorganic nitrogen and/or fix $N_2$ into biomass partly satisfying local nitrogen loss[19]. In these two-layer phototrophic systems with cyanobacteria, *Chlorobium* occupies the bottom layer of the ACM[13,19,24–26], due to their extreme light-harvesting capabilities of only 0.001 μmol photons $m^{-2} s^{-1}$[27]. The co-occurrence of anoxygenic and oxygenic photosynthesis in the ACM is not only common to stratified lakes and basins but has also been hypothesized to be a widespread feature of the euxinic Proterozoic ocean[28]. Both anoxygenic and/or oxygenic photosynthesis has a capacity to fix $CO_2$ into biomass[29], contributing to an important production of organic matter in the ACM.

We hypothesize that the production of labile organic matter by phototrophs in the ACM, coupled with its remineralization to ammonium by local heterotrophic bacteria, might have a significant influence on nitrogen cycling in oxygen-deficient zones.

We tested this hypothesis in Lake Tanganyika—the largest freshwater oxygen-deficient zone in the world. Located along the East African Rift, Lake Tanganyika (maximum depth of 1,470 m) supports an active nitrogen cycling community at the top of an anoxic zone with euxinic bottom waters at deeper depths[10,30–32], similar to the Black Sea and to some coastal marine oxygen-deficient zones[33]. Moreover, the presence of an ACM in the northern basin and its absence in the southern basin (Fig. 1a), provides an ideal framework to evaluate the impact of deep phototrophic communities on nitrogen cycling in the oxygen-deficient zone.

To quantify the effect of the ACM on carbon and nitrogen cycling in Lake Tanganyika we employed a suite of stable isotope tracer experiments. We performed incubation experiments with isotopically labeled nitrogen species: $^{15}NH_4^+$, $^{15}NO_3^-$, and $^{15}NO_2^-$; in concert with $^{13}C/^{15}N$-algae to quantify potential rates of inorganic and organic nitrogen transformation processes in the ACM. In addition, we used the $^{13}C/^{15}N$-algal amendment experiments to quantify rates of $CO_2$ production, in parallel with carbon fixation experiments and natural abundance isotope ratios of $\delta^{13}C_{DIC}$ to evaluate the transformation of inorganic and organic carbon in the ACM. Together, our biogeochemical tracer experiments provide a comprehensive survey of inorganic carbon and nitrogen transformation processes and its potential coupling with organic matter turnover in the oxygen-deficient zone.

## Results and discussion

**Characterization of the ACM in Lake Tanganyika**. Our main stations 2 and 7, sampled in early May at the onset of the dry season, were located in the northern and southern basins of Lake Tanganyika, respectively. The epilimnion was characterized by a warm (27–28 °C) and relatively well-mixed water mass from 0 to 30 m, with the thermocline occurring from 30 to 75 m (Figs. 1b and 2c, k). However, station 7, and the southern basin in general, exhibited a steeper thermal gradient throughout the metalimnion than compared to station 2 (Figs. 1c and 2c, k). The steep thermocline at station 7, also coincided with a peak in $N^2$, otherwise known as the Brunt–Väisälä frequency (Fig. 2c, k), which also reflects changes in the temperature gradient. The onset of the south-east trade winds in May[34] is known to induce nutrient upwelling and the upward tilting of the thermocline, resulting in a shallow and steep temperature gradient to develop in the southern basin[35].

The epilimnion of stations 2 and 7 exhibited a maximum in the rates of primary productivity linked to in situ chlorophyll fluorescence peaking at 45 and 36 m, respectively, with a minimum occurring at 100 m depth (Fig. 2a, i, d, l). A higher in situ chlorophyll fluorescence was reported at station 7, which is consistent with previous reports of higher overall primary productivity in the southern basin compared to the northern basin as a result of nutrient upwelling[36]. Below the primary chlorophyll maximum, nitrate concentrations peaked at 9–10 μM as a result of organic matter remineralization and nitrification (Fig. 2b, j). In addition, both stations exhibited similar nitrate inventories (~800 mmol $NO_3^-$ $m^{-2}$) and the nitrate fluxes from the metalimnion into the underlying oxygen-deficient zone were of comparable magnitude (0.24 and 0.29 mmol $NO_3^-$ $m^{-2} d^{-1}$, for stations 2 and 7, respectively).

However, below the primary chlorophyll peak, we observed contrasting profiles between stations 2 and 7. At station 2, the vertical turbidity signal from 50 to 120 m depth exhibited a high degree of vertical structure (Fig. 2c); particularly in contrast to station 7, where we find a much smoother turbidity profile over the same depth range (Fig. 2k). Interestingly, the turbidity peak at 105 m, at station 2, coincided with a small increase in the $N^2$

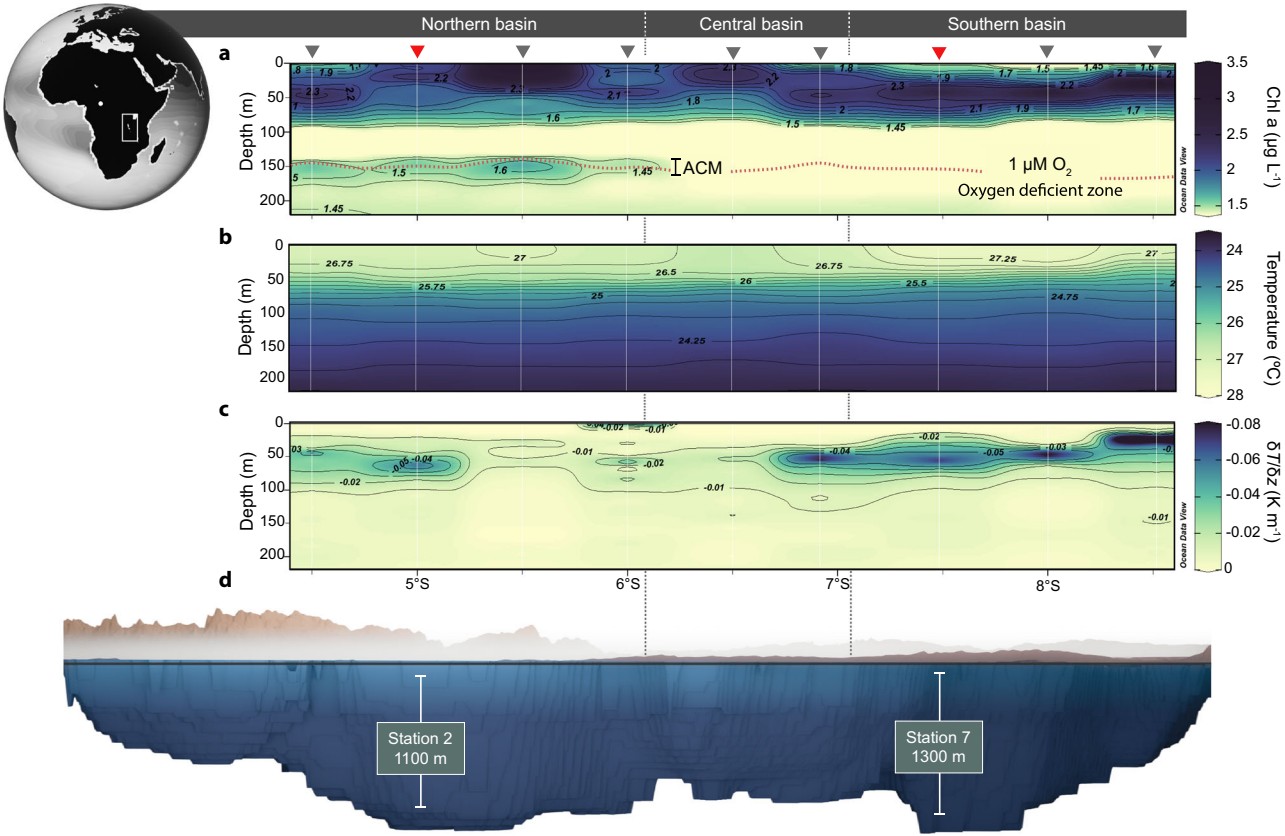

**Fig. 1 Distribution of the anoxic chlorophyll maximum (ACM) in the oxygen-deficient zone of Lake Tanganyika.** Data shown are from the April–May 2018 campaign, while a return expedition was made in April–May 2019 to better resolve the vertical structure and nitrogen transformation processes in the ACM, shown in Figs. 2 and 3. **a** In situ chlorophyll a fluorescence. The oxygen-deficient zone is delimited at an $O_2$ cutoff of 1 µM indicated by the dotted line. The main sampling stations 2 and 7 are marked by red triangles, while additional sampling stations are indicated by black triangles. **b** Temperature distribution. **c** Vertical temperature gradient. **d** Bathymetry of Lake Tanganyika and the sill that separates the North, Central, and South basins, based on the bathymetry map by Cohen et al.[57]. The lake water column is denoted in blue, while the surrounding mountain terrain is indicated in brown.

stability peak from 95 to 120 m depth (Fig. 2c; Peak A), indicating some degree of particle aggregation occurring at a deep layer above the oxygen-deficient zone. At station 7, the vertical $N^2$ profile revealed no large changes in density, and only relatively minor noise occurring from 65 to 140 m depth (Fig. 2k).

Furthermore, at station 2, the oxycline was positioned at a shallower depth and the nitracline was present well into the anoxic zone than compared to station 7 (Fig. 2a, b, i, j). At the bottom of the nitracline, station 2 supported an ACM spanning from 130 to 170 m depth, with a maximum in situ fluorescence occurring at 140 m (Fig. 2a). Congruently, fluorescent cells, as detected by epifluorescence microscopy comprised nearly 40% of the total cell counts or up to $4.6 \times 10^5$ cells ml$^{-1}$ in the ACM. The photosynthetically active radiation (PAR) at the ACM was below the limit of detection at less than <0.05 µmol photons m$^{-2}$ s$^{-1}$ as the light irradiance was strongly attenuated at ~100 m depth (Supplementary Fig. S1a). However, because the PAR measurements were performed in the morning, the transition from morning to mid-day light, which results in a nearly 2-fold increase in surface light irradiance (Supplementary Fig. S1a), would have likely resulted in deeper light penetration at station 2. Albeit, the exact amount of PAR that would reach the ACM at mid-day remains unclear.

In the ACM, recovered 16S rRNA genes identified both Synechococcales, reaching up to 1.4% of the microbial community at 150 m depth; and Chlorobiaceae, which attained up to 21.6% at a slightly deeper depth of 156 m (Supplementary Fig. S2b, c). Members of Chlorobiaceae are ostensibly supported

by the reduced sulfur flux from the hypolimnion[31], coupled to its ability to thrive under extreme low-light conditions[27]. Although Chlorobiaceae appear to dominate in the ACM, it is possible that we missed the maximum Synechococcales peak between 133 and 150 m depth. Indeed, previous studies have reported microstratification in two-layer phototrophic systems[26]. Nevertheless, how members of Synechococcales could persist under such extreme light conditions in the ACM is enigmatic. Niche partitioning between oxygenic and anoxygenic light-harvesting zones has been reported in other highly stratified lakes, even at vanishingly low light availabilities[13,26]. Alternatively, a study in the Black Sea, which cultivated deep anoxic populations of *Synechococcus* (recovered from 750 m depth) found that they were capable of a heterotrophic mode of growth in the absence of light[37]. Thus, the ACM likely sustains active anoxygenic photosynthesis by green sulfur bacteria. However, it is less clear if the low-light availability could support oxygenic photosynthesis by ACM Synechococcales, alternatively, they may grow via chemoorganoheterotrophy.

Apart from phototrophs, general bacterial cell densities, although strongly attenuated with depth, reached a small maximum of $1.1 \times 10^6$ cells ml$^{-1}$ at 135 m (Supplementary Fig. S3b). The combined maximum in total cell densities (both phototrophic and non-phototrophic) coincided with the turbidity signal at 135 m (Fig. 2c), in line with other ACM studies[26]. The ACM at station 2 was situated between the nitrate and ammonium gradients, and a solitary nitrite peak emerged (up to 0.66 µM) at the ACM maximum (Fig. 2b). Interestingly,

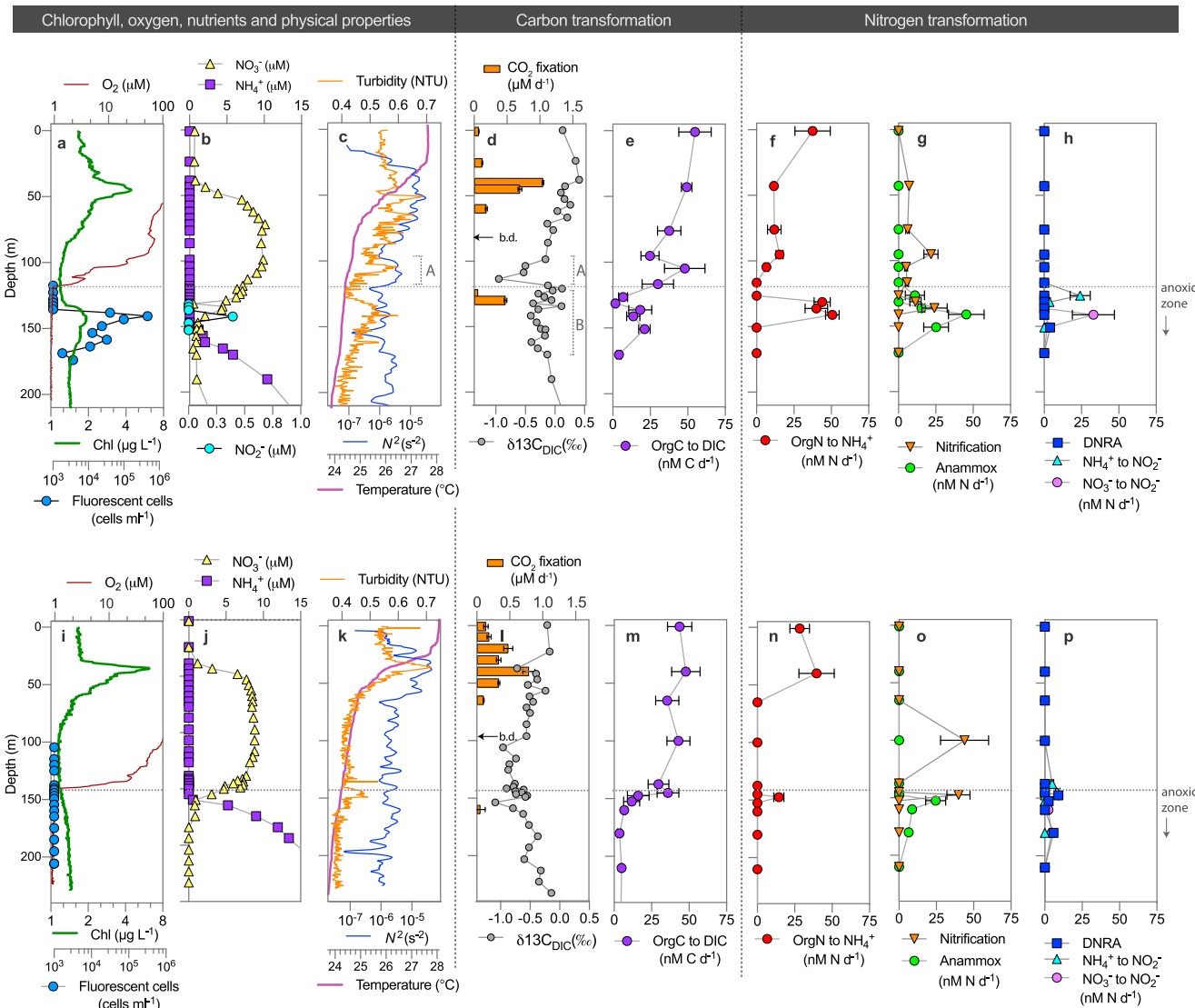

**Fig. 2 Nutrient chemistry and the vertical distribution of carbon and nitrogen transformation processes in Lake Tanganyika.** The data shown is from the April–May 2019 expedition. Stations 2 and 7 are indicated in the top (**a–h**) and bottom (**i-p**) panels, respectively. The oxygen-deficient zone is delimited at an $O_2$ cutoff of 1 μM indicated by the dotted line. Note that the x-axes related to oxygen and chlorophyll profiles are not shown on linear scales. The bracketed lines in panels **c** and **d** indicate peaks A and B discussed in the text. Error bars on carbon fixation rates, panels **d** and **l**, represent the standard deviation; while error bars on panels **e–h** and **m–p** represent the standard error (see "Methods"). Abbreviations are as follows: b.d. below detection limits, DNRA dissimilatory nitrate reduction to ammonium, DIC dissolved inorganic carbon (i.e., $CO_2$).

concentrations of total organic carbon (TOC) and total organic nitrogen (TON) exhibited a relatively noisy vertical profile across the ACM from 130 to 150 m depth. Here, TOC concentrations fluctuated from 85 to 109 μM, and TON concentrations ranged from 6 to 11 μM (Supplementary Fig. S3d, e).

By contrast, at station 7, no obvious ACM was detected based on the in situ chlorophyll fluorescence and by fluorescent cell counts, with a limit of detection ~$10^3$ cells ml$^{-1}$ (Fig. 2i). Moreover, most of the light irradance was attenuated at 80 m depth (measured at mid-day), well above the top of the anoxic zone (140 m; Supplementary Fig. S1b). The relatively high primary production sustained by the wind-driven upwelling in the southern basin likely contributed to the increased light attenuation. Even in the absence of an ACM, the top of the anoxic zone at station 7 was denoted by a small turbidity signal at 145 m, and was accompanied by increases in TOC (101–115 μM) and TON concentrations (6–10 μM; Supplementary Fig. S3l, m).

Furthermore, bacterial cell densities observed a small maximum near the turbidity peak (145 m) attaining up to $5.7 \times 10^5$ cells ml$^{-1}$ (Fig. 2k, S3j), although the median cell densities were nearly twofold lower than compared to station 2. Overall, at station 7, the maxima of TOC/TON, bacterial abundances, and turbidity, observed a small peak at the top of the anoxic zone, where the ammonium and nitrate gradients also intersected (Fig. 2j).

Beyond ~175 m depth, both stations 2 and 7 exhibited a gradual increase in the in situ chlorophyll fluorescence (Fig. 2a, i). In the euxinic Black Sea, Callieri et al.[37] have ascribed the deep fluorescence in the mesopelagic to the presence of an anoxic population of *Synechococcus* (reaching up to $10^3$ cells ml$^{-1}$ at 750 m), which may grow heterotrophically in the absence of light. Previous molecular surveys in Lake Tanganyika have also recovered 16S rRNA gene sequences affiliated to both *Synechococcus*, and green non-sulfur bacteria with similarities to *Chloroflexi*, in the deep anoxic zone from 200 to 1,000 m[30].

**The complex interaction of heterotrophic and autotrophic activity in the ACM**. To examine carbon cycling processes at stations 2 and 7, we analyzed the $\delta^{13}C_{DIC}$ to distinguish autotrophic and heterotrophic activity. In parallel, we performed incubation experiments with amended $^{13}C$-$HCO_3^-$ and $^{13}C/^{15}N$-lyophilized algal biomass to quantify rates of carbon fixation and potential rates of organic matter remineralization to DIC (i.e., organic carbon remineralization), respectively.

At stations 2 and 7, the primary chlorophyll peak was characterized by high carbon fixation with measured rates reaching up to 1,000 and 700 nM C $d^{-1}$, respectively (Fig. 2d, l). This was also consistent with the heavy $\delta^{13}C_{DIC}$ signature: since carbon fixation preferentially incorporates $^{12}C$-DIC into biomass causing an enriched $^{13}C$-DIC pool (Fig. 2d, l). Below the primary chlorophyll peak, a shift to dominance by heterotrophy coupled to oxygen respiration was evidenced by a strong decline in $\delta^{13}C_{DIC}$ ($\Delta0.9$–1.4‰), whereby the remineralization of organic matter to $^{12}C$-DIC dilutes the $^{13}C$-DIC pool. At station 2, the 105 m turbidity maximum was associated with a particularly significant $\delta^{13}C_{DIC}$ decrease of $\Delta0.8$‰ and a peak in the potential rates of organic carbon remineralization up to 51 nM C $d^{-1}$ (Fig. 2c, d; Peak A). The presence of this deep density anomaly in the oxic water column, which supported high turbidity and enhanced organic matter remineralization, possibly limited surface exported organic matter to the oxygen-deficient zone. In contrast, at station 7, a similarly-sized local decrease in the $\delta^{13}C_{DIC}$ profiles was not evident from 50 to 140 m depth (Fig. 2l). Moreover, the oxygenated water column exhibited relatively consistent rates of organic carbon remineralization of ~40 nM C $d^{-1}$ (Fig. 2m).

Entering the anoxic zone, at station 2 we observed a strong shift to heavy $\delta^{13}C_{DIC}$, which was accompanied by an increase in the rates of carbon fixation up to 416 nM C $d^{-1}$ (Fig. 2d, e), and an increase in the $C:N_{TOC:TON}$ ratio (Supplementary Fig. S3g). Across the ACM, $\delta^{13}C_{DIC}$ reported a noisy vertical structure (Fig. 2; Peak B), which was similar to the vertical structure of the TOC profile (Supplementary Fig. S3d, e). Interestingly, rates of potential organic carbon remineralization, while generally attenuated with depth reported a prominent secondary increase in the ACM up to 21 nM C $d^{-1}$ (Fig. 2e). The secondary increase in the rates of organic carbon remineralization, in combination with the elevated carbon fixation rates and the noisy $\delta^{13}C_{DIC}$, and TOC profiles, suggests that the ACM hosts a complex mix of autotrophic and heterotrophic activity.

In contrast, at station 7, the shift to heavy $\delta^{13}C$ ($\Delta0.7$‰) at the top of the anoxic zone was less pronounced than observed at station 2 ($\Delta1.1$‰; Fig. 2l), and was accompanied by lower rates of carbon fixation (46 nM C $d^{-1}$). Nevertheless, the heavy $\delta^{13}C$ signal occurred in line with the small turbidity peak at 145 m (Fig. 2k) and coincided with a local increase in organic carbon remineralization (Fig. 2m). Thus, in absence of an ACM at station 7, autotrophic and heterotrophic processes appear to be more narrowly confined to the area just below the oxycline than compared to station 2.

Given the inherent complexity of the carbon cycle, we suspect that there is some degree of uncertainty in our organic carbon remineralization rates as the $^{13}C/^{12}C$ isotope ratio of DIC may be susceptible to dilution by concurrent processes. For instance, $^{12}C$-$HCO_3^-$ production from ambient organic matter remineralization as well as the assimilation of remineralized $^{13}C$-$HCO_3^-$ into biomass, could contribute to diluting the $^{13}C/^{12}C$ ratio. Apart from DIC production, a fraction of the remineralized algal biomass might accumulate in the dissolved organic carbon pool (i.e. acetate), which is missed in this survey, but that could be considered in future analyses. Moreover, different organic matter types and quality could be assessed as this may alter DIC production. These caveats possibly

contribute to an underestimation of the DIC production rates, which were much lower relative to the rates of carbon fixation. Nevertheless, the potential rates of organic carbon remineralization support that an active heterotrophic community occurs in the ACM at station 2, and at the top of the anoxic zone at station 7.

In the oxygen-deficient zone in Lake Tanganyika, a community of heterotrophic bacteria is likely responsible for the sequential breakdown of algal biomass to $CO_2$. Certain phyla play an important role in the remineralization of organic matter, and the seminal water column study by Bergauer et al.[38], indicates that phyla such as Alphaproteobacteria, Gammaproteobacteria, Deltaproteobacteria, Bacteroidetes, and Actinobacteria are key contributors to water column organic matter uptake and remineralization. These same major phyla were also identified in our 16S rRNA gene analysis in the anoxic waters at stations 2 and 7 (Supplementary Fig. S2d). Specifically, we identified a high abundance of Gammaproteobacteria, mostly within the order Betaproteobacteriales, which averaged 36% of the 16S rRNA microbial community at stations 2 and 7 (Supplementary Fig. S2d, j). In addition, we identified a moderate fraction of Bacteroidetes that consisted mainly of the orders Sphingobacteriales (8%), Chlorobiales (5%), and Ignavibacteriales (2%), as well as the phylum Actinobacteria (8.6%). Alphaproteobacteria, Deltaproteobacteria, and Firmicutes were also identified, but in lower relative abundances (1–3%). Bergauer et al.[38] have shown that Gammaproteobacteria and Bacteroidetes contain an abundance of high molecular weight transport systems that are suggested to be involved in the uptake and degradation of labile/semi-labile organic material. Bacteroidetes and Firmicutes were further shown, using similar $^{13}C$-labeled algal additions in a marine sediment study, to initially catalyze the degradation of the algal tracer to smaller molecular weight compounds, such as acetate[39]. We suspect that Bacteroidetes and Gammaproteobacteria could serve an analogous role in degrading organic matter in Lake Tanganyika, and in our algal addition experiments.

Other groups that might specialize in the uptake and remineralization of more labile organic material include Actinobacteria, which have been shown to contain an abundance of amino acid and carbohydrate-based transport systems[40]. In addition to Actinobacteria, the ACM at station 2 contained sequences affiliated to Ignavibacteria, which reached up to 8% of the microbial community (Supplementary Fig. S2d). Ignavibacteria are non-pigmented, heterotrophic bacteria that are distantly related to Chlorobi; and have been previously shown in anaerobic anammox reactors to facilitate the remineralization of proteins, possibly providing substrates for anammox in a close metabolic coupling[41]. Thus, our cursory inspection of the 16S rRNA diversity identifies putative heterotrophic phyla that could function to remineralize labile/semi-labile organic matter in the anoxic zone. Such a community is likely sustained by surface exported organic matter to the anoxic zone. In addition, the co-occurrence of photoautotrophic taxa and lithoautotrophs associated with the nitrogen cycle (Supplementary Fig. S2k, e), and the relatively high rates of carbon fixation in the ACM (Fig. 2d), suggest that this community of heterotrophs is also supported by the input of labile organic carbon.

**Organic nitrogen remineralization in the ACM**. From the double-labeled $^{13}C/^{15}N$-algal additions we also quantified the potential for organic nitrogen remineralization to ammonium at stations 2 and 7. Ammonium, which is liberated from organic matter degradation by heterotrophic bacteria, is a potential substrate for nitrogen cycling activities[42]. Given that nitrification can be an important sink of ammonium, we performed experiments

with $^{13}C/^{15}N$-algae and allylthiourea, an inhibitor of ammonium oxidation (see "Methods"). Based on these experiments, we find moderate rates of organic matter remineralization to ammonium (i.e. organic nitrogen remineralization) associated with surface waters reaching up to 37.5 and 39.6 nM N $d^{-1}$. This is likely attributable to heterotrophic activity (Fig. 2f, n). Rates then subsided as a function of depth in a similar pattern as the rates of organic carbon remineralization.

Moreover, at station 2, rates of organic nitrogen remineralization, recovered to a mean of 44.9 nM N $d^{-1}$ at 135–145 m, coinciding with the ACM. Notably, $^{13}C/^{15}N$-algal additions without amended allylthiourea resulted in no detectable $^{15}NH_4^+$ production from 135 to 145 m depth (Supplementary Fig. S3h), affirming that nitrification was an important sink of remineralized ammonium in the ACM. Although algal additions were performed in the dark, precluding in situ $O_2$ production from oxygenic photosynthesis, low micromolar levels of $O_2$ introduced from sample handling of anoxic waters could sustain microaerobic nitrification (see "Methods"). In the same algal additions, we also detected labeled $N_2$ gas production, suggesting that anammox and/or denitrification were a competing sink for remineralized ammonium discussed below (Supplementary Fig. S3).

Furthermore, the volumetric rates of organic nitrogen remineralization (45 nM N $d^{-1}$) exceeded the rates of organic carbon remineralization (24 nM C $d^{-1}$) in the ACM at station 2 (Fig. 2e, f). It is possible that the DIC and $NH_4^+$ regeneration ratio of ~0.5:1 is exaggerated as a result of the addition of more N-rich algal material (C:N ratio of 3.2) compared to the ambient organic matter pool (Supplementary Fig. S3g, o). However, the same algal additions applied in the oxygenated water column at both stations 2 and 7, observed the opposite pattern, whereby the rates of organic carbon remineralization exceeded the rates of organic nitrogen remineralization. Indeed, previous water column studies have reported selective remineralization of nitrogen-rich organic matter, such as amino acids, coupled to denitrification in in situ experiments[43]. In globally compiled marine datasets, selective remineralization of nitrogen over carbon was shown to increase with depth (from the thermocline down to intermediate waters), with models also reproducing this pattern of preferential heterotrophic remineralization[44,45]. It is tempting to speculate that the ACM contributes to selective organic nitrogen remineralization, but this requires further investigation in the future using various labeled organic matter substrate additions.

At station 7, rates of organic nitrogen remineralization (with amended allylthiourea) were mostly below the limit of detection in the oxygen-deficient zone, with the exception of a local maximum occurring at the top of the anoxic zone up to 14.4 nM $d^{-1}$ at 148 m (Fig. 2n). The same peak in organic nitrogen remineralization also coincided with the turbidity maximum and the peak in organic carbon remineralization, suggesting that the top of the anoxic zone is active in the turnover of organic matter.

Overall, our measured rates of organic nitrogen remineralization, which reached a maximum of 50 nM $d^{-1}$, were within the range of ammonium regeneration estimates for the water column of other oxygen-deficient zones, which range between 21 and 333 nmol N $L^{-1}$ $d^{-1}$[17]. We suspect that our measured rates of organic nitrogen remineralization are subject to a degree of uncertainty, similar to the rates of organic carbon remineralization. For instance, the $^{15}N/^{14}N$ isotope ratio may be susceptible to dilution by concurrent processes, such as by $^{14}NH_4^+$ production from ambient organic matter and by the rapid assimilation/dissimilation of remineralized $^{15}NH_4^+$ by anammox and nitrification (discussed below). Moreover, a fraction of the remineralized algal biomass possibly accumulates in the dissolved organic nitrogen pool (i.e., urea)[42], and may serve as an alternative substrate for anammox and nitrification[46,47]. As mentioned earlier, future

surveys might also consider performing similar incubations with different organic matter types and quality. Despite the degree of uncertainty in the absolute values, the algal incubations provide insight into the potential for heterotrophic activity across the water column.

**Contribution of organic nitrogen remineralization to nitrogen cycling.** In separate incubations using inorganic $^{15}N$-labeled nitrogen species, we affirmed the presence of active nitrogen cycling in the oxygen-deficient waters of Lake Tanganyika. Ammonium oxidation to nitrate, which we refer to as nitrification hereafter, was most active in the euphotic zone and at the nitrate maximum, reaching up to 21.7 and 43.9 nM N $d^{-1}$ at stations 2 and 7, respectively (Fig. 2g, o). At the same depths, we also detected the production of labeled nitrate in our algal incubation experiments (Supplementary Fig. S3), indicating a tight coupling of organic matter remineralization and nitrification activity. At station 2, nitrification rates were low or near-zero at the top of the anoxic zone (110–123 m), but recovered to a maximum of 24.1 nM N $d^{-1}$ at 140 m, coinciding with the ACM. At station 7, we find a secondary nitrification peak of 39.8 nM N $d^{-1}$ at the top of the anoxic zone (Fig. 2o).

Our 16S rRNA gene survey identified ammonium-oxidizing Bacteria and Archaea within Nitrosomonadaceae and Nitrosopumilaceae (Thaumarchaeota), respectively (Supplementary Fig. S2e, k). Members of the Thaumarchaeota largely dominated in the anoxic zone, in line with a previous lipid biomarker study in Lake Tanganyika[32]. Ammonium-oxidizing bacteria also co-occurred with nitrite-oxidizing bacteria, Nitrospirae. In other marine water column studies, sequences affiliated to ammonium- and nitrite-oxidizing prokaryotes, have been shown to peak in the anoxic waters of the ACM[18], where they may thrive at extremely low oxygen and ammonium concentrations[48,49].

The $^{15}N$-algal experiments also observed the production of labeled $N_2$ gas in samples collected in the anoxic zone (Supplementary Fig. S3), suggesting that organic matter remineralization was tightly coupled with nitrogen loss processes denitrification/anammox. Previous rate process measurements by Schubert et al.[10] estimated that anammox contributed up to 13% of the $N_2$ production rate, with the remainder ascribed to denitrification based on the production of $^{30}N_2$ gas from $^{15}NO_3^-$ additions. At stations 2 and 7, we detected only the production of $^{29}N_2$ gas (i.e., no $^{30}N_2$ production was detected) in the $^{15}NO_3^-$ additions, suggesting that complete denitrification to $N_2$ was not as active at the time of sampling. Previous marine studies have found that denitrification can occur sporadically due to the sudden inputs of organic matter[5,50]. The lack of $^{30}N_2$ gas production from the $^{15}NO_3^-$ additions, combined with the presence of $^{29}N_2$ production measured in most other incubations ($^{15}NO_3^-$, $^{15}NH_4^+$, and $^{15}NO_2^-$) was consistent with anammox activity. Interestingly, however, $^{30}N_2$ production was detected in $^{15}NH_4^+$ and $^{15}NO_2^-$ incubations, but only at the upper and lower edges of the anoxic zone where nitrification and DNRA activity occurred, respectively (Fig. 2g, h, o, p). The $^{30}N_2$ production in the former and latter could be ascribed to a nitrification-anammox[7] and DNRA–anammox coupling[51] (anammox bacteria have also been shown to facilitate DNRA activity themselves[52]). Such couplings have been previously demonstrated in oxygen-deficient waters to generate $^{30}N_2$ gas in similar experiments[7,51,52].

Furthermore, our 16S rRNA microbial community survey at stations 2 and 7, observed an overlap of Planctomycetes (comprising of the OM190/CL500-3 and the anammox phylum *Candidatus* Brocadia) with nitrifying prokaryote sequences mentioned above (Supplementary Fig. S2e, k). In a previous

Lake Tanganyika survey, anammox lipids have been shown to overlap with Thaumarchaeota in the anoxic zone[10,32]. We, therefore, suggest that anammox was an important process contributing to nitrogen loss at stations 2 and 7. Moreover, evidence from our molecular data and rate process measurements highlight that anammox activity is tightly coupled with organic nitrogen remineralization and nitrification. Even though anammox appeared to dominate at stations 2 and 7, we do not rule out the possibility that denitrification could play an important role in Lake Tanganyika, especially following the development of large algal blooms[35].

At station 2, rates of anammox were detected between 130 and 155 m depth, reaching up to 45.4 nM N d$^{-1}$ (Fig. 2g), in line with rates measured by Schubert et al.[10]. In addition, anammox activity was most active in incubations with $NO_2^- + NH_4^+$ additions compared to the only $NH_4^+$ additions, suggesting that nitrite is an important constraint on anammox[53]. The requisite nitrite needed to drive anammox was supplied via nitrate reduction to nitrite, which was highest (32.9 nM N d$^{-1}$) at 145 m depth (Fig. 2h), where also the greatest anammox activity was observed. Moreover, ammonium oxidation to nitrite was detected at the top of the anoxic zone (24.1 nM N d$^{-1}$, 130 m depth; Fig. 2h). The necessary ammonium for anammox at station 2 was, in part, provided by the hypolimnetic ammonium flux, local organic matter remineralization, and to a minor extent by DNRA in the oxygen-deficient zone (DNRA rates measured up to 4.0 nM N d$^{-1}$; Fig. 2h). With depth-integrated estimates (Fig. 3a), indicating that local organic nitrogen remineralization could sustain 65% of the ammonium demand by nitrification and anammox in the ACM.

At the non-ACM station 7, we find that the rate maxima of the various nitrogen transformation processes largely converged at the top of the anoxic zone between the narrow gradients of nitrate and ammonium, and near the small turbidity maximum (Fig. 2j, k: ~145 m). Here, nitrification, anammox, and DNRA reached a maximum of 39.8, 24.5, and 9.1 nM N d$^{-1}$, respectively (Fig. 2o, p), coinciding with the peak of organic nitrogen remineralization (Fig. 2n). The rates of nitrate reduction to nitrite (4.6 nM N d$^{-1}$) and ammonium oxidation to nitrite (8.6 nM N d$^{-1}$) also occurred slightly above and below the oxic–anoxic redoxcline. Our depth-integrated estimates at station 7 (Fig. 3b), indicate that local organic nitrogen remineralization could support 8% of the ammonium demand associated to nitrification and anammox. Alternatively, the large hypolimnion ammonium flux and the higher rates of DNRA contributed to the vast majority of the ammonium requirements by nitrification and anammox.

**The role of the ACM in nitrogen cycling and its fate in Lake Tanganyika.** The community of phototrophs in the ACM at station 2, which was comprised mainly of Chlorobiaceae (21.6%) and to a lesser extent by Synechococcales (1.4%; Supplementary Fig. S2b, c), were positioned between the gradients of nitrate and ammonium. Previous studies have demonstrated that *Chlorobium* and *Synechcococcus* serve as important sinks of dissolved inorganic nitrogen (e.g., $NO_3^-$ and $NH_4^+$)[19,54] as well as organic nitrogen compounds (e.g., urea and amino acids)[55]. In addition, *Chlorobium* has also been shown to fix $N_2$ into biomass, partly offsetting nitrogen loss in the ACM of some stratified lakes[19]. The assimilation of nitrogen, combined with photo/litho-autotrophic $CO_2$ fixation, has the potential to generate significant biomass. Viral cell lysis of intact cells[12], or direct excretions of organic matter by phototrophs[42], may contribute to liberating labile organic matter into its local environment.

We posit that this local production of organic matter by deep photo/litho-autotrophs, combined with the accumulation of sinking organic material from the photic zone at the ACM, likely sustained the active heterotrophic community identified by our algal addition experiments in the ACM. The community of heterotrophs in the ACM further enhanced ammonium remineralization fueling greater nitrogen loss and expanded activity across a broader depth range (Fig. 3a); compared to the non-ACM scenario where nitrogen cycling activity was mostly constrained to the top of the anoxic zone (Fig. 3b). Moreover, if in situ oxygen production by ACM cyanobacteria like *Synechococcus* is active, this may further support microaerobic nitrification and organic matter remineralization in these anoxic waters as proposed elsewhere[13,17,18]. Alternatively, nitrification could be sustained by the vertical mixing of oxygen into the anoxic zone. Given that the ACM is a widespread feature of most marine oxygen-deficient zones, and in various freshwater lakes and enclosed basins, we suspect it serves a similarly important role in regulating nitrogen loss in these aquatic environments.

Factors that control the fate of the ACM in Lake Tanganyika are also important to consider, as possible changes due to climate change may influence the ACM, and therefore affect the lake's capacity to remove fixed nitrogen. Thermal stratification induced by gradual lake warming has been shown to exert significant control over phytoplankton communities as it creates a barrier to vertical mixing and the exchange of nutrients, diminishing epilimnetic primary productivity and enabling light penetration to deep-water phototrophic blooms[56]. In Lake Tanganyika, primary productivity and fish stocks have been shown to exhibit a negative correlation with increasing lake temperature[57,58], along with decreasing light attenuation with depth[59]. Indeed, over the past century, the northern basins have been warming disproportionately faster than the southern basin[60], which has possibly contributed to the provenance of the ACM. This could be further compounded by the presence of density anomalies in the stratified water column (e.g., Fig. 2c; peak A) that physically isolate phototrophic communities at the base of the photic zone. As a consequence of the ACM development in the north, we posit that the enhanced rates of organic matter respiration in deep waters could have contributed to the historic shoaling of the oxycline[34,59]. Furthermore, the shallowing of the oxycline, in combination with active nitrification in the ACM could have enabled the nitracline to persist well into the anoxic zone compared to the southern basin (Fig. 2b, j).

In the southern basin, remote wind-forcing weakens the warming rate in the south[60], and it also deepens the oxycline and enhances nutrient upwelling fueling epilimnetic primary productivity[36]. Higher primary production in surface waters, in turn, results in stronger water column light attenuation compared to the northern basin, which likely limits the formation of an ACM (e.g., station 7). Furthermore, the strong vertical mixing in the water column may prevent the formation of a stable phototrophic community in deep waters. In marine oxygen-deficient zones, for example, phototrophic communities have been shown to occupy a relatively stable water mass[23].

Based on our results, we predict, that amid increasing thermal stratification and declining winds due to climate change in Lake Tanganyika[34,57,58,60,61], the ACM in the northern basin will strengthen and possibly expand southward to the central and southern basins. Under such a scenario, the ACM expansion could shift some primary production to the oxygen-deficient zone and increase the rate of organic matter remineralization, thereby accelerating the removal of fixed nitrogen by a nitrification–anammox coupling (Fig. 3a). The localization of primary productivity in the oxygen-deficient zone and the enhanced nitrogen loss could further weaken the food web supporting fish productivity in the oxygenated surface waters. How much the mechanism outlined in Fig. 3a contributes to

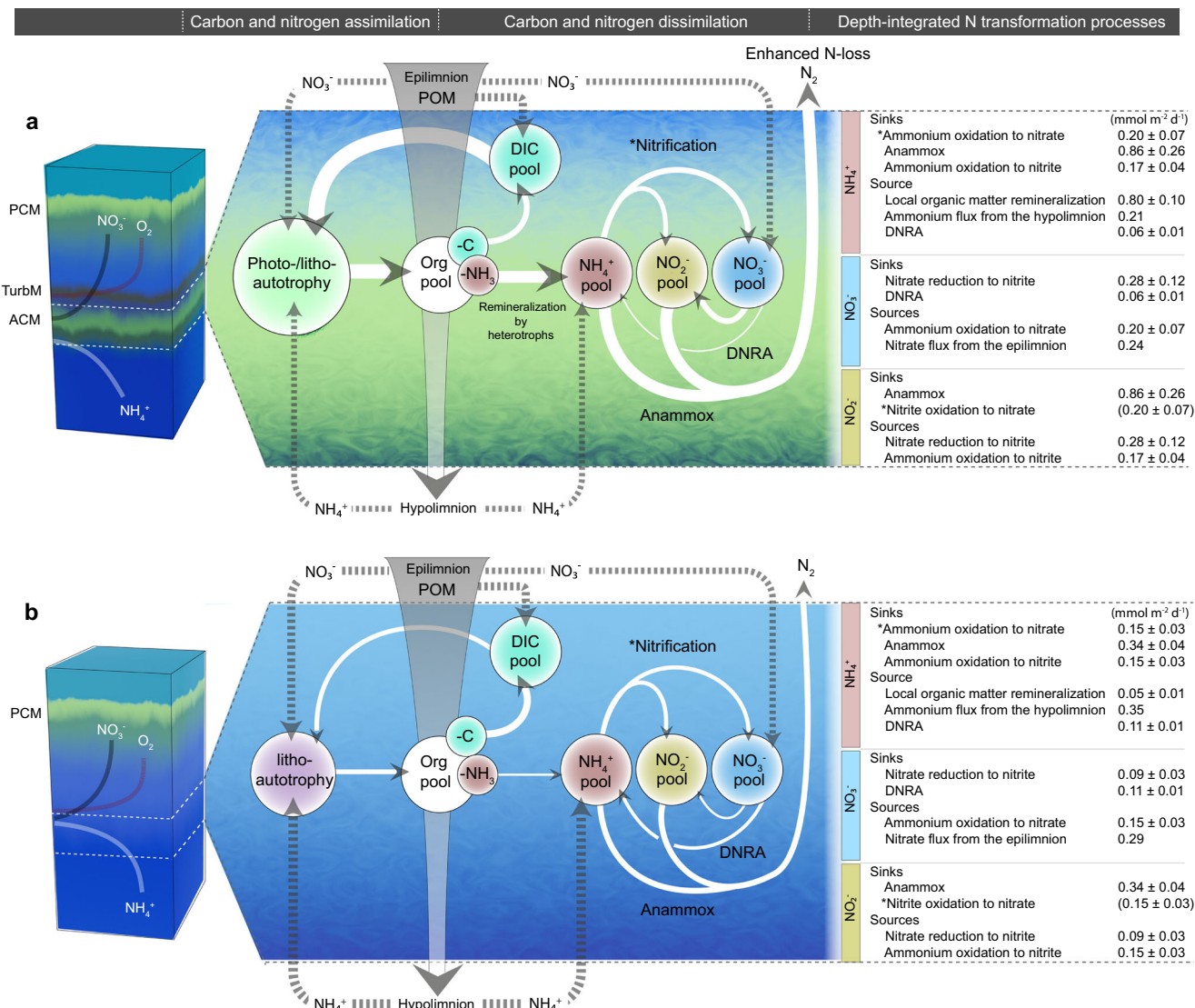

**Fig. 3 Conceptual model of carbon and nitrogen transformation processes in the presence and absence of the anoxic chlorophyll maximum (ACM).** The Lake Tanganyika schematic represents a synthesis of ideas from measurements performed at station 2 (ACM scenario) and at station 7 (non-ACM scenario) in panels **a** and **b**, respectively. The phototrophic community in the ACM comprised mainly of green sulfur bacteria (21.6%) and to a lesser extent by Synechococcales (1.4%; Supplementary Fig. S2b, c). Lithoautotrophic activity might also co-occur in the ACM as a result of the various nitrogen cycling processes (Supplementary Fig. S2e, k). Abbreviations are as follows: PCM primary chlorophyll maximum, TurbM turbidity maximum associated with the enhanced density gradient, ACM anoxic chlorophyll maximum, POM particulate organic matter, DIC dissolved inorganic carbon, DNRA dissimilatory nitrate reduction to ammonium. The various nutrient pools are represented by different colored circles. Fluxes of nutrients are indicated by the dotted gray lines, while solid lines indicate rate transformation measurements, the size of the arrows reflect the depth-integrated values (in mmol m$^{-2}$ d$^{-1}$) in the adjacent panel. Depth-integrations were performed from the top of the anoxic zone (1 μM O$_2$) down to 180 m depth (Fig. 2), with error values representing the standard error. Note, that our $^{15}$NH$_4^+$ additions used to determine rates of ammonium oxidation to nitrate are referred to as nitrification (indicated by the asterisk), these measurements do not specifically imply that comammox (i.e., complete nitrification by a single organism[76]) was involved. In the nitrate budget, we assume that the rate of nitrite oxidation to nitrate (indicated by the asterisk) would be equivalent or similar to the rate of ammonium oxidation to nitrate, based on the measurements from the $^{15}$NH$_4^+$ additions.

constraining fish stocks remains to be explored; but, will be important to address in the future given that Lake Tanganyika sustains the protein-rich diets for millions of people in the surrounding African communities.

## Methods

**Sampling, nutrient analysis, and flux determinations**. We sampled Lake Tanganyika onboard the R/V *Maman Benita* in April–May 2018 and again on a return expedition in April–May 2019, at the end of the rainy season. Our high-resolution vertical analyses were mainly focused at station 2 (5°05′S, 29°31′E; 1,100 m water depth) and station 7 (7°24′S, 30°28′E; 1,300 m water depth), which were sampled

on April 26–28th and April 29th to May 1st, 2019, respectively (Fig. 1). In situ chlorophyll-fluorescence, temperature, oxygen, and PAR profiles were monitored using a Sea-Bird SBE 19plus CTD probe. Photon flux density was determined by calibrating the PAR profiles (Li-Cor LI-193 Spherical Underwater Quantum Sensor) with a surface reference sensor (Li-Cor LI-190R Quantum Sensor). We collected waters in Niskin bottles for nutrient analysis (Day 1), for carbon fixation experiments (Days 1 and 2), and for process rate measurements (Days 2 and 3). Over the course of the multi-day sampling period, we found that oxygen, chlorophyll, and temperature profiles varied very little at each station.

Nitrate and nitrite concentrations were determined onboard using acidic vanadium(II) solution combined with a UV–VIS NOVO 60 spectrophotometer[62]. Ammonium concentrations were determined using a Turner Trilogy fluorometer after its derivatization with *o*-phthadialdehyde, sodium sulfite, and sodium

borate[63]. The detection limits for nitrate, nitrite, and ammonium were 0.39, 0.09, and 0.12 μM, respectively. The detection limits were calculated using the regression coefficient and the residual standard deviation of the daily calibration line. Fluxes of nitrate and ammonium were calculated according to Fick's law, using the measured concentrations and the vertical turbulent diffusivity in the thermocline (between 90 and 300 m) of $1.0 \times 10^{-5}$ m$^2$ s$^{-1}$[31]. Fluxes were calculated across the depths where nutrients exhibited a linear regression gradient ($R^2 = 0.95$, or greater).

TOC and nitrogen (TN) were quantified from water aliquots (20 ml) collected in pre-combusted glass vials. The pH of the sample was adjusted to pH 2 by adding HCl (30%, ultrapure), and stored at ambient temperature. TOC was determined by high-temperature catalytic combustion using a Shimadzu TOC-VCPH/CPN instrument. Total dissolved nitrogen was detected using a TNM-1 module. The final sample TOC and TN concentrations were determined from standard preparations.

**Carbon fixation experiments**. Carbon fixation experiments and rate calculations were performed according to Schunck et al.[64]. Briefly, at each sampled depth triplicate bottles, representing a control (no added label) and two duplicate experiments (with amended $^{13}$C–HCO$_3^-$), were carefully filled off from the Niskin. Waters from the oxygenated epilimnion were collected in 4.5 L polycarbonate bottles, while waters from anoxic depths (e.g., from the ACM) were collected in 5.3 L glass bottles. The polycarbonate bottles were capped with a polypropylene membrane, while the glass bottles were capped with a butyl rubber stopper. To each bottle (except for the controls), 4.5 ml of $^{13}$C-bicarbonate solution (1 g $^{13}$C bicarbonate in 50 ml water; Sigma Aldrich) was added. The experiment bottles were incubated for approximately 30 min under shaking, afterwards a subsample was collected for the determination of the labeling percent, which averaged 3.7%. The headspace was refilled with water from the same depth and then incubated in on-deck incubators covered with shaded light filters (LEE Filters) representing the irradiance and light spectrum at the respective sample depth (Supplementary Fig. S1). Incubations in the deep anoxic zone were shaded with a black aluminum foil that would offer only residual light penetration, mimicking extreme light conditions at this depth. Hence, the measured rates of CO$_2$ fixation in the anoxic zone would include both non-photosynthetic and photosynthetic activity. Even though the light levels were not monitored in the incubations, our measured rate at the ACM of 0.42 μM C d$^{-1}$ (Fig. 2d), was in close agreement with values reported in other green sulfur bacteria containing chemoclines of 0.55–2.04 μM C d$^{-1}$[19,65].

Carbon fixation incubations were terminated after 24 hours by filtration on GF/ F filters (Whatman, UK), the filters were then oven-dried (60 °C for 48 h) and stored at ambient temperatures. At land-based facilities, filters were fumed under HCl atmosphere to remove residual inorganic carbon. The $^{13}$C enrichment was analyzed by an element analyzer EA-IRMS (varioPYRO cube, Elementar coupled with an IsoPrime IRMS, GV Instrument). In the anoxic collected samples, the temperature differences between the in situ water temperature and the on-deck incubation temperature averaged 4 °C, with some extremes reaching up to 8 °C. Due to the relatively small mean temperature difference, the determined rates of CO$_2$ fixation were not adjusted for temperature. Furthermore, the anoxic waters collected at stations 2 and 7 were not purged with helium, therefore trace amounts of oxygen contamination could contribute to enhanced non-photosynthetic carbon fixation, such as nitrification. However, as mentioned earlier, our measured rates of carbon fixation were not anomalously high compared to other chemocline carbon fixation studies.

**Process rate measurements**. Isotope $^{15}$N-labeling experiments were performed on waters collected from stations 2 and 7 (12 depths per station representing both oxic and anoxic conditions) following an established protocol[66]. The $^{15}$N addition experiments were as follows: (Exp1) $^{15}$NO$_3^-$; (Exp2) $^{15}$NO$_2^-$ + $^{14}$NH$_4^+$; (Exp3) $^{15}$NH$_4^+$ + $^{14}$NO$_2^-$; (Exp4): $^{15}$NH$_4^+$; (Exp5) $^{13}$C/$^{15}$N-lyophilized algae biomass (98% $^{13}$C- and $^{15}$N-labeled *Spirulina platensis*, Sigma-Aldrich); (Exp6) $^{13}$C/$^{15}$N-lyophilized algae biomass + allythiourea. The concentration of added nitrate, nitrite, ammonium, and allythiourea were 20, 10, 10, and 84 μM, respectively. At concentrations of 100 μM, Martins-Habbena et al.[67] have shown that allythiourea is a partial inhibitor of ammonium oxidizing archaea, although its effects on freshwater clades identified here (Supplementary Fig. S2e, k) remains untested. Martins-Habbena et al.[67] also find that allythiourea is a strong inhibitor of ammonium oxidizing bacteria. The added lyophilized algal biomass, composed of a mixture of organic containing compounds such as amino acids, DNA nucleotides, acetamide, and creatine (in order of most abundant organic nitrogen sources), is representative of labile organic matter with a C:N ratio of 3.2. The lyophilized algal material was prepared fresh prior to each station (80 mg resuspended in 5 ml MQ). The added algal stock represented roughly 50% of the TOC background as verified by TOC/TN analysis (added substrate concentrations ranged from 90 to 150 μM depending on the sample depth; Supplementary Fig. S3d, l).

For each $^{15}$N-label addition experiment, a 250 ml glass serum bottle was filled directly from the Niskin and overflowed two- to three-times the serum bottle volume. Filled bottles were then immediately capped with a butyl rubber stopper and crimped with an aluminum cap. The waters collected from anoxic depths were purged with Helium for 15 min, and during the purging, the $^{15}$N-label was added and mixed. Oxic collected waters were not purged with Helium in order to

maintain ambient oxygen concentrations. The $^{15}$N amended waters were transferred to 12 ml exetainers (Labco, UK) and sealed with helium purged caps[68]. Exetainer experiments were incubated in the dark at ambient temperatures and terminated at designated time points (0, 6, 12, 24, and 36 h) with the addition of solid Cu(I)Cl (~0.15 w/v final concentration), which is a general inhibitor of cellular processes[69].

At land-based facilities, we added a 2 mL helium headspace to the exetainer samples. From the headspace, the production of $^{14}$N$^{15}$N and $^{15}$N$^{15}$N was determined by gas chromatography isotope ratio mass spectrometry (GC-IRMS; Isoprime, Manchester, UK equipped with an autosampler). Following the determination of labeled N$_2$ gas, the production of dissolved $^{15}$N substrates ($^{15}$NH$_4^+$, $^{15}$NO$_2^-$, and $^{15}$NO$_3^-$) were measured after their chemical conversion to N$_2$ gas. For the determination of ammonium production from $^{15}$NO$_3^-$ and $^{15}$N-algal additions conversions were performed using the alkaline–hypobromide method[51,70]. Nitrate and nitrite production was determined after its chemical conversion to N$_2$ using spongy cadmium and with the addition of sulfamic acid[48]. Following these chemical conversions, the production of $^{14}$N$^{15}$N and $^{15}$N$^{15}$N was measured by GC-IRMS analysis. The slope of the linear regression was used to calculate the rates of N$_2$ production as a function of time, and the standard error was derived from the deviation in the linear slope over the five-time points. $t$-tests were applied to ensure rates were significantly different from zero[71]. Rates were also corrected for the $^{15}$N labeling percent.

The production of $^{13}$C-DIC was also determined by the $^{13}$C/$^{15}$N-algal addition experiment (Exp5). To quantify rates of DIC production, a 2 ml aliquot of the sample water was transferred to a 12 ml exetainer. The sample was then purged with helium for 2 minutes and capped. To acidify the sample 50 μL orthophosphorous acid (85 wt%) was added. The samples were mixed and allowed to equilibrate for 15 h at room temperature prior to analysis by GC-IRMS. The production of $^{13}$C-DIC was determined from the linear slope and the final rate was calculated according to $^{13}$C-DIC prepared standard solutions. Due to the relatively small temperature difference (~3.5 °C) between in situ water temperatures and the ambient air temperature, the $^{13}$C/$^{15}$N transformation rates were not temperature corrected. Furthermore, despite the near anoxic conditions in the Helium purged experiments, trace amounts of oxygen contamination (up to 1–2 μM) are difficult to avoid during sample handling, which could support some microaerobic activity. In separate laboratory experiments, we also tested the abiotic release of dissolved inorganic carbon and nitrogen from freshly prepared $^{13}$C/$^{15}$N-algal biomass amended in sterile synthetic media (incubated in the dark and at room temperature). These control experiments found negligible production of DIC and NH$_4^+$, which suggested that the bulk of the remineralization activity reported at stations 2 and 7 can be attributed to biological turnover.

For the δ$^{13}$C$_{DIC}$ determination in lake water, samples were collected in a 12 mL exetainer (prefiltered through a 0.2 μm pore-size). Samples were head-spaced and acidified as above. The δ$^{13}$C$_{DIC}$ was determined by GC-IRMS[72]. The accuracy and precision of the measurements were verified to be less than 5% using the Carrara-Marmor standard (ETH, Zurich). Samples were adjusted against the standard δ$^{13}$C$_{DIC}$ of Carrara-Marmor (+2.1‰).

**Autofluorescent cell enumeration**. Samples (45 mL) collected for cell enumeration by epifluorescence microscopy were immediately fixed in 20% paraformaldehyde solution to a final concentration of 1%. Fixed samples were filtered (20 ml) on board after 8–12 h at 4 °C onto a 0.2 μm pore-size GTTP filter. Filters were stored at 4 °C onboard and were shipped with ice packs. At land-based facilities, filter pieces were stained with 4′,6-diamidino-2-phenylindole (DAPI; 1 μg mL−1) for 10 min at room temperature and then embedded in a mixture of Citifluor/Vectorshield. DAPI and autofluorescent cells were counted on an epifluorescence microscope (Leica). Cell counts were done up to 1,000 DAPI-stained cells from ten random fields of views, with a detection limit of around 10$^3$ cells ml−1.

**Microbial community survey by 16 S rRNA gene analysis**. The analysis of the 16S rRNA gene diversity was performed on waters collected at station 2 on April 29th and at station 7 on May 5th, 2018. Water samples were collected onboard onto a 0.2 μm cellulose acetate filter and immediately fixed in RNAlater (Sigma Life Science) and stored at 4 °C for preservation and transport to land-based facilities. At land-based facilities, the filters were stored at −80 °C until DNA extraction. Filters were thawed and washed with TE buffer (1×) prior to DNA extraction using the AllPrep DNA/RNA extraction kit (Qiagen). Extracted DNA was amplified with universal 16S rRNA primers 341F and 806R targeting the V3–V4 region (Supplementary Table 1), and sequenced with Illumina HiSeq technology to generate paired-end reads (averaging ~250 bp in length; Novogene). The sequences were analyzed using the qiime2 pipeline[73]. Briefly, the raw sequence adapters were trimmed and deblur was applied (involving denoising and dereplicating step) in order to obtain the amplicon sequence variants (AVSs). The taxonomic assignment for the ASVs was done using a Naïve Bayes approach using the SILVA database 132 for V3/V4 region. Unclassified and rare ASVs (frequency < 0.1 %) were filtered out. The final biome file was visualized with the phyloseq packages[74,75] using R software. The average number of raw reads generated per sample were 15,525, while the mean number of filtered AVSs was 5,189. Rarefaction curves were generated for each sample, and are shown in Supplementary Fig. S2f, l.

## Data availability

The 16S rRNA genes obtained in the anoxic zone of stations 2 and 7 have been deposited in the NCBI database under the accession number SAMN15480878 (125 m, station 2), SAMN15480879 (133 m, station 2), SAMN15480868 (150 m, station 2), SAMN15480880 (156 m, station 2), SAMN15480869 (175 m, station 2), SAMN15480885 (125 m, station 7), SAMN15480886 (125 m, station 7), SAMN15480886 (142 m, station 7), SAMN15480887(150 m, station 7), and SAMN15480881 (175 m, station 7). All sequence data can be found under the NCBI BioProject PRJNA644886. Water column nutrients and process rate measurements are available at Zenodo.

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

## Acknowledgements

We thank our TAFIRI research hosts in Kigoma, Tanzania particularly Julieth Mosille, Athanasio Mbonde, and Ismael Kimirei for the warm welcome and for their assistance onboard. We are grateful to the captain and crew of the R/V *Maman Benita* for their assistance and technical support throughout the sampling campaign. We thank the organizational expertize of Mupape Mukuli, the technical support by Christian Dinkel, and the administrative support by Eliane Scharmin. We thank our co-collaborators Ole Seehausen, Catherine E. Wagner, Julian Junker, and Tim Kalvelage in setting up this project. Thanks to Gesa Eirund and Helmut Bürgmann for their helpful discussions and insightful comments on the manuscript. Thanks to Serge Robert for the general mass spectrometry assistance, and for his help in analyzing EA-IRMS samples. The project was been funded by the Swiss National Science Foundation (CR23I2-166589) and the Eawag.

## Author contributions

C.M.C., B.E., K.B.L.B., B.W., and C.J.S. designed the study; C.M.C., B.E., and K.B.L.B. performed the experiments; C.M.C., B.E., K.B.L.B., and C.J.S. analyzed the data; C.M.C. and B.E. conceptualized/developed the ideas for the paper; C.M.C. wrote the paper with substantial input from all co-authors.

## Competing interests

The authors declare no competing interests.
