## [Peer Review File · Nature Communications]

REVIEWER COMMENTS

Reviewer #1 (Remarks to the Author):

The manuscript submitted by Callbeck and colleagues investigate the microbial composition and the carbon and nitrogen biogeochemistry of the secondary chlorophyll maximum found below the oxycline at Tanganyika lake. Over the course of two samplings, the authors studied the distribution of relevant microbial processes related to the carbon and nitrogen cycling by using stable isotopes techniques to trace both carbon and nitrogen processes. In addition, the microbial community was characterized by molecular biology analysis and microscopic counts, providing a better perspective of the microbial ecology of the community. Although the existence and role of a deep chlorophyll maximum is not new, its presence in one of the largest lakes of Africa such as Tanganyika is a surprise.

In general, the manuscript is written mixing up marine and freshwater environments, not being clear if previous studies and general descriptions are referring to one system or the other. Although certain analogies could be established, it should be more focused to the studied environment and, considering that no similar measurements were performed in oceanic SCM, authors should be more cautious extrapolating the results.

Although a number of techniques and rates were measured or estimated, methodology is not clear and several questions arise that should be addressed in order to get a better idea of the processes measured and the reliability of the data. Several microbial processes were estimated with incubations but the conditions of the incubations and the experimental procedures are sometimes not well described.

Overall, I consider that it is an interesting and relevant manuscript for Nature communications but some improvements should be done in the text to be accepted for publication.

TITLE:

Title is too vague and does not represent well the information provided in the manuscript. Deep and secondary chlorophyll maximum are not the same, being in this case a secondary chlorophyll maximum. In addition, the results were obtained in the Tanganyika lake and, even if the system is analogous to the oceanic Oxygen Minimum Zones (OMZs), the results can not be considered universal for all oxygen deficient waters.

INTRODUCTION

There is a general tendency to mix references from marine and freshwater environments without make any distinction. Although the SCM found could be analogous to the marine OMZs, the use of references should be more related to the studied environment. In OMZ SCM, the SCM is almost exclusively composed by Prochlorococcus. Contrary, the authors found Synechococcus as the main microalgae in Tanganyika, being similar to the deep algae found in the Black Sea, and the same ecology than the marine Prochlorococcus should not be assumed.

L90: Previous studies in SCM have shown the dominance of Prochlorococcus, but not Synechococcus.

L91-92: The references for lakes are about the deep chlorophyll maximum (DCM) in lakes, which it is not the same than the SCM found at the OMZs.

RESULTS AND DISCUSSION

Temperature profiles are not shown in the manuscript. Stratification and water layers would be more clearly identified if the temperature and density profiles were plotted instead of the Brunt-Väisälä frequency. I would recommend including the temperature in Fig. 1 or Fig. 2 (preferable) or in supplementary material.

L119-end of the section: Considering the primary chlorophyll peak and the distribution of nitrate, it is not clear to me why the chlorophyll decreases if there are plenty of nitrate and the light is enough to produce another deep chlorophyll peak below 130 m. In the OMZ SCM, a specific group of Prochlorococcus has adapted to the dim and anoxic conditions in where nutrients start to be available. However, in this case, there are enough nutrients in the water column after 40-50 m, suggesting a light limitation after 50 m. In Station 2 there are also several layers with high turbidity, being surprising the transmission of enough light for Synechococcus to grow at 140-170 m depth. Could the authors provide a complete light profile (in $\mu\text{mol photons m}^{-2} \text{ s}^{-1}$) and discuss this discrepancy?

Cell abundances are much higher than the reported for Pacific OMZ SCM. Considering the high abundances of Chlorobiaceae in the rRNA (up to 22-23% compared with 1.5% for *Synechococcus*) and the lack of *Synechococcus* cells (and data) in upper depths, how reliable are the *Synechococcus* cell counts? Could the distribution of *Synechococcus* be an incorrect match of fluorescent cells? Could be possible to differentiate *Synechococcus* from *Chlorobium* cells by fluorescence microscopy?

Authors should also consider the manuscript of Callieri et al. (2019, "The mesopelagic anoxic Black Sea as an unexpected habitat for *Synechococcus* challenges our understanding of global "deep red fluorescence"." ISME J 13(7): 1676-1687). At both stations fluorescence signal increases below the oxycline and could be caused by a non-photosynthetic growth of *Synechococcus*.

L154-167: The calculations for the ammonium supply from the bottom layer seems to be based on a manuscript for the injection of nitrate caused by eddies (McGilucuddy 1997) so further explanations of the modelling assumptions and conditions are required.

L170-end of the section: The use of labelled lyophilized algal biomass could provide an idea of the organic matter remineralization, but I think it can not be named "in situ rates" (L172). Moreover, it is not so clear to me that the addition of fresh and labile organic matter could produce reliable remineralization rates instead of potential remineralization rates. More details from these incubations are needed and it will be better if the authors include some references with comparison with other procedures. In M&M it is said that the amount of material added were equivalent to approximately 50% of the in situ values but it is not clear what do the authors refer to, dissolved or particulate organic carbon? If those data are available, it would be convenient to show the data to compare the rates. In addition, the enrichment should be specified in terms of carbon pool (DOC or POC) and total amount added.

The comparison between the rates of carbon fixation and organic carbon remineralization are largely unbalance for both epilimnion and OMZ. Carbon remineralization rates are 1-2 orders of magnitude lower than the carbon fixation, even in the OMZ of the station 7. Could the carbon fixation rates be overestimated? Maybe the opposite, could the respiration of organic material be underestimated? Some critical review of these data and discussion are needed.

Changes of $\delta^{13}\text{C}$ -DIC are also discussed in this section but no error bars are provided in the data. Considering the small changes measured and discussed, what is the methodological error in the $\delta^{13}\text{C}$ -DIC determinations? Are the discussed changes significant?

L193: It is not clear to me why the enhanced remineralization and the rapid oxygen decrease is related to the higher penetration of nitrate. Contrary, enhanced nitrification could lead to deeper nitrate penetration but also to higher nitrate concentrations, which it is not obvious in the figure. In addition, high nitrification rates should be also observed in the carbon fixation and $\delta^{13}\text{C}$ -DIC as it is an autotrophic process. Could the authors make this section more clear?

L196-205: The manuscript assumes that the measurement of carbon fixation in the oxygen deficient zone can be attributed to photosynthesis. However, if other N-related autotrophic processes of anammox and nitrification (25 + 50 nM d⁻¹) are subtracted from the rates of carbon fixation inside the anoxic zone (ca. 400 nM d⁻¹), the assumed photosynthesis would exceed by a factor of 10 the organic matter remineralization. To maintain anoxic or very low oxygen conditions, net oxygen consumption must exceed the production of oxygen, being not compatible with the data shown.

L216: Algal addition experiments were performed in the dark and in theoretical anoxic conditions for the samples of the oxygen deficient zone. Authors claim that high nitrification rates are coupled to the production of ammonium however, what is the oxygen source for the nitrification? If no oxygen is produced in darkness and there is anoxic, no nitrite or nitrate could be produced from ammonium. In addition, no detectable $^{15}\text{NH}_4^+$ was measured in the algal additions without allulthiourea but, what about in the samples with algal additions with allulthiourea?

For the same experiments. Labelled N_2 was detected and only attributed to anammox, what about denitrification? If nitrification was tightly coupled to the ammonium release, could the denitrifiers consume a fraction of the labelled nitrite too?

L226-246: Considering the rates and ratio of the DIC and N release from the experiments, how did the authors check if there was a significant chemical release of nutrients? Freezing has been proved to release the intracellular nutrient content of microalgae (ex. Lomstein et al. 1990. Marine Ecology Progress Series 61: 97-105). C:N ratio values of 4 and 3 would correspond to the degradation of a few

amino acids, highly enriched in N, but away of the decomposition of the microalgal biomass in general. Although the preferential degradation of N compounds could be also possible, reliable controls should be performed to validate these results.

L253: The lack of N₂ production in nitrate amended experiments do not imply the absence of denitrification but only complete denitrification. Nitrate and nitrite reduction are both denitrification processes so, similar to the marine OMZs, denitrification might provide the N compounds for anammox. Experiments with labelled and unlabeled nitrate and ammonium were performed so it must be clear if denitrification as nitrite reduction or anammox occurred in the samples, being now not clear in the text or the figures.

L260: It is confusing the use of "upper oxycline" here, at 130 m depth it seems to be the lower part of the oxycline. Similarly, "the base of the oxycline" is used to describe depths at 147-180 m for the Station 7, being not clear in the figure what the authors refers to.

L274 (and throughout the text): what do the authors refer when talk about "nitrification"? Ammonium oxidation is sometimes described independently and then, does nitrification refers to nitrite oxidation? Complete nitrification?

L275: 15N experiments were performed in "anoxic" conditions and in darkness. Considering the lack of internal oxygen production, could the nitrification rates be produced by oxygen contamination during the incubations?

L307: This sentence seems to be a bit pretentious. Organic matter remineralization can occur in both oxic and anoxic environments so SCM does not expand organic matter remineralization.

L310-end of the paragraph: Synechococcus can also grow in the dark conditions found below the oxycline in the Black Sea by heterotrophic metabolism (Callieri et al. 2019. ISME J 13(7): 1676-1687), therefore its simple presence does not imply active carbon fixation.

L339: Deep chlorophyll maximum is not the same than SCM. Authors could include this manuscript (Marquez-Artavia et al. 2019. Deep-Sea Research Part II-Topical Studies in Oceanography 169: 104686.) describing temporal variability and relationship of the OMZS SCM with density gradients.

MATERIAL AND METHODS

L376-378: Were the detection limits of the methods theoretical or calculated?

L385: TOC and TN data are not shown in the manuscript. It would improve the discussion of some aspects of the data.

L392: The manuscript use for reference details how to measure N-fixation rates, but it is not related to carbon fixation experiments. Provide a valid procedure and reference.

L393: One single control and two replicates is not enough to provide robust determinations of the rates. In addition, what about the dark fixation rates? Could the authors use them to estimate C-fixation by non-photosynthetic processes?

L395: Was the water from the oxygen deficient zone de-oxygenated? The entrance of oxygen could enhance non-photosynthetic C-fixing processes (such as nitrification).

L398-400: What was the final 13C concentration or the labelling percent?

L401-404: On deck incubations does not seems the best option for deep-water incubations. What was the temperature of the incubation and the in situ temperature of the samples? Temperature profiles should be also shown. In addition, what were the lights levels in the incubations? The values are especially relevant for the deepest samples (SCM depths) so they should be shown as the use of high light levels would also produce unrealistic C-fixation rates.

L415-417: TOC-TN data should be shown. How much algal biomass was added? How was the procedure? Direct addition of the dust, addition of the resuspended material... In addition, some more details of the algal material must be provided: What algae were added? What was the C:N ratio of the material? How many replicates were used? What kind of control experiments were used?

L423: Samples were incubated at ambient temperature but what was the in situ temperature? Were rates temperature corrected?

L425: Incubations were terminated with the addition of Cu(I)Cl, could you provide a reference for the use of that compound? I found that it is practically insoluble in water, how much did you add to ensure the stop of biological processes?

L454: If samples were stained with DAPI, all microorganisms were stained and enumerated; do you have total cell counts? It would help for the discussion of microbial abundance. In addition, how were

the *Synechococcus* cells distinguished from the rest of microorganisms (including *Chlorobium*)? L458-471: Microbial community was characterized by metagenomics but this information is barely used in the manuscript. In addition, results shown in Fig. S1 are also quite scarce. Why do not show the relative abundances of the most relevant functional groups? Which groups were included as "nitrifiers"? I think it would be better to separate in ammonium oxidizing bacteria and archaea, nitrite oxidizing bacteria, denitrifiers, etc...

Reviewer #2 (Remarks to the Author):

The authors present a study of nitrogen cycling in Lake Tanganyika, i.e. biological N-cycling rates, and physical N-fluxes. The northern basin has an SCM at about 150 m, while the southern basin does not. They argue that the SCM is a "hotspot" of N-cycling and fixed N-loss. I think the article and subject matter are suitable for publication in *Nature Comm.*; this is the one of the few examples of the importance of the SCM in lakes. However I have numerous questions and comments I would like to see addressed beforehand. So, in no particular order of importance, are my other questions and comments.

- 1) I think the title is misleading. When I see "oxygen minimum zone" I think about the global ocean, not Lake Tanganyika. I think the lake name needs to be in the title.
- 2) Line 46. "Stable isotope" makes me think of natural abundance. I would say "Stable isotope tracer experiments".
- 3) Introduction first paragraph. They use "oxygen minimum zone" and "oxygen deficient zone" apparently interchangeably. To many oceanographers they are different. All O₂ profiles have a minimum concentration of oxygen, whereas oxygen deficient means not enough oxygen for aerobic respiration to dominate. Define term(s) and stick to them. Also, not all eutrophic upwelling zones have "vast OMZs, and finally the three large denitrifying ODZs extend well off shore 500-1000km.
- 4) Line 93. Pronoun reference is unclear. Do they mean *Synechococcus* or anammox? Both of these require nitrogen assimilation and carbon fixation.
- 5) Line 98. Define "oxic-anoxic transition". Is it the actual narrow zone where the transition happens or is it larger? Where does it start and end? Fuchsman et al show the SMC is in the functionally anoxic zone. Looking at their profiles in figure 2, I would say their SMC is in the functionally anoxic zone also. It is clearly where the oxygen profile quits going down. If that's not functionality anoxic then the transition zone appears to go to at least the bottom of their plots.
- 6) Paragraph starting on line 99, first sentence. I don't get where they are going with "heterotrophs". It seems clear from the literature that photosynthesis is important in the SCM (their references 9 & 10). Are they trying to say that chemoautotrophic production by anammox is a significant OM source? If they argue, I would argue that it's a very minor one because much more nitrogen is consumed than nitrogen assimilated for growth.
- 7) Lines 101-02. "freshwater world"? Just say "largest freshwater oxygen deficient zone in the world"?
- 8) Line 103. What is a "community of nitrogen cycling processes"? Just say "... supports an active nitrogen cycling community".
- 9) Line 104. North-south gradient. I don't see so much of a gradient. I would call it a contrast between presence and absence.
- 10) Line 108. Again, insert "tracer" before the word "experiments".
- 11) Results and discussion section. Here they start presenting N-transformation rates. They have anammox rates from the 15NH₄ tracer experiments. Presumably they measured 29N₂. Did they measure 30N₂? If they only measured 29N₂ they can't separate anammox from denitrification. Did they measure denitrification?
- 12) Lines 130-31. Forget the biogeochemistry and physics (which covers all four basic sciences) and just say below the primary chl. Max we observed distinct profile differences.
- 13) Line 134. Turbidity peak. This is an interesting observation but I don't think it is discussed later.
- 14) Line 160. I don't understand the Predicted NH₄. Their reference 25 is about the Sargasso sea, a

totally different system. McGillicuddy & Robinson try to calculate the upward flux of NO₃ supporting Primary Production in the S.S. This is a relatively straightforward system. Here they are in a complicated N-Cycling system with all the processes taking place in the hotspot zone they are trying to model. Give me some more specifics about how the prediction was done. It is not clear to me from looking at the reference.

15) Sentence beginning on line 162. Basically this sentence says that the missing ammonium is due to processes that make it missing. This is obviously true but doesn't really say anything.

16) Line 165, "gradients exceeded concentrations". This is comparing apples and oranges. Which do they mean, gradients and gradients or concentrations and concentrations? Also, the gradient (curvature) in the ammonium profiles in figure 2a indicates there must be consumption in that zone or increasing production with depth.

17) Line 210. Here they talk about allylthiourea inhibition. I'm not so familiar with lakes, but in the oceans nearly all nitrifiers are archaea. Archaeal nitrifiers are not very sensitive to ATU (Martins-Habben et al 2014, Env. Micro.) In the text they say they detected bacterial and archaeal nitrifiers. What is shown in Suppl. Fig. 2? If there are archaeal nitrifiers and no nitrification was detected with added ATU does that mean they were unimportant?

18) Paragraph beginning on line 226. In the SCM OrgC recycling is about 25nM/d and org N recycling is 2 times that. Why is this? They suggest some organism is preferentially recycling NH₄ what organism is doing this without recycling much C at all. It's not the NO₃ reducers because they are after the C. and it's not aerobic organisms because this is the ODZ. They talk about the changing C:N ratio in the SCM from 7.7:1 to 3.1:1 deeper in the water column. However, the observed regeneration ratio is ~0.5:1. At the end of the paragraph they talk about preferential degradation of amino acids, but there are no amino acids with C:N ratios of 0:5. Have I missed something here?

19) Line 236. Here and beyond they talk about C:N ratios that are never shown anywhere. Put a figure in the supplemental.

20) Line 253. Now you tell me about the 30N₂ results. Put this earlier.

21) Line 305. This section is all about the importance of the SMC for enhancing nitrogen cycling and loss in the ecosystem. They measure all sorts of N-cycling rates and fluxes and then derive their conclusions. All of their rates except C-fixation are in nM/d. But the one thing they neglect to point out is that their measured C-fixation in the SCM is about 0.5 μ M/d which is 500 nM/d. Given a more or less co-occurring oxygen production of 500 nM/d this about 10 times bigger than their N-cycling rates. This is a lot of oxidation potential and it should have a big effect on the N cycling rates in the SCM. Oxygenic reactions in the SCM are only briefly discussed.

22) Paragraph beginning on line 335. The climate models mentioned here that predict expanding OMZ are talking about the large marine OMZs not lake OMZs. The ocean OMZs are quite different from the Lake Tanganyika. First heterotrophic denitrification is an important process. Secondly, there is not a large NH₄ flux up from the bottom and third, The SCM only exists at the top of a very much thicker OMZ.

23) Line 416. Delete "was later"

Figure 2.

1) Is the apparent chlorophyll in the deep waters real and why does it seem to increase in the ODZ at station 7?

2) Also the dotted red line mentioned in the caption is actually the oxygen profile in the figure. There is a dashed horizontal black line in some of the panels which likely the oxygen minimum zone. If so it would be helpful if this was in all the panels so we compare them side by side e.g. it is not clear to me where the OMZ starts in panels a and i. At the right of the panels they show the processes, they show "oxygen deficient zone". The ODZ appears to extend only a little deeper than 200 m. Is the water below that not oxygen deficient. Is the bracketed ODZ the same as the SCM layer in figure 3 even though there is no chlorophyll in the lower ODZ

Supplemental figure 2.

In the supplemental figure how do you get a local increase in % surface illumination at about 150 m. At station 2 the light level at about 120 m is only about 0.1%, but it is almost 0.2% at 150 m. Where

does the added light come from?

Reviewer #3 (Remarks to the Author):

This manuscript by Callbeck and colleagues reported their study on how the establishment of the secondary chlorophyll maximum (SCM) could accelerate nitrogen-cycling processes and nitrogen loss in the oxygen deficient zone of Lake Tanganyika. They conducted an impressive, comprehensive suite of incubation experiments with ^{15}N -labeled tracers to measure rates of nitrification, anammox, nitrate reduction, DNRA and remineralisation of algal organic matter ($^{13}\text{C}/^{15}\text{N}$ -dual-labeled) into CO_2 and NH_4^+ , as well as that of CO_2 fixation via ^{13}C -DIC-incubations, further complemented with DIN, d^{13}C -DIC and CTD profiling. They examined these processes at two different stations, one in the north basin (Station 2) characterised by the presence of an SCM within the oxygen minimum zone, and one in the south (Station 7) without SCM. Their results have clearly demonstrated that the presence of SCM has enhanced organic N remineralisation and N-loss from the system. As Lake Tanganyika is the second-largest freshwater lake on Earth and the largest anoxic freshwater system, these findings are important for the N-budget of freshwater systems that have generally been understudied in global N-cycling. Moreover, due to its analogy with marine oxygen deficient zones and that SCM is increasingly found to be relatively common therein, findings from this study could lend important insights into how marine N-budgets may change in future ocean should SCM become more widespread.

Nevertheless, while the main conclusions drawn are sound, the developing arguments in the Discussion and data interpretation are sometimes confusing and illogical, with tendency of overinterpretation over certain minute details, and the modeling approach, flux calculations and molecular analyses lack details.

First of all, the assertion that stratification was enhanced at Station 2 is not fully evidenced in the presented data, and neither is the causality of stratification to SCM development. The temperature gradient (dT/dz) shown in Fig 1 is indeed a good proxy for density gradient across the transect, but the highest values, hence strongest gradients, were actually found in the upper epilimnion in the South basin where Station 7 was located, and this also coincided with higher chl a. At least from the colour scale used, it is unclear that Station 2 experienced enhanced stratification, as the values are similar between the two stations at and below the oxygen deficient zone. The Brunt-Vařsala frequency (N_2) in Fig 2 also indicated similar values throughout the water column at the two stations, except perhaps a thicker layer of enhanced stability (high N_2) and a minor secondary maximum $\sim 105\text{m}$ at Station 2 (though the max N_2 was actually higher at Station 7). While these might be better indication of more stable/ stratified waters at Station 2, the values at/below oxygen deficient zone were indistinguishable across the stations. The authors overlaid the stratification signals (lines 130-140) in my opinion. In addition, Station 7 was labelled with enhanced upwelling in Fig 1 (and in text), but there was no supporting evidence shown. In general, there is also a tendency of overinterpretation of depth profiles, including both physical and biogeochemical data (e.g. lines 119-165, 187-205).

Meanwhile, there seem to be an important omission in the hydrochemical background description – O_2 distribution and the depth where oxygen deficient zone started at the two stations. Oxygen penetration was shallower at station 2 compared to Station 7, which most certainly have impacts on water column biogeochemistry. Although the authors to some extent attributed this to increased remineralisation of organic matter at the turbidity maximum at around 110m (lines 193-5), the measured C-remineralisation rate is not much different from that measured around 100m at Station 7 (no measurement was taken at exactly the same depth) that it is insufficient to fully explain the dramatic oxygen decline. This necessitate another explanation. The authors then further suggested that the above rapid O_2 utilisation led to deeper penetration of nitrate into oxygen deficient zone – On one hand, this argument makes little sense, as the depletion of O_2 would prompt earlier use of nitrate

as an alternative electron acceptor such that it would decline in concentration more rapidly at a shallower rather than deeper depth – contrary to what was claimed. On the other hand, nitrate penetration depths are practically the same at the two stations, that it was the O₂ penetration depth that became shallower at station 2 instead. This larger overlap between oxygen deficient zone and presence of nitrate would certainly impact N-biogeochemistry, thus the underlying causes and consequences deserve more detailed discussion and analyses.

Moreover, the shallowing of oxygen deficient zone at station 2 coincided with the occurrence of SCM – what are the potential explanation behind, and would there be any mechanistic relationship between? There have been suggestions that SCM occurs only when the oxygen deficient zone overlap with the 1% downwelling blue (490nm) irradiance. The PAR at the two stations appeared similar, except for the subsurface peak around the SCM depth which indeed overlapped with oxygen deficient zone – though why a peak in PAR there also need further explanation.

The SCM itself was thought to be made of primarily cyanobacteria like *Synechococcus*. However, the sequencing data showed that *Synechococcus* only had a small contribution to the community ($\leq 1.5\%$), whereas there were a lot more green sulfur bacteria *Chlorobium* (up to 25%) that could also be counted as autofluorescent cells. This is quite different from how we understood SCM. How would this impact the water-column biogeochemistry? While on this topic, little information has been provided for the 16S rRNA gene sequencing results. How many reads were generated per sample? Are the values for % community referring to total number of sequencing reads obtained for each sample, or ASV? And if your dataset has been rarefied?

Based on concentration profiles of ammonium, the authors further calculated its diffusional fluxes, but it is not entirely clear over what depth or depth range those fluxes were calculated for, and that the use of what I understood as a single value for turbulent diffusivity for the entire water column seems unrealistic. A model was then applied to 'predict' ammonium concentration according to density. What are the boundary conditions used and the assumptions behind? The model used was based on simple one for nitrate in the Sargasso Sea with single source and sink and under the assumption, amongst others, that sink is much greater than supply. As there are multiple sources and sinks for ammonium within the oxygen deficient zone alone, the applicability of this model is highly doubtful. A reaction-diffusion model, such as the one developed by Berg et al (1998, L&O) would have been more appropriate.

On the discussion on remineralisation of organic nitrogen versus carbon, the production of DIC was compared against that NH₄. On one hand, the authors should be cautious that the measured are not strictly gross rates as the consumption of DIC and NH₄ at the same depths were just as fast or faster, and that the impact would be at different degrees between the two due to the different processes involved in their respective consumption (apart from assimilation to organic matter). On the other hand, why did you expect them to be in similar magnitude, and even performed t-test, etc. (lines 226-235)? Even without the interference from concurrent production/consumption, their gross production should follow Redfield ratio and not 1:1. Besides, why used the total organic carbon to total nitrogen (TN) ratio to examine preferential organic nitrogen remineralisation? The TN, if not having been filtered, includes all forms of organic and inorganic, dissolved and particulate nitrogen (PON, DON, NH₄, NO₂, NO₃). Using this to compare against TOC doesn't tell you how much N has been remineralised – the value of TN itself would just stay the same before and after remineralisation – unless anammox/denitrification/N₂-fixation occurred.

On the balance between autotrophy and heterotrophy, while it is apparent that there is definitely higher CO₂ fixation than remineralisation at the upper boundary of the oxygen deficient zone at Station 2, and also higher than observed at station 7, the authors should bear in mind that the detection limit of the two process rates are different (probably by an order of magnitude). Stable isotope tracers are not as sensitive as e.g. radiolabelled tracers, to measure CO₂ fixation considering the amount of water sample used, and it is not exactly a fair comparison against remineralisation to

examine the 'balance' between the two processes. Hence, some of the conclusions drawn especially regarding whether it was a predominantly autotrophic or heterotrophic community in the oxygen deficient zone and immediately above, may be overinterpreted (lines 185-205).

While it is obvious that the station with an SCM showed greater overall N-loss, it would have been good to examine how the amount of organic matter actually affect N-cycling/N-loss. Although TOC/TN was measured, it was unclear how many samples were taken and the data were not shown. Also, since nitrate/nitrite/ammonium have been measured, an estimate could be made for at least TON. Both TOC and TON values could be compared with process rates data to further evaluate the influence from organic matter, as hypothesised in Introduction and also shown by others in at least marine oxygen deficient waters.

Overall, this study has presented a very nice set of data that in itself deserve publication, but the manuscript in its current state needs more work on its data interpretation and Discussion, to focus on what the data were actually showing, whilst avoiding overinterpretation and unnecessary speculations.

Other specific comments:

Line 46: should bear in mind that your measured fluxes are not strictly gross rates though. Should correct for isotope dilution for the concurrent consumption/production processes.

Line 131: "multiple layers with high turbidity" this is an exaggeration considering the turbidity signals

Line 132: I can't see an enhanced 'density' gradient at 110m.

Lines 137-8: Have you measured (or previously) Fe/Mn oxide precipitates here or microbial biomass at this depth here to support this statement?

Line 139: What is the evidence for upwelling?

Line 145: See main comments above on 16S sequencing results reporting

Lines 165-7: This modelled results of apparent ammonium production seems to go against your measured fluxes (Fig. 3) which showed an overall net ammonium loss

Lines 178-8: again, these are not gross rates and should be corrected for isotope dilution

Line 185: Sharp density gradient – I don't see it.

Line 187-8: increased particle retention time – you only have shown turbidity or concentration, which is not equivalent to turnover time

Line 191: the so-called gradual decline is overly subtle and only down to top of oxygen deficient zone

Line 220: only Planctomycetes and not anammox bacteria?

Line 223: single peak or not doesn't matter, it's the magnitude that is of concern here

Line 230: the C remineralisation rates = $\sim 3x$ and $3.5x$ of NH_4 production, meaning that the former 'exceeded' the latter by a factor of 2 and 2.5 respectively!

Line 232, 234: same as above on 'exceeding' by how many folds

Lines 244-6: there were also a number of autotrophs as shown by your annamox and nitrification

rates. As mentioned earlier, your CO₂ fixation rate measurements were not as sensitive your remineralisation measurements.. See earlier major comments on the organic nitrogen remineralisation

Lines 283-6: repetitive from before

Lines 293-4: What algae were in your labelled substrates?

Lines 306+: This section can be shortened as some is rather repetitive

Line 329: also autotrophs were present

Lines 334+: As mentioned in main comments above, the link to stratification is weak based on the evidence and arguments presented here, and so this section is largely speculative

Lines 346: The percentage of cyanobacteria is very low in your sample! There were way more green sulfur bacteria

Lines 355-364: Speculations aside, if you consider the overall biological production of the water column, should also consider the degree of production that has shifted to the SCM. Has it balanced the reduced production in the primary chlorophyll maximum compared to the water column without SCM but higher surface production like Station 7?

Figure 3: Missing in this budget is the nitrite oxidation to nitrate, which would be important for the balance of nitrite as well as that of nitrate. Also, I think a table would suffice, and clearer for comparison purposes. The drawing doesn't add much.

The 'Synechococcus pool' is not justified as a separate pool from 'Org', as you showed that they only contributed little to the microbial community actually, and certainly DIC would go directly into 'Org' as well, as there are other autotrophs beside Synechococcus. Nitrification should be shown as a 2-step process for budget purposes, unless you found comammox. Colour scheme for NO₂ and NO₃ have been mixed up on the budget side.

We greatly appreciate the three reviewers for their support of this work, their attention to detail and their constructive criticism. Below is a point-by-point response to the reviewers' comments, indicated in the red-type face. The grey highlighted line numbers refer to changes made in the revised manuscript (e.g. lines XX-XX). A complete list of references is included at the end of the response to reviewers' comments.

REVIEWER COMMENTS

Reviewer #1 (Remarks to the Author):

The manuscript submitted by Callbeck and colleagues investigate the microbial composition and the carbon and nitrogen biogeochemistry of the secondary chlorophyll maximum found below the oxycline at Tanganyika lake. Over the course of two samplings, the authors studied the distribution of relevant microbial processes related to the carbon and nitrogen cycling by using stable isotopes techniques to trace both carbon and nitrogen processes. In addition, the microbial community was characterized by molecular biology analysis and microscopic counts, providing a better perspective of the microbial ecology of the community. Although the existence and role of a deep chlorophyll maximum is not new, its presence in one of the largest lakes of Africa such as Tanganyika is a surprise.

In general, the manuscript is written mixing up marine and freshwater environments, not being clear if previous studies and general descriptions are referring to one system or the other. Although certain analogies could be established, it should be more focused to the studied environment and, considering that no similar measurements were performed in oceanic SCM, authors should be more cautious extrapolating the results.

Although a number of techniques and rates were measured or estimated, methodology is not clear and several questions arise that should be addressed in order to get a better idea of the processes measured and the reliability of the data. Several microbial processes were estimated with incubations but the conditions of the incubations and the experimental procedures are sometimes not well described.

Overall, I consider that it is an interesting and relevant manuscript for Nature communications but some improvements should be done in the text to be accepted for publication.

We thank the reviewer for their constructive feedback and overall support of this work. The reviewer was correct to highlight that we did not thoroughly distinguish between marine and freshwater environments, and that some of the methodological details were not clear in the original manuscript, which we have now remedied in the revised version. Moreover, we have added additional data, according to the reviewer's suggestion including vertical profiles of temperature, DAPI cells counts, TOC/TN, and light profiles (in $\mu\text{mol photons m}^{-2} \text{s}^{-1}$), which we believe has strengthened the overall findings of the manuscript. We thank the reviewer for the suggestion to include these data.

TITLE:

Title is too vague and does not represent well the information provided in the manuscript. Deep and secondary chlorophyll maximum are not the same, being in this case a secondary chlorophyll maximum. In addition, the results were obtained in the Tanganyika lake and, even if the system is analogous to the oceanic Oxygen Minimum Zones (OMZs), the results can not be considered universal for all oxygen deficient waters.

We thank the reviewer for raising this point and agree that the title was vague in the original manuscript. We have subsequently reworded the title to more explicitly describe the key finding, and

the specific study site. In the revised manuscript it now reads as, “Anoxic chlorophyll maximum accelerates organic matter remineralization and nitrogen loss in Lake Tanganyika”. We have also made a greater distinction between the ecology of marine oxygen minimum zones, stratified basins and freshwater lakes throughout the revised manuscript, particularly in the introduction, please see comments below.

INTRODUCTION

There is a general tendency to mix references from marine and freshwater environments without make any distinction. Although the SCM found could be analogous to the marine OMZs, the use of references should be more related to the studied environment. In OMZ SCM, the SCM is almost exclusively composed by *Prochlorococcus*. Contrary, the authors found *Synechococcus* as the main microalgae in Tanganyika, being similar to the deep algae found in the Black Sea, and the same ecology than the marine *Prochlorococcus* should not be assumed.

We thank the reviewer for their feedback and agree that the microalgae communities in the ACM of freshwater/enclosed basins and the ACM communities observed in marine oxygen minimum zones was not well distinguished in the original manuscript.

In the revised version of the manuscript, we now more carefully differentiate the ecology of the phototrophic communities in the ACM between marine and freshwater environments on **lines 95-111**. It now reads as follows, “In marine oxygen deficient zones, the ACM tends to be dominated by the low-light adapted cyanobacteria genera *Prochlorococcus* and to a lesser extent by *Synechococcus* (Goericke et al. 2000; Franz et al. 2012; Garcia-Robledo et al. 2017; Fuchsman et al. 2019a; Aldunate et al. 2019). The marine ACM, which can occur at a irradiance of $5 \mu\text{mol photons m}^{-2} \text{s}^{-1}$ (Garcia-Robledo et al. 2017), is considered a prevalent feature of all major marine upwelling regions (Goericke et al. 2000; Franz et al. 2012; Garcia-Robledo et al. 2017; Fuchsman et al. 2019a; Aldunate et al. 2019), and can persist year-round within a stable isopycnal layer (Loginova et al. 2016; Márquez-Artavia et al. 2019). Oxygenic photosynthesis by cyanobacteria, has been shown to generate molecular oxygen (O_2) possibly supporting microaerobic nitrification in the ACM (Garcia-Robledo et al. 2017). In euxinic basins and in stratified freshwater lakes, the ACM can comprise of Cyanobacteria (such as *Plankthotrix*, *Phormidium* and *Synechococcus*) (Repeta et al. 1989; Camacho et al. 2001; Manske et al. 2005; Halm et al. 2009; Brand et al. 2016), that have also been shown to generate O_2 in situ (Brand et al. 2016). In addition, the reduced sulfur fluxes in these systems can sustain anoxygenic photosynthetic green sulfur bacteria that occupy the bottom layer of the ACM (Repeta et al. 1989; Camacho et al. 2001; Manske et al. 2005; Halm et al. 2009; Brand et al. 2016), due to their extreme light harvesting capabilities of only $0.001 \mu\text{mol photons m}^{-2} \text{s}^{-1}$ (Marschall et al. 2010). Unlike cyanobacteria, green sulfur bacteria such as *Chlorobium*, produce sulfur, and either assimilate dissolved inorganic nitrogen and/or fix N_2 into biomass partly satisfying local nitrogen loss (Halm et al. 2009). The co-occurrence of anoxygenic and oxygenic photosynthesis in the ACM, is not only common to stratified lakes and basins, but has also been hypothesized to be a widespread feature of the euxinic Proterozoic ocean (Johnston et al. 2009). Both anoxygenic and/or oxygenic photosynthesis in the ACM have a capacity to fix CO_2 into biomass (Thiel et al. 2018), contributing to an important production of organic matter.”

L90: Previous studies in SCM have shown the dominance of *Prochlorococcus*, but not *Synechococcus*.

We agree with the reviewer, and have now made this distinction more clear in the revised manuscript on **lines 95 to 96**, it now reads as “In marine oxygen deficient zones, the ACM tends to be dominated by the low-light adapted cyanobacteria genera *Prochlorococcus* and to a lesser extent by *Synechococcus* (Goericke et al. 2000; Franz et al. 2012; Garcia-Robledo et al. 2017; Fuchsman et al. 2019a; Aldunate et al. 2019).”

L91-92: The references for lakes are about the deep chlorophyll maximum (DCM) in lakes, which it is not the same than the SCM found at the OMZs.

We agree with the reviewer that the secondary chlorophyll maximum is not exactly the same as the deep chlorophyll maximum, which could lead to confusion. For a more unifying definition, we now refer to a phototrophic bloom that occurs in the suboxic/anoxic zone, as an anoxic chlorophyll maximum (or ACM), which could encompass either a secondary chlorophyll maximum or a deep chlorophyll maximum (now revised on lines 89 to 91). We would also like to highlight, this adopted definition is somewhat similar to the terminology used by Márquez-Artavia et al. 2019.

RESULTS AND DISCUSSION

Temperature profiles are not shown in the manuscript. Stratification and water layers would be more clearly identified if the temperature and density profiles were plotted instead of the Brunt–Väisälä frequency. I would recommend including the temperature in Fig. 1 or Fig. 2 (preferable) or in supplementary material.

We agree with the reviewer and have therefore included temperature profiles in both Figures 1 and 2 in the revised manuscript. In Figure 2c, k, we have included the temperature, turbidity and Brunt–Väisälä frequency in the same panel for comparison. The changes in the temperature profile as a function of depth, was the main driver of the Brunt–Väisälä frequency in Lake Tanganyika. We still include the Brunt–Väisälä frequency in Figure 2c, k, as it provides a useful indicator of the deep stratified layer at station 2 (105 m depth), which nicely complements the turbidity profile.

In addition, we have added a more complete description of the temperature profiles and thermocline on lines 134-142. It now reads as follows, “The epilimnion was characterized by a warm (27-28 °C) and relatively well mixed water mass from 0-30 m, with the thermocline occurring from 30-75 m (Figs. 1b and 2c, k). However, station 7, and the southern basin in general, exhibited a steeper thermal gradient throughout the metalimnion than compared to station 2 (Figs. 1c, 2c, k). The steep thermocline at station 7, also coincided with a peak in N^2 , otherwise known as the Brunt-Väisälä frequency (Fig. 2c, k), which also reflects changes in the temperature gradient. The onset of the south-east trade winds (Verburga and Hecky 2009) in May, is known to induce nutrient upwelling and the upward tilting of the thermocline, resulting in a shallow and steep temperature gradient to develop in the southern basin (Ehrenfels et al. 2020).”

L119-end of the section: Considering the primary chlorophyll peak and the distribution of nitrate, it is not clear to me why the chlorophyll decreases if there are plenty of nitrate and the light is enough to produce another deep chlorophyll peak below 130 m. In the OMZ SCM, a specific group of *Prochlorococcus* has adapted to the dim and anoxic conditions in where nutrients start to be available. However, in this case, there are enough nutrients in the water column after 40-50 m, suggesting a light limitation after 50 m. In Station 2 there are also several layers with high turbidity, being surprising the transmission of enough light for *Synechococcus* to grow at 140-170 m depth. Could the authors provide a complete light profile (in $\mu\text{mol photons m}^{-2} \text{ s}^{-1}$) and discuss this discrepancy?

We thank the reviewer for the discussion and agree that providing the complete light irradiance profiles (now shown in Figure S1 and expressed in $\mu\text{mol photons m}^{-2} \text{ s}^{-1}$) provide greater context for the ACM development in Lake Tanganyika.

At station 2, the light irradiance (measured in the morning sun, surface irradiance of $\sim 400 \mu\text{mol photons m}^{-2} \text{ s}^{-1}$) was strongly attenuated with depth and ceased to decrease at $\sim 100 \text{ m}$ (reaching a baseline of $0.05 \mu\text{mol photons m}^{-2} \text{ s}^{-1}$) (Fig. S1a, b). Although the maximum light penetration depth was likely deeper at mid-day given that the surface irradiance could have been up to $\sim 750 \mu\text{mol photons m}^{-2} \text{ s}^{-1}$ (e.g. Fig S1d). Nevertheless, how *Synechococcus* could persist under such extreme light conditions is enigmatic. Niche partitioning between oxygenic and anoxygenic light

harvesting zones has been reported in other highly stratified lakes, even at a vanishingly low light availabilities (Brand et al. 2016). Alternatively, *Synechococcus* in the ACM could thrive via a heterotrophic mode of growth, which has been posited for *Synechococcus* cells identified in the dark anoxic waters of the Black Sea (Callieri et al. 2019).

At station 7, the light penetration (measured at mid-day $\sim 750 \mu\text{mol photons m}^{-2} \text{s}^{-1}$) was strongly attenuated with depth and ceased to decrease at $\sim 80 \text{ m}$ (Fig. S1), well above the top of the anoxic zone (145 m). The relatively high primary production sustained by upwelling in the southern basin likely contributed to the increased light attenuation (Fig. 2i), hindering deep light penetration into the anoxic zone. Whereas in the Northern basin (e.g. station 2), the reduced upwelling of nutrients diminishes epilimnetic primary productivity and increases the water clarity (Verburg et al. 2003; O'Reilly et al. 2003), possibly enabling light penetration to deep-water phototrophic blooms. Thus, we agree with the reviewer that light penetration appears to hinder the formation of the ACM at station 7, even despite the nutrient replete conditions at deeper depths.

In the revised version of the manuscript, we now include the complete light irradiance profiles ($\mu\text{mol photons m}^{-2} \text{s}^{-1}$) in Figure S1 and a description of the findings for stations 2 and 7 on lines 165-181 and 193-196, respectively. Moreover, our overarching summary at the end of the manuscript (lines 399-419) also discusses the factors that control the fate of the ACM at greater length in the revised manuscript.

Cell abundances are much higher than the reported for Pacific OMZ SCM. Considering the high abundances of Chlorobiaceae in the rRNA (up to 22-23% compared with 1.5% for *Synechococcus*) and the lack of *Synechococcus* cells (and data) in upper depths, how reliable are the *Synechococcus* cell counts? Could the distribution of *Synechococcus* be an incorrect match of fluorescent cells? Could be possible to differentiate *Synechococcus* from *Chlorobium* cells by fluorescence microscopy?

We thank the reviewer for raising this point. The fluorescent cell counting by microscopy (shown in Fig. 2a, i) do not specifically discriminate between any particular group of phototrophs (i.e. *Chlorobium* or *Synechococcus*). Instead it represents a reliable quantification of the total number of phototrophs, or pigmented cells for a given depth down to a detection limit of around $10^3 \text{ cells ml}^{-1}$. Unfortunately, it would be difficult to reliably differentiate *Synechococcus* from *Chlorobium* based on the fluorescent cell counting alone. This would require a more targeted molecular approach, such as using CARD-FISH, but even then, the activity of the specific groups could not be known.

Our 16S rRNA genes survey at the ACM identified both *Synechococcales*, reaching up to 1.4% of the microbial community at 150 m depth; and *Chlorobium*, which attained up to 21.6% at a slightly deeper depth of 156 m (Fig. S2b, c). *Chlorobium* is likely supported by the reduced sulfur flux from the hypolimnion (Durisch-Kaiser et al. 2011), coupled to its ability to thrive under extreme low-light conditions (Marschall et al. 2010). Although *Chlorobium* appears to dominate in the ACM, it is possible that we missed the maximum *Synechococcus* peak between 133-150 m. Indeed, previous studies have reported microstratification in two-layer phototrophic systems (Camacho et al. 2001). Thus, anoxygenic photosynthesis by green sulfur bacteria likely prevails in the ACM of Lake Tanganyika, however, we cannot preclude oxygenic photosynthesis by *Synechococcus*.

In the revised manuscript, we have now better emphasized the role of anoxygenic photosynthesis in the ACM, and the possibility of a two-layer phototrophic system: in the abstract (lines 48-49), in the introduction (lines 100-111) and in the results and discussion (lines 169-181).

Authors should also consider the manuscript of Callieri et al. (2019, "The mesopelagic anoxic Black Sea as an unexpected habitat for *Synechococcus* challenges our understanding of global "deep red fluorescence"." ISME J 13(7): 1676-1687). At both stations fluorescence signal increases below the oxycline and could be caused by a non-photosynthetic growth of *Synechococcus*.

The reviewer is correct to highlight that at both stations 2 and 7, the fluorescence chlorophyll signal increases beyond 175 m depth (Fig. 2a, i). In the similarly euxinic Black Sea, Callieri et al. 2019 have ascribed the deep fluorescence in the mesopelagic to the presence of an anoxic population of *Synechococcus* (reaching up to 10^3 cells ml⁻¹ at 750 m), which may grow heterotrophically in the absence of light. Previous molecular surveys in Lake Tanganyika have also recovered 16S rRNA gene sequences affiliated to both *Synechococcus* and *Chlorobium* in the deep anoxic zone (200-1000 m)(De Wever et al. 2005). We thank the reviewer for the publication reference, and have subsequently included the discussion above, in the revised version of the manuscript on lines 203-208.

L154-167: The calculations for the ammonium supply from the bottom layer seems to be based on a manuscript for the injection of nitrate caused by eddies (McGilucuddy 1997) so further explanations of the modelling assumptions and conditions are required.

We agree with the reviewer that the modelled ammonium gradient according to the density profile, described by McGuilucuddy 1997, was not methodically outlined in the original manuscript, leading to a fair degree of confusion regarding its application in this study. After very careful consideration, we believe the predicted ammonium flux, based according to the density profile, was superfluous to the overall crux of the manuscript. The various rate transformation measurements coupled with the $\delta^{13}\text{C}_{\text{DIC}}$ profile, and the other chemical and physical parameters, together provide good evidence that the ACM sustains active carbon and nitrogen cycles, including enhanced ammonium production from organic matter remineralization. Thus, for the sake of brevity, we have replaced the analysis of the predicted ammonium flux, with a more thorough expansion of other aspects of the discussion that we believe provide greater context to the ACM and non-ACM conditions at stations 2 and 7, respectively. These expanded discussion points include a deeper description of the temperature, light, TOC, TON, and cell density profiles, as well as a more thorough discussion of the phototrophic ecology in the ACM (see lines 133-208).

L170-end of the section: The use of labelled lyophilized algal biomass could provide an idea of the organic matter remineralization, but I think it can not be named “in situ rates” (L172). Moreover, it is not so clear to me that the addition of fresh and labile organic matter could produce reliable remineralization rates instead of potential remineralization rates. More details from these incubations are needed and it will be better if the authors include some references with comparison with other procedures. In M&M it is said that the amount of material added were equivalent to approximately 50% of the in situ values but it is not clear what do the authors refer to, dissolved or particulate organic carbon? If those data are available, it would be convenient to show the data to compare the rates. In addition, the enrichment should be specified in terms of carbon pool (DOC or POC) and total amount added.

We thank the reviewer for raising this important discussion. In the revised manuscript, we have made clear on lines 257-267 and 302-313 the degrees of uncertainties in the absolute values associated with the rates of organic matter remineralization. Therefore, we agree with the reviewer that experiments offer insight into the potential for heterotrophic activity across the water column, which is now explicitly indicated throughout the manuscript and namely on lines 214 and 270.

In comparison to other water column estimates, our measured rates of organic nitrogen remineralization, which reached a maximum of 50 nM d^{-1} , seem to be within the range of ammonium regeneration estimates for the water column of other oxygen deficient zones, which range between $21\text{-}333 \text{ nmol N L}^{-1} \text{ d}^{-1}$ (Kalvelage et al. 2015). In the revised manuscript, this comparison is now indicated on lines 302-304. Moreover, we have added additional details pertaining to the algal addition experiments in the Methods (on lines 495-501). It now reads as, “The added lyophilized algal biomass, composed of a mixture of organic containing compounds such as amino acids, DNA nucleotides, acetamide, and creatine (in order of most abundant organic nitrogen sources), is

representative of labile organic matter with a C:N ratio of 3.2. The algal material was prepared fresh prior to each station (80 mg resuspended in 5 ml MQ). The added algal stock represented roughly 50% of the total organic carbon background as verified by TOC/TN analysis (added concentrations ranged from 90-150 μM depending on the sample depth; Fig. S3d, l).

The comparison between the rates of carbon fixation and organic carbon remineralization are largely unbalanced for both epilimnion and OMZ. Carbon remineralization rates are 1-2 orders of magnitude lower than the carbon fixation, even in the OMZ of the station 7. Could the carbon fixation rates be overestimated? Maybe the opposite, could the respiration of organic material be underestimated? Some critical review of these data and discussion are needed.

The reviewer is correct to highlight that the carbon fixation rates are much higher than the carbon remineralization rates (Fig. 2d, e, l, m), and we thank the reviewer for raising this discussion.

It is worth reiterating that in the ACM, we have a tight coupling of both heterotrophic and autotrophic processes, and this inherent complexity is reflected not only the $\delta^{13}\text{C}_{\text{DIC}}$ profile (Fig. 2d), but also by the presence of putative heterotrophic and autotrophic microbes that overlap in the ACM (Fig. S2). Given the inherent complexity in carbon cycling, we suspect that there is some degree of uncertainty in our organic carbon remineralization rates, as the $^{13}\text{C}/^{12}\text{C}$ isotope ratio of DIC may be susceptible to dilution by concurrent processes. For instance, $^{12}\text{C}\text{-HCO}_3^-$ production from ambient organic matter remineralization, as well as the assimilation of remineralized $^{13}\text{C}\text{-HCO}_3^-$ into biomass could contribute to diluting the $^{13}\text{C}/^{12}\text{C}$ ratio. Apart from DIC production, a fraction of the remineralized algal biomass might accumulate in the dissolved organic carbon pool (i.e. acetate), which is missed in this survey, but that could be considered in future analyses. Moreover, different organic matter types and quality could be assessed as this may alter DIC production. These caveats possibly contribute to an underestimation of the DIC production rates, which were much lower relative to the rates of carbon fixation. Nevertheless, the potential rates of organic matter remineralization support that an active heterotrophic community occurs in the ACM at station 2, and at the top of the anoxic zone at station 7. Again, we thank the review for raising this discussion above, and have added these caveats in the revised manuscript on lines 257-267.

Changes of $\delta^{13}\text{C}\text{-DIC}$ are also discussed in this section but no error bars are provided in the data. Considering the small changes measured and discussed, what is the methodological error in the $\delta^{13}\text{C}\text{-DIC}$ determinations? Are the discussed changes significant?

We thank the review for raising this question. The accuracy and precision of the measurements were verified using *Carrara marmor* standard (ETH, Zurich), with reproducibility differences of $\pm 0.05\text{‰}$ (see line 543). This uncertainty, which falls within the range of other published values of $\pm 0.03\text{-}0.07\text{‰}$ (Humphreys et al., 2005; Earth Syst. Sci. Data), is well within the confidence of the measurements and therefore the increases/decreases observed between depths are significant. Moreover, the general heavy $\delta^{13}\text{C}_{\text{DIC}}$ signature associated with the primary and secondary chlorophyll maximum (Fig. 2d), can be further supported by the measured rates of CO_2 fixation in the incubation experiments.

L193: It is not clear to me why the enhanced remineralization and the rapid oxygen decrease is related to the higher penetration of nitrate. Contrary, enhanced nitrification could lead to deeper nitrate penetration but also to higher nitrate concentrations, which it is not obvious in the figure. In addition, high nitrification rates should be also observed in the carbon fixation and $\delta^{13}\text{C}\text{-DIC}$ as it is an autotrophic process. Could the authors make this section more clear?

We agree with the reviewer that this statement was rather ambiguous in the original manuscript, reviewer 3 also echoed a similar point. We agree with the reviewer that nitrification could contribute to deeper nitrate penetration into the anoxic zone at station 2. In the revised version of the manuscript we have subsequently clarified that at station 2, the oxycline was much shallower (~ 120 m) than

compared to station 7 (~140 m; Fig. 2a, i), and the nitracline penetrated slightly deeper into the anoxic zone (Fig. 2b, j). At the bottom of the nitracline, station 2 supported an ACM spanning from 130-170 m depth (Fig. 2a). We posit that as a consequence of the ACM development in the North, enhanced rates of organic matter respiration in the ACM could have contributed to the historic shoaling of the oxycline (Verburg et al. 2003; Verburga and Hecky 2009). Furthermore, the shallowing of the oxycline, in combination with active nitrification in the ACM ostensibly enabled deeper nitrate penetration into the anoxic zone compared to the southern basin (Fig. 2b, j). We have clarified that stations 2 and 7 exhibited contrasting oxycline and nitracline depths on lines 161-163. As written above, we have also provided a more detailed explanation as to why we observe the shoaling of the oxycline and a deeper nitracline at station 2 on lines 410-414.

L196-205: The manuscript assumes that the measurement of carbon fixation in the oxygen deficient zone can be attributed to photosynthesis. However, if other N-related autotrophic processes of anammox and nitrification ($25 + 50 \text{ nM d}^{-1}$) are subtracted from the rates of carbon fixation inside the anoxic zone (ca. 400 nM d^{-1}), the assumed photosynthesis would exceed by a factor of 10 the organic matter remineralization. To maintain anoxic or very low oxygen conditions, net oxygen consumption must exceed the production of oxygen, being not compatible with the data shown.

We thank the reviewer for the discussion and realize that our accounting of key carbon fixing processes was not clearly outlined in the original manuscript, leading to some confusion regarding O_2 production and consumption attributed to carbon fixation rates.

To clarify, in the ACM heterotrophic activity is likely complemented by autotrophic processes that include: oxygenic photosynthesis by *Synechococcus*, anoxygenic photosynthesis by *Chlorobium*, as well as chemolithoautotrophic carbon fixation by anammox and nitrification. In addition, assimilatory carbon by-fixation/anaplerotic carbon fixation by heterotrophs (Erb 2011), has also been observed in aquatic environments (Alonso-Saez et al. 2010). Given that anoxygenic photosynthesis by *Chlorobium* generate reduced sulfur species instead of O_2 , and were abundant in the ACM (Fig. S2), we believe that not all carbon fixation can be attributed to oxygenic photosynthesis. A relevant example of this was highlighted in Lake Cadagno, where oxygenic and anoxygenic photosynthesis each contributed to roughly half of the carbon photoassimilation rate (Camacho et al., 2001), with similar findings reported in another two-layer phototrophic ACM (Brand et al. 2016). The question of how much O_2 is produced in the ACM at station 2, via oxygenic photosynthesis, remains unknown. Although the presence of nitrifiers in the anoxic zone based on the 16S rRNA survey, combined with measured rates of nitrification, suggest that the ACM could sustain some degree of O_2 production, but likely not as much as the reviewer suggested given the abundance of *Chlorobium* (anoxygenic photosynthesis). In the revised version of the manuscript, we have better outlined the key processes of oxygenic and anoxygenic photosynthesis in the introduction (lines 100 to 111) and have clarified the many processes that could be contributing to CO_2 fixation (apart from oxygenic photosynthesis) on lines 252 to 256.

Furthermore, we have more clearly indicated that the algal additions represent potential rates, and that a degree of uncertainty still exists with the absolute values as indicated on lines 257-267. Nevertheless, the algal incubations provide insight into the potential for heterotrophic activity across the water column.

L216: Algal addition experiments were performed in the dark and in theoretical anoxic conditions for the samples of the oxygen deficient zone. Authors claim that high nitrification rates are coupled to the production of ammonium however, what is the oxygen source for the nitrification? If no oxygen is produced in darkness and there is anoxic, no nitrite or nitrate could be produced from ammonium. In addition, no detectable $^{15}\text{NH}_4^+$ was measured in the algal additions without allulthiurea but, what about in the samples with algal additions with allulthiurea?

We thank the reviewer for raising this question. While the experiments are “theoretically anoxic” as the reviewer eluded to above, they are in practice, close to, or near anoxic. Trace amounts of oxygen contamination in these types of experiments are difficult, if not, impossible to avoid even if all necessary precautions are taken, which include carefully filling off samples from the Niskin bottle, degassing exetainer caps (De Brabandere et al. 2012), and purging collected anoxic waters with pure helium (Holtappels et al. 2011). Oxygen contamination often occurs at the last step of the exetainer fill off when samples are transferred from 250 ml glass serum bottles (pre-purged with helium) to the 12 ml exetainers (with degassed caps). Using a highly sensitive trace oxygen sensor we find up to 1-2 μM O_2 in samples collected from the anoxic zone (which originally contained less than 1 μM O_2 , based on the in-situ oxygen sensor; Fig. 2). Thus, while our dark algal addition experiments preclude in situ oxygen production by photosynthesis, nitrification activity is likely sustained by very low micromolar levels of oxygen introduced from sample handling. Indeed, previous studies have also shown that nitrification can occur at nanomolar levels of O_2 (Bristow et al. 2016). In the revised version of the manuscript we have indicated that trace amounts of oxygen could still persist in these algal exetainer experiments in the main text (lines 282-285), and in the methods (lines 533 to 535).

For the same experiments. Labelled N_2 was detected and only attributed to anammox, what about denitrification? If nitrification was tightly coupled to the ammonium release, could the denitrifiers consume a fraction of the labelled nitrite too?

We thank the reviewer for the discussion, we have specifically addressed this comment in more detail below (see $^{30}\text{N}_2$ production in nitrite and ammonium addition experiments). In this specific section of the manuscript (lines 285-287), we have clarified that, “In the algal additions, we also detected labelled N_2 gas production, suggesting that anammox and/or denitrification were a competing sink for remineralized ammonium discussed below (Fig. S3).”

L226-246: Considering the rates and ratio of the DIC and N release from the experiments, how did the authors check if there was a significant chemical release of nutrients? Freezing has been proved to release the intracellular nutrient content of microalgae (ex. Lomstein et al. 1990. Marine Ecology Progress Series 61: 97-105). C:N ratio values of 4 and 3 would correspond to the degradation of a few amino acids, highly enriched in N, but away of the decomposition of the microalgal biomass in general. Although the preferential degradation of N compounds could be also possible, reliable controls should be performed to validate these results.

We thank the reviewer for the discussion concerning the abiotic release of ammonium and DIC from algal incubations. In the study by Lomstein et al., 1990, they froze sediment in liquid nitrogen to extract intracellular pools of ammonium from microalgae. In this study, we did not freeze the algal substrate at any point in the sampling process, and the algal substrate was stored at room temperature and in the dark according to the supplier instructions (Sigma-Aldrich). The algal biomass was prepared fresh before each station in order to minimize any possible degradation of organic material to DIN or DIC. In control experiments, testing the abiotic release of ammonium and DIC from freshly prepared $^{13}\text{C}/^{15}\text{N}$ -algal biomass in sterile synthetic media, we find that the algal substrate yielded negligible production of DIC and ammonium (i.e. $< 1 \text{ nM d}^{-1}$)(now detailed on lines 535-539). Thus, the measured rates from our incubation experiments can be attributed to biotic organic matter remineralization. However, we acknowledge that preferential organic matter remineralization may be exaggerated by the addition of more N-rich algal material (C:N ratio of 3.2) compared to the ambient organic matter pool (Fig. S3g, o). In the revised manuscript, we have indicated this caveat on lines 288-290; and have downplayed the idea of selective remineralization throughout the manuscript.

L253: The lack of N_2 production in nitrate amended experiments do not imply the absence of denitrification but only complete denitrification. Nitrate and nitrite reduction are both denitrification processes so, similar to the marine OMZs, denitrification might provide the N

compounds for anammox. Experiments with labelled and unlabeled nitrate and ammonium were performed so it must be clear if denitrification as nitrite reduction or anammox occurred in the samples, being now not clear in the text or the figures.

We thank the reviewer for raising this interesting discussion. Indeed, the reviewer was correct to point out that the lack of N_2 production in nitrate amended experiments does not imply the absence of denitrification. To be more specific, the lack of $^{30}N_2$ production in the $^{15}NO_3^-$ additions excluded nitrate reduction to N_2 (i.e. complete denitrification). The reviewer was also correct in highlighting that we did not consider nitrite reduction to N_2 by denitrifiers as another possible avenue of N_2 production, we thank the reviewer for raising this point. In the revised manuscript, we investigate the production of $^{30}N_2$ from nitrite and ammonium addition experiments in more detail, which was overlooked in the original manuscript. It now reads as follows on lines 342-353, “At the time of our sampling, at stations 2 and 7, no $^{30}N_2$ production was detected in the $^{15}NO_3^-$ additions, suggesting that complete denitrification to N_2 was absent. Alternatively, denitrification may occur sporadically in Lake Tanganyika due to the sudden inputs of organic matter, as reported for other marine studies (Dalsgaard et al. 2012; Fuchsman et al. 2019b). The lack of $^{30}N_2$ gas production from the $^{15}NO_3^-$ additions, combined with the presence of $^{29}N_2$ production measured in most other incubations ($^{15}NO_3^-$, $^{15}NH_4^+$ and $^{15}NO_2^-$) was instead diagnostic of anammox. Interestingly, however, $^{30}N_2$ production was detected in $^{15}NH_4^+ + ^{14}NO_2^-$ incubations, but only at the top of the anoxic zone where nitrification activity occurred (Fig. 2g, o). In addition, $^{30}N_2$ gas production was reported in $^{15}NO_2^- + ^{14}NH_4^+$ incubations at the bottom of the anoxic zone overlapping with DNRA (Fig. 2h, p). We attribute the $^{30}N_2$ production in the former and latter to a nitrification-anammox and DNRA-anammox coupling, respectively. Such couplings have been previously shown to generate $^{30}N_2$ gas in similar experiments (Kartal et al. 2007; Lam et al. 2007). Thus, we suggest that anammox was the dominant nitrogen loss pathway at the time of sampling, consistent with observations in marine oxygen deficient zones (Kuypers et al. 2005; Kalvelage et al. 2013; Bristow et al. 2017).”

L260: It is confusing the use of “upper oxycline” here, at 130 m depth it seems to be the lower part of the oxycline. Similarly, “the base of the oxycline” is used to describe depths at 147-180 m for the Station 7, being not clear in the figure what the authors refers to.

We agree with the reviewer that this was confusing, not only at this line, but also throughout the manuscript, as we interchangeably used “oxycline”, “oxic-anoxic interface”, “base of oxycline” and “upper oxycline”. To remedy this, we now describe processes as either operating at the “top of the anoxic zone/oxygen deficient zone” using a cutoff of 1 μM to denote the anoxic zone, which was the case for most measured rates at station 7. Or, at station 2, we refer to activity as operating “at the ACM”, or at the “top of the anoxic zone”. We have changed this throughout the manuscript to provide additional clarity. For example, on this particular line (359-360) it now reads as, “...ammonium oxidation to nitrite was detected at the top of the anoxic zone (24.1 nM $N\ d^{-1}$, 130 m depth; Fig. 2h).”

L274 (and throughout the text): what do the authors refer when talk about “nitrification”? Ammonium oxidation is sometimes described independently and then, does nitrification refers to nitrite oxidation? Complete nitrification?

We thank the reviewer for pointing this out, which was not clear in the original manuscript. We generally refer to nitrification as the oxidation of ammonium to nitrate, unless we are specifically referring to our measurements of ammonium oxidation to nitrite. We refrain from calling ammonium oxidation to nitrate “complete nitrification” as this may be misconstrued for comammox (van Kessel et al. 2015). We have specifically clarified this in the section of the manuscript titled “Contribution of organic nitrogen remineralization to nitrogen cycling” where we discuss the various nitrogen transformation process, it now reads as, “Ammonium oxidation to nitrate, which we refer to as nitrification hereafter...” on line 318.

L275: ^{15}N experiments were performed in “anoxic” conditions and in darkness. Considering

the lack of internal oxygen production, could the nitrification rates be produced by oxygen contamination during the incubations?

As mentioned in the earlier comment posed by the reviewer, trace oxygen contamination (1-2 μM), which is largely unavoidable in such incubation exetainer experiments, likely sustained microaerobic nitrification activity, in the absence of oxygenic photosynthesis. In the revised version of the manuscript, we have indicated that trace amounts of oxygen could still persist in these exetainer experiments in the main text (lines 282-285), and in the methods (lines 533 to 535).

L307: This sentence seems to be a bit pretentious. Organic matter remineralization can occur in both oxic and anoxic environments so SCM does not expand organic matter remineralization.

We agree with the reviewer that original phrasing in this sentence was vague and unspecific. Based on the data, the original intent was to convey that the ACM increases local production of organic matter, which in turn sustains an active heterotrophic community, as identified by our algal addition experiments in this zone. The community of heterotrophs at the ACM further enhanced ammonium remineralization fueling greater nitrogen cycling (Fig. 3a); and expanded activity across a broader depth range, compared to the non-ACM scenario where nitrogen cycling activity was mostly constrained to the top of the oxygen deficient zone (Fig. 3b). In the revised manuscript, we have detailed the above statement on lines 390-393.

L310-end of the paragraph: *Synechococcus* can also grow in the dark conditions found below the oxycline in the Black Sea by heterotrophic metabolism (Callieri et al. 2019. ISME J 13(7): 1676-1687), therefore its simple presence does not imply active carbon fixation.

We generally agree with the reviewer that its simple presence does not indicate activity. In line with the reviewer, we have clarified earlier in the manuscript that *Synechococcus* is posited to also grow heterotrophically in the absence of light according to Callieri et al., 2019 (now indicated on lines 178-180). Given that *Synechococcus* is found near the base of the photic zone, we also cannot preclude photoautotrophic CO_2 fixation (lines 165-181). In the revised manuscript, we more broadly hypothesize that photo/litho-autotrophic activities contribute to the bulk rates of CO_2 fixation in the ACM, please see 384-385, and the updated Fig. 3.

L339: Deep chlorophyll maximum is not the same than SCM. Authors could include this manuscript (Marquez-Artavia et al. 2019. Deep-Sea Research Part II-Topical Studies in Oceanography 169: 104686.) describing temporal variability and relationship of the OMZS SCM with density gradients.

We thank the reviewer for the information and the literature reference, indeed, it provides nice context for the ACM in marine oxygen deficient zones. In the revised version of the manuscript, we have briefly mentioned that the ACM in marine oxygen deficient zones can persist year-round and within a stable isopycnal layer (26 kg m^{-3}), according to Marquez-Artavia et al. 2019. We have mentioned this study in the introduction on lines 96-99 and in the last paragraph of the manuscript, which describes the fate and provenance of the ACM in Lake Tanganyika on lines 417-419.

MATERIAL AND METHODS

L376-378: Were the detection limits of the methods theoretical or calculated?

The detection limits were calculated using the regression coefficient and the residual standard deviation of the daily calibration line. We have updated this information in the revised manuscript on lines 446-448.

L385: TOC and TN data are not shown in the manuscript. It would improve the discussion of some aspects of the data.

We thank the reviewer for highlighting this missing aspect. In the revised version, we now show the TOC, TON, and TN profiles, as well as the corresponding C:N_{TOC:TON} ratio in the supplementary Figure S3. In addition, we have added a brief discussion of the results on lines 188-191 and on 196-198, relating to stations 2 and 7 respectively. At station 2, we find that total organic carbon (TOC) and total organic nitrogen (TON) profiles exhibited a relatively noisy vertical structure from 130-150 m depth, with TOC concentrations fluctuating from 85 μM to 109 μM , and TON concentrations ranging from 6 μM to 11 μM (Fig S3d, e). At station 7, the top of anoxic zone, was denoted by a small turbidity signal at 145 m, and was accompanied also by increases in TOC (101-115 μM) and TON concentrations (6-10 μM ; Fig. S3l, m).

L392: The manuscript use for reference details how to measure N-fixation rates, but it is not related to carbon fixation experiments. Provide a valid procedure and reference.

We thank the reviewer for their close attention to detail. Indeed, this reference was for N₂ incubation experiments we apologize for the oversight. We performed these incubation experiments according to Schunck et al., 2013, which was a study that performed bulk measurements of CO₂ fixation in the coastal waters of the Peruvian oxygen minimum zone. We have corrected this reference in the revised version on line 460.

L393: One single control and two replicates is not enough to provide robust determinations of the rates. In addition, what about the dark fixation rates? Could the authors use them to estimate C-fixation by non-photosynthetic processes?

We agree that triplicates would have been better, however, the rates of CO₂ fixation exhibit very little variation between the duplicate samples, as indicated by the small error bars (Fig. 2d, l), which would provide a fair degree of confidence in the measured values. Moreover, the bulk CO₂ fixation rates were further corroborated by the heavy $\delta^{13}\text{C}_{\text{DIC}}$ signature associated with the primary and secondary chlorophyll maximum (Fig. 2d).

We do not have separate measurements for non-photosynthetic carbon fixation (i.e. precluding photosynthesis). The collected waters were incubated in on-deck incubators with shaded foil to mimic the light irradiance at depth. Incubations in the deep anoxic zone were shaded with a black aluminum foil that would enable only residual light penetration, mimicking extreme light conditions at this depth. Therefore, measured rates of CO₂ fixation in the anoxic zone included both non-photosynthetic and photosynthetic activity. We have subsequently clarified this on lines 471-474 of the revised manuscript, and have indicated this in Fig. 3.

L395: Was the water from the oxygen deficient zone de-oxygenated? The entrance of oxygen could enhance non-photosynthetic C-fixing processes (such as nitrification).

The waters collected for carbon fixation in the anoxic zone, at stations 2 and 7, were not purged with helium, and could therefore contain trace amounts of oxygen that sustain enhanced microaerobic nitrification activity, and perhaps higher rates of non-photosynthetic carbon fixation. We have clarified this caveat in the revised manuscript on lines 482-484. Although the much higher rates of carbon fixation at the ACM station 2, compared to the non-ACM station 7, suggest that photosynthetic CO₂ fixation comprises a large fraction of the bulk CO₂ fixation rate in the former. We do not exclude the possibility that oxygen contamination slightly enhanced non-photosynthetic CO₂ fixation overall at both station 2 and 7.

L398-400: What was the final ¹³C concentration or the labelling percent?

The reviewer was correct to highlight that this information was missing in the original manuscript. The final labelling percent in the ¹³C-DIC incubations averaged 3.7%, and we have now added this information in the revised version on lines 468-469.

L401-404: On deck incubations does not seem the best option for deep-water incubations. What was the temperature of the incubation and the in situ temperature of the samples? Temperature profiles should be also shown. In addition, what were the light levels in the incubations? The values are especially relevant for the deepest samples (SCM depths) so they should be shown as the use of high light levels would also produce unrealistic C-fixation rates.

We thank the reviewer for raising this point. In the anoxic collected samples, the temperature differences between the in situ water temperature and the on-deck incubation temperature averaged 4°C, with some extremes reaching up to 8°C. Due to the relatively small mean temperature difference the determined rates of CO₂ fixation were not temperature adjusted. In the revised manuscript, we have indicated this information on lines 478-482.

On-deck incubators were covered with shaded light filters representing the irradiance and light spectrum at the respective sample depth (LEE Filters). Incubations in the deep anoxic zone were shaded with a black aluminum foil that would offer only residual light penetration, mimicking extreme light conditions at this depth. In the revised manuscript, we have indicated this information on lines 471-474.

L415-417: TOC-TN data should be shown. How much algal biomass was added? How was the procedure? Direct addition of the dust, addition of the resuspended material... In addition, some more details of the algal material must be provided: What algae were added? What was the C:N ratio of the material? How many replicates were used? What kind of control experiments were used?

We thank the reviewer for the comment, we agree that these details were not thoroughly outlined in the original manuscript. The ¹³C/¹⁵N-lyophilized algae biomass (98% ¹³C- and ¹⁵N-labelled Sigma-Aldrich) was of *Spirulina platensis*. The added lyophilized algal biomass is composed of a mixture of organic containing compounds such as amino acids, DNA nucleotides, acetamide, and creatine (in order of most abundant organic nitrogen sources) and therefore represents labile organic matter with a C:N ratio of 3.2 (see lines 490-501). Single amendment experiments were performed per depth, which is common for ¹⁵N labelling studies employing a large suite of tracers including ¹⁵N-NO₃⁻; ¹⁵N-NO₂⁻; ¹⁵N-NH₄⁺; ¹³C/¹⁵N- algae, and ¹³C/¹⁵N- algae+ allylthiourea (Bristow et al. 2017; Callbeck et al. 2018). It is worth mentioning that these large suite of addition experiments (exps1-6) are designed to complement each other, providing a degree of redundancy as highlighted by (Holtappels et al. 2011). The slope of the linear regression was used to calculate the rates of N₂ production as a function of time. The standard error was derived from the deviation in the linear slope over the five time points (now indicated on lines 521-522). *t*-tests were applied to ensure rates were significantly different from zero (Thamdrup and Dalsgaard 2002). Sterile control experiments testing the abiotic release of DIC and ammonium from the algal substrate (mentioned earlier) exhibited negligible abiotic release (also indicated on lines 535-539).

L423: Samples were incubated at ambient temperature but what was the in situ temperature? Were rates temperature corrected?

We thank the reviewer for raising this question. In the anoxic collected samples, the temperature differences between the in situ water temperature and the on-deck incubation temperature averaged 4°C, with some extremes reaching up to 8°C. Due to the relatively small mean temperature difference the determined rates of CO₂ fixation were not temperature adjusted. In the revised manuscript, we have indicated this information on lines 478-482.

L425: Incubations were terminated with the addition of Cu(I)Cl, could you provide a

reference for the use of that compound? I found that it is practically insoluble in water, how much did you add to ensure the stop of biological processes?

We thank the review for the question. Indeed, solid Cu(I)Cl is insoluble, and is commonly used to terminate biological reactions in incubation experiments (e.g. Milucka et al. 2015; Oswald et al. 2016, 2017). Copper is generally considered an inhibitor of cellular processes (Macomber and Imlay 2009). We would add solid Cu(I)Cl to a 12 ml extainer (0.15% w/v final concentration), which is now detailed on lines 510-511 of the revised manuscript.

L454: If samples were stained with DAPI, all microorganisms were stained and enumerated; do you have total cell counts? It would help for the discussion of microbial abundance. In addition, how were the *Synechococcus* cells distinguished from the rest of microorganisms (including *Chlorobium*)?

We thank the reviewer for the suggestion to include DAPI cell counts. In the revised manuscript, we now include the vertical profiles of total bacterial cell counts (total DAPI cells minus the phototrophic cell counts) for stations 2 and 7 in the supplementary Fig. S3b, j. At station 2, general bacterial cell densities, although strongly attenuated with depth, did observe a small maximum of 1.1×10^6 cells ml⁻¹ at 135 m near the ACM (Fig. S3b), which coincided with the turbidity signal at 135 m (Fig. 2c). At station 7, bacterial cell densities also observed a small maximum near the turbidity peak (145 m) attaining up to 5.7×10^5 cell ml⁻¹ (Figs. 2k, S3k), although the median cell densities were nearly 2-fold lower than compared to station 2. In the revised manuscript, we now included this information above for stations 2 and 7 on lines 183-186 and 198-200, respectively.

As mentioned in our earlier response, epifluorescence microscopy alone cannot specifically distinguish *Synechococcus* from *Chlorobium*, as their morphologies are similar, hence the cell counts are presented as total fluorescent cells in Figure 2. Instead we rely on the 16S rRNA survey (Fig. S2) to provide some insight into the identity of the phototrophic members in the ACM.

L458-471: Microbial community was characterized by metagenomics but this information is barely used in the manuscript. In addition, results shown in Fig. S1 are also quite scarce. Why do not show the relative abundances of the most relevant functional groups? Which groups were included as “nitrifiers”? I think it would be better to separate in ammonium oxidizing bacteria and archaea, nitrite oxidizing bacteria, denitrifiers, etc...

We thank the reviewer for raising this discussion. In the revised version of the manuscript, we have now elaborated in greater detail the 16S rRNA microbial community and some of the key players involved:

- Apart from identifying key phototrophs (*Synechococcus* and *Chlorobium*), we have provided additional information regarding the putative heterotrophs in the ACM, that appeared to be overrepresented in the 16S rRNA gene survey. Some members include *Bacteroidetes*, *Firmicutes*, *Ignavibacteria*, and *Hydrogenophilaceae* (Fig. S2d, j). This information is now discussed on lines 243-247.
- With respect to nitrogen cycling, we have better distinguished the key ammonium- and nitrite-oxidizing prokaryotes on lines 325-329, and also in Fig. S2e, k. For instance, our 16S rRNA gene survey identified ammonium oxidizing *Bacteria* and *Archaea* within *Nitrosomonadaceae* and *Nitrosopumilaceae* (*Thaumarchaeota*), respectively. Members of the *Thaumarchaeota* largely dominated in the anoxic zone, in line with a previous lipid biomarker study in Lake Tanganyika (Schouten et al. 2012). Ammonium-oxidizing bacteria also co-occurred with nitrite-oxidizing bacteria, *Nitrospirae* (Fig. S2).
- We have also further elaborated on the 16S rRNA diversity of *Planctomycetes* on lines 334-339.

Our aim in this study was to comprehensively resolve carbon and nitrogen cycling biogeochemistry under an ACM and non-ACM scenario in Lake Tanganyika. We plan to further

explore the relevant functional pathways and key organisms and their ecophysiology in far greater detail in a more metagenomics-based manuscript in the future. We feel that adding this information here, although interesting, would go beyond the scope of the current manuscript, but we appreciate the reviewer raising this discussion.

Reviewer #2 (Remarks to the Author):

The authors present a study of nitrogen cycling in Lake Tanganyika, i.e. biological N-cycling rates, and physical N-fluxes. The northern basin has an SCM at about 150 m, while the southern basin does not. They argue that the SCM is a “hotspot” of N-cycling and fixed N-loss. I think the article and subject matter are suitable for publication in Nature Comm.; this is the one of the few examples of the importance of the SCM in lakes. However I have numerous questions and comments I would like to see addressed beforehand. So, in no particular order of importance, are my other questions and comments.

We thank the reviewer for their constructive feedback and overall support. We feel their comments greatly contributed to improving the manuscript, as outlined below.

1) I think the title is misleading. When I see “oxygen minimum zone” I think about the global ocean, not Lake Tanganyika. I think the lake name needs to be in the title.

We agree with the reviewer that the title was vague in the original manuscript. We have subsequently reworded the title to more explicitly describe the key finding and the specific study site. In the revised manuscript it now reads as, “Anoxic chlorophyll maximum accelerates organic matter remineralization and nitrogen loss in Lake Tanganyika”.

2) Line 46. “Stable isotope” makes me think of natural abundance. I would say “Stable isotope tracer experiments”.

We thank the reviewer for highlighting this, and we therefore have made this change throughout the manuscript, beginning from **line 47** onward.

3) Introduction first paragraph. They use “oxygen minimum zone” and “oxygen deficient zone” apparently interchangeably. To many oceanographers they are different. All O₂ profiles have a minimum concentration of oxygen, whereas oxygen deficient means not enough oxygen for aerobic respiration to dominate. Define term(s) and stick to them. Also, not all eutrophic upwelling zones have “vast OMZs, and finally the three large denitrifying ODZs extend well off shore 500-1000km.

We agree with the reviewer that we were not consistent with our use of “oxygen minimum zone” and “oxygen deficient zone” in the introduction and throughout the manuscript. In the revised version, we now apply the term “oxygen deficient zone or anoxic zone” throughout the text and figures. Of course, the literature can vary with respect to its definition of the oxygen deficient zone with previous studies using an O₂ cutoff ranging between 1-20 μM. In this study, we define the oxygen deficient zone at a more stringent cutoff of 1 μM, which is roughly where O₂ concentrations ceased to decrease in our O₂ profiles presented in Figure 2a, i. O₂ concentrations of less than 1-2 μM could be generally viewed as, anoxic, or near anoxic, as now clearly defined **on line 73** of the revised manuscript.

In addition, we have reworded the sentence referring to “vast OMZs” to read as follows (**on lines 75-77**), “Marine eutrophic upwelling regions, where large oxygen deficient zones can form, contribute to nearly one-fifth of global N₂ production at 81 Tg N per year, and freshwater lakes contribute up to 31 Tg N per year (Seitzinger et al. 2006).”

4) Line 93. Pronoun reference is unclear. Do they mean *Synechococcus* or anammox? Both of these require nitrogen assimilation and carbon fixation.

We agree with the reviewer that pronoun was unclear. The paragraph has since been revised and references to anammox have been disentangled from our introduction of phototrophic communities, according to changes suggested by reviewer 1. Anammox and nitrogen cycling activities are introduced on lines 83-89, while phototrophic communities are described separately on lines 95-111.

5) Line 98. Define “oxic-anoxic transition”. Is it the actual narrow zone where the transition happens or is it larger? Where does it start and end? Fuchsman et al show the SMC is in the functionally anoxic zone. Looking at their profiles in figure 2, I would say their SMC is in the functionally anoxic zone also. It is clearly where the oxygen profile quits going down. If that's not functionality anoxic then the transition zone appears to go to at least the bottom of their plots.

We agree with the reviewer that this was confusing, not only at this line, but also throughout the manuscript, as we interchangeably used “oxycline”, “oxic-anoxic interface”, “base of oxycline” and “upper oxyline”. To remedy this, we now describe processes as either operating at the “top of the anoxic zone/oxygen deficient zone” using a cutoff of 1 μM to denote the anoxic zone. We have changed this throughout the manuscript to provide additional clarity. For instance, on this particular line on 86 it now reads as, “...active nitrogen turnover of N_2 is generally greatest near the top of the anoxic zone.”

6) Paragraph starting on line 99, first sentence. I don't get where they are going with “heterotrophs”. It seems clear from the literature that photosynthesis is important in the SCM (their references 9 & 10). Are they trying to say that chemoautotrophic production by anammox is a significant OM source? If they argue, I would argue that it's a very minor one because much more nitrogen is consumed than nitrogen assimilated for growth.

We agree with the reviewer that this sentence was a little ambiguous in the original manuscript. We were not trying to suggest that anammox is a significant organic matter source, here we agree with the reviewer that its contribution would only be minor. The original idea was to convey that local organic matter production by phototrophs in the ACM, could enhance or stimulate heterotrophic activity and hence accelerate organic matter remineralization to ammonium that, in turn, could influence the nitrogen cycle.

We have reworded the sentence for clarity it now reads as follows (on lines 112-114), “We hypothesize that the production of labile organic matter by phototrophs in the ACM, coupled to its remineralization to ammonium by local heterotrophic bacteria, might have an important influence on nitrogen cycling in the oxygen deficient zone.”

7) Lines 101-02. “freshwater world”? Just say “largest freshwater oxygen deficient zone in the world”?

We agree with the reviewer, this was now changed on line 115.

8) Line 103. What is a “community of nitrogen cycling processes”? Just say “... supports an active nitrogen cycling community”.

We agree with the reviewer, this was now changed on lines 116.

9) Line 104. North-south gradient. I don't see so much of a gradient. I would call it a contrast between presence and absence.

We agree with the reviewer, and have therefore changed it on lines 118-120 to read as, “Moreover, the presence of an ACM in the Northern basin and its absence in the Southern basin (Fig. 1a),

provides an ideal framework to evaluate the impact of deep phototrophic communities on nitrogen cycling activities in the oxygen deficient zone.”

10) Line 108. Again, insert “tracer” before the word “experiments”.

We agree with the reviewer and have made this change in the revised manuscript.

11) Results and discussion section. Here they start presenting N-transformation rates. They have anammox rates from the $^{15}\text{NH}_4$ tracer experiments. Presumably they measured $^{29}\text{N}_2$. Did they measure $^{30}\text{N}_2$? If they only measured $^{29}\text{N}_2$ they can't separate anammox from denitrification. Did they measure denitrification?

We thank the reviewer for raising this point, and agree that this was not well described in the original version of the manuscript. At stations 2 and 7, no $^{30}\text{N}_2$ production was detected in the $^{15}\text{NO}_3^-$ additions, suggesting that complete denitrification to N_2 was absent, or occurs sporadically in Lake Tanganyika due to the sudden inputs of organic matter, as reported for other marine studies (Dalsgaard et al. 2012; Fuchsman et al. 2019b). The lack of $^{30}\text{N}_2$ gas production from the $^{15}\text{NO}_3^-$ additions, combined with the presence of $^{29}\text{N}_2$ production measured in most other incubations ($^{15}\text{NO}_3^-$, $^{15}\text{NH}_4^+$ and $^{15}\text{NO}_2^-$) was instead diagnostic of anammox, consistent with observations in marine oxygen deficient zones (Kuypers et al. 2005; Hamersley et al. 2007; Bristow et al. 2017; Callbeck et al. 2017). Interestingly, however, $^{30}\text{N}_2$ production was detected in $^{15}\text{NH}_4^+ + ^{14}\text{NO}_2^-$ incubations, but only at the top of the anoxic zone where nitrification activity occurred (Fig. 2g). In addition, $^{30}\text{N}_2$ gas production was reported in $^{15}\text{NO}_2^- + ^{14}\text{NH}_4^+$ incubations at the bottom of the anoxic zone overlapping with DNRA (Fig. 2h). We attribute the $^{30}\text{N}_2$ production in the former and latter to an nitrification-anammox and DNRA-anammox coupling, respectively. Such couplings have been previously shown to generate $^{30}\text{N}_2$ gas in similar experiments (Kartal et al. 2007; Lam et al. 2007). Thus, we suggest that anammox was the dominant nitrogen loss pathway at the time of sampling. This description is now included on lines 341-353 of the revised manuscript.

12) Lines 130-31. Forget the biogeochemistry and physics (which covers all four basic sciences) and just say below the primary chl. Max we observed distinct profile differences. We agree with the reviewer, this was now been changed on line 153 of the revised manuscript.

13) Line 134. Turbidity peak. This is an interesting observation but I don't think it is discussed later.

We agree with the reviewer that this turbidity peak and its association with an enhanced density gradient just above the oxygen deficient zone is interesting. In the revised manuscript, we briefly iterate that this feature (Fig. 2d; peak A) could have contributed to the isolation of the deep phototrophic community at the base of photic zone in the ACM (on lines 408-410). Moreover, the enhanced turbidity maximum above the oxygen deficient zone is also depicted in Fig. 3a.

14) Line 160. I don't understand the Predicted NH_4 . Their reference 25 is about the Sargasso sea, a totally different system. McGillicuddy & Robinson try to calculate the upward flux of NO_3 supporting Primary Production in the S.S. This is a relatively straightforward system. Here they are in a complicated N-Cycling system with all the processes taking place in the hotspot zone they are trying to model. Give me some more specifics about how the prediction was done. It is not clear to me from looking at the reference.

We agree with the reviewer that the modelled ammonium gradient according to the density profile, described by McGillicuddy 1997, was not methodically outlined in the original manuscript, leading to a fair degree of confusion regarding its application in this study. After very careful consideration, we believe the predicted ammonium flux, based according to the density profile, was superfluous to the overall crux of the manuscript. The various rate transformation measurements coupled with the $\delta^{13}\text{C}_{\text{DIC}}$ profile, and the other chemical and physical parameters, together provide good evidence that the SCM sustains an active nitrogen cycle and enhanced ammonium production from organic matter

rem mineralization. Thus, for the sake of brevity, we have replaced the analysis of the predicted ammonium flux, with a more thorough expansion of other aspects of the discussion that we believe provide greater context to the ACM and non-ACM conditions at stations 2 and 7, respectively. These expanded discussion points include a deeper description of the temperature, light, TOC, TON, and cell density profiles, as well as a more thorough discussion of the phototrophic ecology in the ACM (see lines 133-208).

15) Sentence beginning on line 162. Basically this sentence says that the missing ammonium is due to processes that make it missing. This is obviously true but doesn't really say anything.

As mentioned above, the predicted ammonium flux was superfluous to the overall crux of the manuscript, and for the sake of brevity we have removed this analysis and have instead expanded on the various other aspects of the discussion that we believe add greater context to the ACM and non-ACM at stations 2 and 7, respectively (see lines 133-208).

16) Line 165, "gradients exceeded concentrations". This is comparing apples and oranges. Which do they mean, gradients and gradients or concentrations and concentrations? Also, the gradient (curvature) in the ammonium profiles in figure 2a indicates there must be consumption in that zone or increasing production with depth.

As mentioned above, this paragraph has since been changed to the expand on other aspects including temperature, light, TOC, TON, and cell densities profiles at stations 2 and 7 (see lines 133-208).

17) Line 210. Here they talk about allylthiourea inhibition. I'm not so familiar with lakes, but in the oceans nearly all nitrifiers are archaea. Archaeal nitrifiers are not very sensitive to ATU (Martins-Habbena et al 2014, Env. Micro.) In the text they say they detected bacterial and archaeal nitrifiers. What is shown in Suppl. Fig. 2? If there are archaeal nitrifiers and no nitrification was detected with added ATU does that mean they were unimportant?

We thank the reviewer for highlighting this study regarding ATU inhibition on nitrifiers. We have better distinguished the key ammonium- and nitrite-oxidizing prokaryotes on lines 325-329, and also see Fig. S2e, k. Briefly, our 16S rRNA gene survey identified ammonium oxidizing *Bacteria* and *Archaea* within *Nitrosomonadaceae* and *Nitrosopumilaceae* (*Thaumarchaeota*), respectively. Members of the *Thaumarchaeota* largely dominated in the anoxic zone, in line with a previous lipid biomarker study in Lake Tanganyika (Schouten et al. 2012). Ammonium-oxidizing bacteria also co-occurred with nitrite-oxidizing bacteria, *Nitrospirae* (Fig. S2).

With respect to the concentrations of ATU and its inhibition of nitrifiers. The main identified ammonium-oxidizing archaea was *Candidatus Nitrosotenuis* within *Nitrosopumilaceae*. It is a fairly distant relative to *Nitrosopumilus maritimus*, which was used by Martins-Habbena et al 2014 to test ATU inhibition. Although we are dealing with two different taxa, the findings of Martins-Habbena et al., 2014 show partial inhibition (~50-85 %) at 100 μ M ATU. They also find that ATU strongly inhibited ammonium oxidizing bacteria at even lower concentrations. Because we added ATU to a final concentration of 84 μ M, we suspect that this was sufficient enough to induce, at least, partial inhibition, enabling ammonium to accumulate in the ATU + 15 N-algal addition experiments. Moreover, the lack of ammonium production in the 15 N-algal addition only, would further support the idea that ATU induced some degree of inhibition, otherwise the ammonium production rates would have been similar between the two experiments with and without ATU.

In the Methods on lines 491-495, we have now briefly commented on this study stating, "The concentration of added nitrate, nitrite, ammonium and allylthiourea were 20 μ M, 10 μ M, 10 μ M and 84 μ M, respectively. At concentrations of 100 μ M, Martens-Habbena et al. 2015 have shown that

allylthiourea is a partial inhibitor of ammonium oxidizing archaea, although its effectiveness on the freshwater taxa identified here (Fig. S2e, k) remains untested. Martens-Habbenha et al. 2015 also find that allylthiourea is a strong inhibitor of ammonium oxidizing bacteria.”

18) Paragraph beginning on line 226. In the SCM OrgC recycling is about 25nM/d and org N recycling is 2 times that. Why is this? They suggest some organism is preferentially recycling NH₄ what organism is doing this without recycling much C at all. It's not the NO₃ reducers because they are after the C. and it's not aerobic organisms because this is the ODZ. They talk about the changing C:N ratio in the SCM from 7.7:1 to 3.1:1 deeper in the water column. However, the observed regeneration ratio is ~0.5:1. At the end of the paragraph they talk about preferential degradation of amino acids, but there are no amino acids with C:N ratios of 0:5. Have I missed something here?

We thank the reviewer for raising this discussion concerning selective remineralization of organic nitrogen. Ammonium, is liberated from organic matter degradation by heterotrophic bacteria, as a by-product of metabolizing organic nitrogen sources (Capone et al. 2008). For example, some bacteria employ extracellular enzymes and peptidases/deaminases which cleave off amine groups generating ammonium, which is either excreted as waste or is assimilated into biomass. In the oxygen deficient zone, a community of heterotrophic bacteria are likely responsible for the sequential breakdown of algal biomass to CO₂ and NH₄⁺. Putative heterotrophs, that appeared to be overrepresented in the 16S rRNA gene survey included *Bacteroidetes*, *Firmicutes*, *Ignavibacteria*, and *Hydrogenophilaceae* (Fig. S2d). In a marine sediment study using ¹³C-labelled algal biomass, Müller et al. 2018 find that *Bacteroidetes* and *Firmicutes* initially hydrolyze/ferment the algal tracer to smaller molecular weight compounds, such as acetate, which is then degraded by sulfate- and nitrate-reducing bacteria. In anammox reactors, *Chlorobium* and *Ignavibacteria* have been shown to employ extracellular peptidases to metabolize proteins and organic matter, with indications that these taxa are tightly coupled with anammox(Lawson et al. 2017). The aforementioned taxa, may, or may not be directly involved in dissimilatory nitrogen transformation processes, and could grow via fermentation, as suggested by (Müller et al. 2018). In the revised manuscript, we have added this information above on lines 244-252 and 271-273.

Indeed, the measured DIC and NH₄⁺ regeneration ratio in our algal experiments was ~0.5:1 in the ACM, albeit, this ratio is likely exaggerated by the addition of N-rich algal material. Nevertheless, previous studies have reported selective remineralization of nitrogen-rich organic matter, such as amino acids, coupled to denitrification in in-situ experiments(Van Mooy et al. 2002). In globally compiled marine datasets, selective remineralization of nitrogen over carbon was shown to increase with depth (from the thermocline down to intermediate waters), with models also reproducing this pattern of preferential heterotrophic remineralization(Letscher and Moore 2015; Zakem and Levine 2019). In the revised manuscript we have now briefly noted this observation on lines 287-295, but have downplayed the idea of selective remineralization throughout the manuscript. In addition, we realized that the TOC:TN ratio, was not an ideal indicator of preferential remineralization in the original manuscript. In fact, the C:N ratio (either of TOC:TN or TOC:TON) can be convoluted by various processes, such as carbon and nitrogen fixation, which could mask selective remineralization, and therefore we have also downplayed this aspect as well.

19) Line 236. Here and beyond they talk about C:N ratios that are never shown anywhere. Put a figure in the supplemental.

As mentioned above we place less weight on the C:N ratio in the revised manuscript, as the reviewers have correctly pointed out that it could be convoluted by various processes namely, carbon fixation, nitrogen fixation, organic carbon and nitrogen remineralization etc. Nevertheless, we have still included the TOC, TON, and TN profiles, as well as the corresponding C:N_{TOC:TN} and C:N_{TOC:TON} ratio in the supplementary Figure S3.

20) Line 253. Now you tell me about the $^{30}\text{N}_2$ results. Put this earlier.

We thank the reviewer for highlighting this, we have hopefully remedied this in the revised version of the manuscript. On lines 285-287 it now reads as, "In the same algal additions, we also detected labelled N_2 gas production, suggesting that anammox and/or denitrification were a competing sink for remineralized ammonium discussed below (Fig. S3)." Later on, in the discussion (on lines 332-354) we further elaborate on the $^{29}\text{N}_2$ and $^{30}\text{N}_2$ production from the incubation experiments, and the main nitrogen loss pathways anammox and denitrification in Lake Tanganyika.

21) Line 305. This section is all about the importance of the SMC for enhancing nitrogen cycling and loss in the ecosystem. They measure all sorts of N-cycling rates and fluxes and then derive their conclusions. All of their rates except C-fixation are in nM/d. But the one thing they neglect to point out is that their measured C-fixation in the SCM is about 0.5 $\mu\text{M}/\text{d}$ which is 500 nM/d. Given a more or less co-occurring oxygen production of 500 nM/d this about 10 times bigger than their N-cycling rates. This is a lot of oxidation potential and it should have a big effect on the N cycling rates in the SCM. Oxygenic reactions in the SCM are only briefly discussed.

We thank the reviewer for the question, which was also echoed by reviewer 1. We realize that our accounting of key carbon fixing processes was not clearly outlined in the original manuscript. To clarify, in the ACM, autotrophic processes ostensibly include: oxygenic photosynthesis by *Synechococcus*, anoxygenic photosynthesis by *Chlorobium*, as well as chemolithoautotrophic carbon fixation by anammox and nitrification. In addition, assimilatory carbon by-fixation/anaplerotic carbon fixation by heterotrophs (Erb 2011), has also been observed in aquatic environments (Alonso-Saez et al. 2010). Given that anoxygenic photosynthesis by *Chlorobium* generate reduced sulfur species instead of O_2 , and were abundant in the ACM (Fig. S2), we believe that not all carbon fixation can be attributed to oxygenic photosynthesis. In the ACM of Lake Cadagno, oxygenic and anoxygenic photosynthesis each contributed to roughly half of the carbon photoassimilation rate (Camacho et al., 2001), with similar findings reported in another two-layer phototrophic ACM (Brand et al. 2016). The question of how much O_2 is produced in the ACM at station 2, via oxygenic photosynthesis, remains unknown. Although the presence of nitrifiers in anoxic zone based on the 16S rRNA survey, combined with measured rates of nitrification, suggest that the ACM could sustain some degree of O_2 production, but likely not as much as the reviewer suggested given the abundance of *Chlorobium* (anoxygenic photosynthesis).

In the revised version of the manuscript, we have better outlined the key processes of oxygenic and anoxygenic photosynthesis in the introduction (lines 100 to 111) and have clarified the many processes that could be contributing to CO_2 fixation (apart from oxygenic photosynthesis) on lines 252 to 256 and in Figure 3. Finally, on lines 393-395 we state, "if oxygenic photosynthesis is active, in situ oxygen production by cyanobacteria like *Synechococcus* may support microaerobic nitrification and organic matter remineralization in these anoxic waters as proposed elsewhere (Stolper et al. 2010; De Brabandere et al. 2012; Kalvelage et al. 2015; Brand et al. 2016; Garcia-Robledo et al. 2017)."

22) Paragraph beginning on line 335. The climate models mentioned here that predict expanding OMZ are talking about the large marine OMZs not lake OMZs. The ocean OMZs are quite different from the Lake Tanganyika. First heterotrophic denitrification is an important process. Secondly, there is not a large NH_4 flux up from the bottom and third, The SCM only exists at the top of a very much thicker OMZ.

We thank the reviewer for the comment and have subsequently dropped the mention of climate models related to marine oxygen deficient zones, and instead, have placed more focus on climate change in Lake Tanganyika. This includes a discussion of increasing lake thermal stratification and declining wind speeds, and its effect on the fate and provenance of the ACM in Lake Tanganyika (see the revised paragraph on lines 399-423).

In the revised version of the manuscript, we are now more careful to distinguish between, stratified freshwater lakes, basins and marine oxygen deficient zones. We agree that these systems are different, but also share some similarities as reviewer 3 had also eluded to in their opening remarks. For instance, the water column of coastal upwelling regions often observe large ammonium fluxes, the presence of dissolved sulfide, and in some cases reports of an ACM, along with active nitrogen cycling related to anammox, denitrification, nitrate reduction to nitrite and nitrification (Naqvi et al. 2000; Lavik et al. 2009; Ulloa et al. 2012; Schunck et al. 2013; Galán et al. 2014; Callbeck et al. 2018), in a similar metabolic model (Kalvelage et al. 2013), as presented here in Figure 3 (albeit without the detailed analysis of the ACM on nitrogen cycling). Lake Tanganyika, also exhibits very similar chemical zonation to other well studied euxinic basins, such as the Black Sea, Cariaco basin, and the Saanich inlet, which have also been compared to coastal marine oxygen deficient zones (Lam et al. 2009; Ulloa et al. 2012), and are akin to conditions that would be found in the ancient euxinic ocean (Johnston et al. 2009; Ulloa et al. 2012). Thus, we generally agree with the reviewer, and have been careful to make these distinctions clear in the revised manuscript, while still highlighting the potential to transfer knowledge between these anoxic environments.

23) Line 416. Delete “was later”

We agree with the reviewer, and this is now correct in the revised manuscript.

Figure 2.

1) Is the apparent chlorophyll in the deep waters real and why does it seem to increase in the ODZ at station 7?

The reviewer is correct to highlight that at both stations 2 and 7, the fluorescence chlorophyll signal increases beyond 175 m depth (Fig. 2a, i). In the similarly euxinic Black Sea, Callieri et al. 2019 have ascribed the deep fluorescence in the mesopelagic to the presence of an anoxic population of *Synechococcus* (reaching up to 10^3 cells ml^{-1} at 750 m), which may grow heterotrophically in the absence of light. Previous molecular surveys in Lake Tanganyika have also recovered 16S rRNA gene sequences affiliated to both *Synechococcus* and *Chlorobium* in the deep anoxic zone (200-1000 m)(De Wever et al. 2005). We thank the reviewer for raising this question, which was also echoed by reviewer 1, and have subsequently included a brief discussion of this in the revised version of the manuscript on lines 203-208.

2) Also the dotted red line mentioned in the caption is actually the oxygen profile in the figure. There is a dashed horizontal black line in some of the panels which likely the oxygen minimum zone. If so it would be helpful if this was in all the panels so we compare them side by side e.g. it is not clear to me where the OMZ starts in panels a and i. At the right of the panels they show the processes, they show "oxygen deficient zone". The ODZ appears to extend only a little deeper than 200 m. Is the water below that not oxygen deficient. Is the bracketed ODZ the same as the SCM layer in figure 3 even though there is no chlorophyll in the lower ODZ

We thank the reviewer for highlighting this, we have changed the dotted red line in the O_2 profile to a solid red line, in all vertical profile figures. We have extended the dashed horizontal black line across all side-by-side panels. The bracketed ODZ line was perhaps a little misleading, in fact, the oxygen deficient zone starts at the top of the dotted line and precedes downward across the entire water column. We have clarified this in the revised version, please see Figs. 2, S2 and S3.

Supplemental figure 2.

In the supplemental figure how do you get a local increase in % surface illumination at about 150 m. At station 2 the light level at about 120 m is only about 0.1%, but it is almost 0.2% at

150 m. Where does the added light come from?

We agree with the reviewer that the small secondary increase in the irradiance value associated with the ACM at 145 m (mainly observed with the April 26 profile at station 2; see new Fig. S1a), is rather bizarre. Our current explanation is this could be attributed to changes in the surface irradiance (i.e. changes in sun angle, clouds etc (see new Fig. S1b), and/or possibly due to minor reflectance induced by particles and phototrophic cells at this depth (Fig. 2a, c). We have appended this brief explanation to the S1 figure caption.

Reviewer #3 (Remarks to the Author):

This manuscript by Callbeck and colleagues reported their study on how the establishment of the secondary chlorophyll maximum (SCM) could accelerate nitrogen-cycling processes and nitrogen loss in the oxygen deficient zone of Lake Tanganyika. They conducted an impressive, comprehensive suite of incubation experiments with ^{15}N -labeled tracers to measure rates of nitrification, anammox, nitrate reduction, DNRA and remineralisation of algal organic matter ($^{13}\text{C}/^{15}\text{N}$ -dual-labeled) into CO_2 and NH_4^+ , as well as that of CO_2 fixation via ^{13}C -DIC-incubations, further complemented with DIN, $\delta^{13}\text{C}$ -DIC and CTD profiling. They examined these processes at two different stations, one in the north basin (Station 2) characterised by the presence of an SCM within the oxygen minimum zone, and one in the south (Station 7) without SCM. Their results have clearly demonstrated that the presence of SCM has enhanced organic N remineralisation and N-loss from the system. As Lake Tanganyika is the second-largest freshwater lake on Earth and the largest anoxic freshwater system, these findings are important for the N-budget of freshwater systems that have generally been understudied in global N-cycling. Moreover, due to its analogy with marine oxygen deficient zones and that SCM is increasingly found to be relatively common therein, findings from this study could lend important insights into how marine N-budgets may change in future ocean should SCM become more widespread.

Nevertheless, while the main conclusions drawn are sound, the developing arguments in the Discussion and data interpretation are sometimes confusing and illogical, with tendency of overinterpretation over certain minute details, and the modeling approach, flux calculations and molecular analyses lack details.

We thank the reviewer for their overall support of the work and appreciate their constructive feedback. We find that their comments stimulated a more thoughtful discussion, which we believe has contributed to a more refined/detailed interpretation of the results in the revised manuscript as outlined below.

First of all, the assertion that stratification was enhanced at Station 2 is not fully evidenced in the presented data, and neither is the causality of stratification to SCM development. The temperature gradient (dT/dz) shown in Fig 1 is indeed a good proxy for density gradient across the transect, but the highest values, hence strongest gradients, were actually found in the upper epilimnion in the South basin where Station 7 was located, and this also coincided

with higher chl a. At least from the colour scale used, it is unclear that Station 2 experienced enhanced stratification, as the values are similar between the two stations at and below the oxygen deficient zone. The Brunt-Väisälä frequency (N^2) in Fig 2 also indicated similar values throughout the water column at the two stations, except perhaps a thicker layer of enhanced stability (high N^2) and a minor secondary maximum ~105m at Station 2 (though the max N^2 was actually higher at Station 7). While these might be better indication of more stable/ stratified waters at Station 2, the values at/below oxygen deficient zone were indistinguishable across the stations. The authors overlaid the stratification signals (lines 130-140) in my opinion. In addition, Station 7 was labelled with enhanced upwelling in Fig 1 (and in text), but there was no supporting evidence shown. In general, there is also a tendency of overinterpretation of depth profiles, including both physical and biogeochemical data (e.g. lines 119-165, 187-205).

We thank the reviewer for raising this discussion concerning the claims of “enhanced stratification at station 2” and “enhanced upwelling at station 7”. We would first like to layout the arguments supporting enhanced upwelling at station 7, especially in light of the temperature profiles which were now added in the revised manuscript (please see the revised Fig. 1b and Fig. 2c, k). At stations 2 and 7, the epilimnion was characterized by a warm (27-28 °C) and relatively well mixed water mass from 0-30 m, with the thermocline occurring from 30-75 m (Figs. 1b and 2c, k). However, station 7, and the southern basin in general, exhibited a steeper thermal gradient throughout the metalimnion than compared to station 2 (Figs. 1c, 2c, k). The steep thermocline at station 7, also coincided with a peak in N^2 , otherwise known as the Brunt-Väisälä frequency (Fig. 2c, k), which also reflects changes in the temperature gradient. Previous studies have established that the onset of the south-east trade winds in May (Verburga and Hecky 2009), forces the upward tilting of the thermocline, resulting in a shallow and steep gradient to develop in the southern basin (Ehrenfels et al. 2020). Moreover, higher *in situ* chlorophyll fluorescence was reported at station 7, which is consistent with previous reports of higher overall primary productivity in the southern basin compared to the northern basin as a result of the nutrient upwelling (Bergamino et al. 2010). Thus, the tilting of the thermocline, and the higher epilimnetic primary productivity, support enhanced upwelling at station 7, in line with previous studies (Verburga and Hecky 2009). In the revised manuscript, we have better clarified these points discussed above on lines 134-148.

We agree with the reviewer that the use of “enhanced stratification at station 2” was misleading given that the entire basin is thermally stratified (Fig. 1b). Indeed, the reviewer was correct to point out that the maximum N^2 signal in the epilimnion was higher at station 7 than compared to station 2 (Fig. 2c, k), which is due to the overall steeper thermocline at station 7. What we failed to emphasize in the original manuscript was that station 2 exhibited a deep stratified layer. For instance, we find evidence of a small N^2 signal from 95-120 m (Fig. 2c; Peak A). Whereas, at station 7, the N^2 stability observed only minor noise from 65-140 m depth (Fig. 2k). Moreover, station 2 was characterized by a relatively noisy vertical turbidity signal with moderately sized peaks occurring at 50, 68, 87 and 105 m depth (Fig. 2c). In contrast, at station 7, we find a much smoother turbidity profile over the same depth range (Fig. 2k). Interestingly, the turbidity peak at 105 m, at station 2, coincided with a small increase in the N^2 stability peak at 95-120 m (Fig. 2c; Peak A), that indicated some degree particle aggregation occurring at a deep layer, before the oxygen deficient zone. To reiterate, we agree with the reviewer that the use of “enhanced stratification at station 2” was misleading, and have since downplayed this terminology throughout the manuscript, and have reworded our description of the turbidity and N^2 vertical profiles on lines 154-161.

With respect to the causality of the ACM, we have placed less focus on stratification as the main driver of ACM formation, and instead emphasize the factors that control the fate of epilimnetic primary productivity (PP), which indirectly regulates light penetration to deeper depths. Lake Tanganyika studies specifically highlight two possible controlling mechanisms of epilimnetic PP, 1)

thermal stratification, which can limit the vertical exchange of nutrients hindering PP in surface waters and decreasing light attenuation, and 2) remote wind-forcing, which drives upwelling that, in turn, enhances PP and increases light attenuation. Furthermore, the strong vertical mixing may hinder the formation of a stable phototrophic community in deep waters, which have been shown to occupy a relatively stable water mass in marine oxygen deficit zones (Márquez-Artavia et al. 2019). We have subsequently re-written the causality of the ACM formation in the last paragraph of the discussion (see lines 399-423).

Meanwhile, there seem to be an important omission in the hydrochemical background description – O₂ distribution and the depth where oxygen deficient zone started at the two stations. Oxygen penetration was shallower at station 2 compared to Station 7, which most certainly has impacts on water column biogeochemistry. Although the authors to some extent attributed this to increased remineralisation of organic matter at the turbidity maximum at around 110m (lines 193-5), the measured C-remineralisation rate is not much different from that measured around 100m at Station 7 (no measurement was taken at exactly the same depth) that it is insufficient to fully explain the dramatic oxygen decline. This necessitates another explanation. The authors then further suggested that the above rapid O₂ utilisation led to deeper penetration of nitrate into oxygen deficient zone – On one hand, this argument makes little sense, as the depletion of O₂ would prompt earlier use of nitrate as an alternative electron acceptor such that it would decline in concentration more rapidly at a shallower rather than deeper depth – contrary to what was claimed. On the other hand, nitrate penetration depths are practically the same at the two stations, that it was the O₂ penetration depth that became shallower at station 2 instead. This larger overlap between oxygen deficient zone and presence of nitrate would certainly impact N-biogeochemistry, thus the underlying causes and consequences deserve more detailed discussion and analyses.

We thank the reviewer for the discussion on what governs the oxycline depth at stations 2 and 7. Reviewer 1 echoed a similar interest in this discussion. Indeed, the reviewer is correct to highlight that, at station 2, the oxycline was much shallower (~120 m) than compared to station 7 (~140 m; Fig. 2a, i), and the nitracline penetrated slightly deeper into the anoxic zone (Fig. 2b, j). At the bottom of the nitracline, station 2 supported an ACM spanning from 130-170 m depth (Fig. 2a). In the original manuscript, we had attributed the shallow oxycline to enhanced organic remineralization occurring at the bottom of the oxygenated zone. For instance, at station 2, the 105 m turbidity maximum, was associated with a particularly significant $\delta^{13}\text{C}_{\text{DIC}}$ decrease of $\Delta 0.8\text{‰}$ and a peak in organic carbon remineralization up to 51 nM C d^{-1} (Fig. 2c, d; Peak A). The enhanced remineralization in the oxygenated zone might contribute to the shallower oxycline at station 2, but as the reviewer inferred, it is likely not the sole explanation. Another possibility is that as a consequence of the ACM presence at station 2, enhanced rates of organic matter respiration in the anoxic zone could have contributed to the shoaling of the oxycline. Furthermore, the shoaling of the oxycline, in combination with active nitrification in the ACM ostensibly enabled deeper nitrate penetration into the anoxic zone compared to the ACM void station 7, located in the southern basin (Fig. 2b, j). Indeed, the ACM dominated northern basin has reported a historical shoaling of the oxycline (Verburg et al. 2003; Verburga and Hecky 2009). In the revised manuscript, we have now better address the underlying causes of the oxycline and nitracline penetration depth on lines 161-163, and later on lines 410-414.

Moreover, the shallowing of oxygen deficient zone at station 2 coincided with the occurrence of SCM – what are the potential explanation behind, and would there be any mechanistic relationship between? There have been suggestions that SCM occurs only when the oxygen deficient zone overlap with the 1% downwelling blue (490nm) irradiance. The PAR at the two stations appeared similar, except for the subsurface peak around the SCM depth which

indeed overlapped with oxygen deficient zone – though why a peak in PAR there also need further explanation.

At station 2, the light irradiance (measured in the morning sun, surface irradiance of $\sim 400 \mu\text{mol photons m}^{-2} \text{ s}^{-1}$) ceased to decrease at $\sim 100 \text{ m}$ (reaching a baseline of $0.05 \mu\text{mol photons m}^{-2} \text{ s}^{-1}$) (see the new revised Fig. S1). Although the maximum light penetration depth was likely deeper at mid-day given that the surface irradiance could have been up to $\sim 750 \mu\text{mol photons m}^{-2} \text{ s}^{-1}$. At station 7, the light penetration (measured at mid-day $\sim 750 \mu\text{mol photons m}^{-2} \text{ s}^{-1}$) was strongly attenuated with depth and ceased to decrease at $\sim 80 \text{ m}$ (Fig. S1), well above the top of the anoxic zone (145 m). The vertical upwelling in the Southern basin, which sustains high primary productivity and high turbidity in the epilimnion (Bergamino et al. 2010) (Fig. 2i), likely hindered deep light penetration into the anoxic zone. Whereas in the Northern basin, the reduced upwelling of nutrients diminishes epilimnetic primary productivity and increases the water clarity (Verburg et al. 2003; O'Reilly et al. 2003). Thus, we surmise that the increased light attenuation at station 2, which nearly overlapped with the oxygen deficient zone, ultimately contributed to the development of the ACM. The revised version of the manuscript now includes a more detailed description of light attenuation, as written above, on lines 166-183 and 193-196. Moreover, our overarching summary at the end of the manuscript (lines 399-423) also discusses the factors that control the fate of the ACM at greater length in the revised manuscript.

We agree with the reviewer that the small secondary increase in the PAR associated with the ACM at 145 m (mainly observed with the April 26 profile at station 2; see new Fig. S1a), is rather bizarre. Our current explanation is this could be attributed to changes in the surface irradiance (i.e. changes in sun angle, clouds etc (see new Fig. S1b), and/or possibly due to minor reflectance induced by particles and phototrophic cells at this depth (Fig. 2a, c). We have appended this brief explanation to the S1 figure caption.

The SCM itself was thought to be made of primarily cyanobacteria like *Synechococcus*. However, the sequencing data showed that *Synechococcus* only had a small contribution to the community ($\leq 1.5\%$), whereas there were a lot more green sulfur bacteria *Chlorobium* (up to 25%) that could also be counted as autofluorescent cells. This is quite different from how we understood SCM. How would this impact the water-column biogeochemistry? While on this topic, little information has been provided for the 16S rRNA gene sequencing results. How many reads were generated per sample? Are the values for % community referring to total number of sequencing reads obtained for each sample, or ASV? And if your dataset has been rarefied?

The reviewer is correct to highlight that *Synechococcus* contributed to only a small fraction of the 16S rRNA community survey. At the ACM, recovered 16S rRNA genes identified both *Synechococcales*, reaching up to 1.4% of the microbial community at 150 m depth; and *Chlorobium*, which attained up to 21.6% at a slightly deeper position of 156 m (Fig. S2b, c). *Chlorobium* is likely supported by the reduced sulfur flux from the hypolimnion (Durisch-Kaiser et al. 2011), coupled to its ability to thrive under extreme light conditions (Marschall et al. 2010). Although *Chlorobium* appears to dominate in the ACM, it is possible that the maximum *Synechococcus* peak was missed between 133-150 m, microstratification in two-layer phototrophic systems has been previously reported (Camacho et al. 2001). Thus, anoxygenic photosynthesis by green sulfur bacteria likely prevails in the ACM of Lake Tanganyika, however, we cannot preclude oxygenic photosynthesis by *Synechococcus*.

Niche partitioning between oxygenic and anoxygenic light harvesting zones has been reported in other highly stratified lakes, even at a vanishingly low light availabilities (Camacho et al. 2001; Brand et al. 2016). In euxinic basins and in stratified freshwater lakes, the ACM can comprise of Cyanophyceae (such as *Plankthotrix*, *Phormidium* and *Synechococcus*) (Repeta et al. 1989; Camacho

et al. 2001; Manske et al. 2005; Halm et al. 2009; Hamilton et al. 2016; Brand et al. 2016), that have also been shown to generate O₂ in situ (Brand et al. 2016). In addition, the reduced sulfur fluxes in these systems can sustain anoxygenic photosynthetic green sulfur bacteria that occupy the bottom layer of the ACM (Repeta et al. 1989; Camacho et al. 2001; Manske et al. 2005; Halm et al. 2009; Hamilton et al. 2016; Brand et al. 2016), due to their extreme light harvesting capabilities of only 0.001 μmol photons m⁻² s⁻¹ (Marschall et al. 2010). Unlike cyanobacteria, green sulfur bacteria such as *Chlorobium*, produce sulfur, and either assimilate dissolved inorganic nitrogen and/or fix N₂ into biomass partly satisfying local nitrogen loss (Halm et al. 2009). The co-occurrence of anoxygenic and oxygenic photosynthesis in the ACM, is not only common to stratified lakes and basins, but has also been hypothesized to be a widespread feature of the euxinic Proterozoic ocean (Johnston et al. 2009).

The ACM in euxinic basins and freshwater lakes is different than the ecology observed in marine oxygen deficient zones. In marine oxygen deficient zones, the ACM tends to be dominated by the low-light adapted cyanobacteria genera *Prochlorococcus* and to a lesser extent by *Synechococcus* (Johnson et al. 1999; Goericke et al. 2000; Franz et al. 2012; Garcia-Robledo et al. 2017; Fuchsman et al. 2019a; Aldunate et al. 2019). Oxygenic photosynthesis by cyanobacteria, has been shown to generate molecular oxygen (O₂) possibly supporting microaerobic nitrification in the ACM (Garcia-Robledo et al. 2017). The common thread between euxinic basins/freshwater lakes and marine oxygen minimum zones, is that both anoxygenic and/or oxygenic photosynthesis in the ACM have a capacity to fix CO₂ into biomass, contributing to an important production of organic matter (Thiel et al. 2018). We have clarified the ecological differences between these ACM dominated systems in the introduction on lines 95-111. In addition, we have now better described the co-occurrence of a two-layer phototrophic system in the ACM, discussed above, on lines 171-181 and 378-385.

We have added additional information regarding the 16S rRNA gene survey. The recovered 16S rRNA gene abundances for key taxa identified in Lake Tanganyika are based on the total number of amplicon sequence variants (AVSs). The average number of raw reads generated per sample were 15525, while the mean filtered AVS reads were 5189, with the rarefaction curve shown in Figure. S2f, I. In the revised manuscript, the above information is now detailed in the Methods on lines 568-570. Moreover, we have provided more taxonomic information pertaining to key heterotrophic bacteria, ammonium- and nitrite-oxidizing prokaryotes and planctomycetes on lines 245-247, 325-329, 334-337, respectively.

Based on concentration profiles of ammonium, the authors further calculated its diffusional fluxes, but it is not entirely clear over what depth or depth range those fluxes were calculated for, and that the use of what I understood as a single value for turbulent diffusivity for the entire water column seems unrealistic. A model was then applied to 'predict' ammonium concentration according to density. What are the boundary conditions used and the assumptions behind? The model used was based on simple one for nitrate in the Sargasso Sea with single source and sink and under the assumption, amongst others, that sink is much greater than supply. As there are multiple sources and sinks for ammonium within the oxygen deficient zone alone, the applicability of this model is highly doubtful. A reaction-diffusion model, such as the one developed by Berg et al (1998, L&O) would have been more appropriate.

We thank the reviewer of the discussion, and realize the flux information needed further clarification in the revised manuscript. It now reads as follows in the Methods on lines 448-452, "Fluxes of nitrate and ammonium were calculated according to Fick's law: $J_i = E_z \times (\partial C / \partial z)$, using the measured

concentrations and the vertical turbulent diffusivity in the thermocline (between 90-300 m) of $1.0 \times 10^{-5} \text{ m}^2 \text{ s}^{-1}$ previously determined by Durisch-Kaiser et al. 2011. Fluxes were calculated across the depths where nutrients exhibited a linear regression gradient ($R^2 = 0.95$, or greater)."

We agree with the reviewer that the modelled ammonium gradient according to the density profile, described by McGuilucuddy 1997, was not methodically outlined in the original manuscript, leading to a fair degree of confusion regarding its application in this study. After very careful consideration, we believe the predicted ammonium flux, based according to the density profile, was superfluous to the overall crux of the manuscript. The various rate transformation measurements coupled with the $\delta^{13}\text{C}_{\text{DIC}}$ profile, and the other chemical and physical parameters, together provide good evidence that the SCM sustains an active nitrogen cycle and enhanced ammonium production from organic matter remineralization. Thus, for the sake of brevity, we have replaced the analysis of the predicted ammonium flux, with a more thorough expansion of other aspects of the discussion that we believe provide greater context to the ACM and non-ACM conditions at stations 2 and 7, respectively. These expanded discussion points include a deeper description of the temperature, light, TOC, TON, and cell density profiles, as well as a more thorough discussion of the phototrophic ecology in the ACM (see lines 133-208).

As the reviewer suggested a reaction-diffusion model, such as the one developed by Berg et al (1998, L&O) might be more appropriate. Lam et al., 2007 (PNAS) have also applied a similar model to investigate zones of nitrification in the Black Sea chemocline. In our study, we believe that a modelled estimate of the various sinks and source, could help to better constrain the zones of anammox and nitrification, although, it might prove difficult to resolve zones of organic matter remineralization to ammonium. As stated earlier, the various rate transformation measurements coupled with the $\delta^{13}\text{C}_{\text{DIC}}$, and the other chemical and physical parameters provide good evidence that the ACM sustains active nitrogen and carbon cycling and along with enhanced organic matter remineralization rates. We nevertheless appreciate the suggestion by the reviewer.

On the discussion on remineralisation of organic nitrogen versus carbon, the production of DIC was compared against that NH_4 . On one hand, the authors should be cautious that the measured are not strictly gross rates as the consumption of DIC and NH_4 at the same depths were just as fast or faster, and that the impact would be at different degrees between the two due to the different processes involved in their respective consumption (apart from assimilation to organic matter). On the other hand, why did you expect them to be in similar magnitude, and even performed t-test, etc. (lines 226-235)? Even without the interference from concurrent production/consumption, their gross production should follow Redfield ratio and not 1:1. Besides, why used the total organic carbon to total nitrogen (TN) ratio to examine preferential organic nitrogen remineralisation? The TN, if not having been filtered, includes all forms of organic and inorganic, dissolved and particulate nitrogen (PON, DON, NH_4 , NO_2 , NO_3). Using this to compare against TOC doesn't tell you how much N has been remineralised – the value of TN itself would just stay the same before and after remineralisation – unless anammox/denitrification/ N_2 -fixation occurred.

We agree with the reviewer that we need to take the absolute rates of DIC and NH_4^+ production cautiously. Given the inherent complexity in carbon cycling, we suspect that there is some degree of uncertainty in our organic carbon remineralization rates, as the $^{13}\text{C}/^{12}\text{C}$ isotope ratio of DIC may be susceptible to dilution by concurrent processes. For instance, $^{12}\text{C}\text{-HCO}_3^-$ production from ambient organic matter remineralization, as well as the assimilation of remineralized $^{13}\text{C}\text{-HCO}_3^-$ into biomass could contribute to diluting the $^{13}\text{C}/^{12}\text{C}$ ratio. Apart from DIC production, a fraction of the remineralized algal biomass might accumulate in the dissolved organic carbon pool (i.e. acetate), which is missed in this survey, but that could be considered in future analyses. Moreover, different organic matter types and quality could be assessed as this may alter DIC production. Similar to the potential rates of

organic carbon remineralization, we suspect that the organic nitrogen remineralization rates are subject to a degree of uncertainty, as the $^{15}\text{N}/^{14}\text{N}$ ratio may be diluted by $^{14}\text{N}\text{-NH}_4^+$ production from ambient organic matter and by the rapid assimilation/dissimilation of remineralized $^{15}\text{N}\text{-NH}_4^+$. Despite the degree of uncertainty in the absolute values, the algal incubations provide insight into the potential for heterotrophic activity across the water column. We have detailed these caveats mentioned above with respect to organic matter remineralization to DIC and NH_4^+ on lines 257-267 and 304-313, respectively.

The reviewer was also correct to point out that analyzing the TOC:TN ratio was not an ideal indicator of preferential remineralization. In fact, the C:N ratio (either of TOC:TN or TOC:TON) can be convoluted by various processes, such as carbon and nitrogen fixation, which could mask selective remineralization. Furthermore, due to the degree of uncertainty in the absolute values, we downplayed the idea of selective nitrogen remineralization. Thus, in the revised manuscript, we have only briefly noted this observation, on lines 287-295, “Furthermore, the rates of organic nitrogen remineralization (44.9 nM N d^{-1}) exceeded the rates of organic carbon remineralization (24.4 nM C d^{-1}) in the ACM (Fig. 2e, f). Albeit, the DIC and NH_4^+ regeneration ratio of 0.5:1 is likely exaggerated by the addition of N-rich algal material (C:N ratio of 3.2) compared to the ambient organic matter pool (Fig. S3g, o). Nevertheless, previous studies have reported selective remineralization of nitrogen-rich organic matter, such as amino acids, coupled to denitrification in in-situ experiments (Van Mooy et al. 2002). In globally compiled marine datasets, selective remineralization of nitrogen over carbon was shown to increase with depth (from the thermocline down to intermediate waters), with models also reproducing this pattern of preferential heterotrophic remineralization (Letscher and Moore 2015; Zakem and Levine 2019).”

On the balance between autotrophy and heterotrophy, while it is apparent that there is definitely higher CO_2 fixation than remineralisation at the upper boundary of the oxygen deficient zone at Station 2, and also higher than observed at station 7, the authors should bear in mind that the detection limit of the two process rates are different (probably by an order of magnitude). Stable isotope tracers are not as sensitive as e.g. radiolabelled tracers, to measure CO_2 fixation considering the amount of water sample used, and it is not exactly a fair comparison against remineralisation to examine the ‘balance’ between the two processes. Hence, some of the conclusions drawn especially regarding whether it was a predominantly autotrophic or heterotrophic community in the oxygen deficient zone and immediately above, may be overinterpreted (lines 185-205).

We thank the reviewer for the discussion concerning the balance of autotrophy and heterotrophy in the water column of stations 2 and 7. We agree with the reviewer that in the original manuscript we placed a large emphasis on the absolute values of organic matter remineralization to resolve the “balance” between autotrophy and heterotrophy. Given our more thorough examination of the caveats associated with the algal addition experiment (now indicated on lines 257-267 and 304-313), we have subsequently framed the organic carbon remineralization experiments as indicators of potential activity across the water column. Hence, we have de-emphasized the absolute values and their magnitude with respect to the rates of carbon fixation on lines 216-243. In this section, we place more weight on the heavy/light shifts in the $\delta^{13}\text{C}_{\text{DIC}}$ profile to indicate changes in autotrophic and heterotrophic processes, which is combined with rates of carbon fixation to provide additional support for the $\delta^{13}\text{C}_{\text{DIC}}$ heavy signal, especially in the primary chlorophyll maximum and in the ACM. Whereas, the secondary increase in organic matter remineralization rates in the ACM, in combination with the carbon fixation rates, and the noisy $\delta^{13}\text{C}_{\text{DIC}}$, and TOC profiles, provide ancillary support for the co-occurrence of autotrophic and heterotrophic processes.

While it is obvious that the station with an SCM showed greater overall N-loss, it would have

been good to examine how the amount of organic matter actually affect N-cycling/N-loss. Although TOC/TN was measured, it was unclear how many samples were taken and the data were not shown. Also, since nitrate/nitrite/ammonium have been measured, an estimate could be made for at least TON. Both TOC and TON values could be compared with process rates data to further evaluate the influence from organic matter, as hypothesised in Introduction and also shown by others in at least marine oxygen deficient waters.

We thank the reviewer for highlighting this missing aspect. In the revised manuscript we now show the TOC, TON, and TN profiles, as well as the corresponding C:N_{TOC:TON} ratio in the supplementary Figure S3. In addition, we have added a brief description of the results on lines 188-191 and on 196-198, relating to stations 2 and 7 respectively. At station 2, we find that total organic carbon (TOC) and total organic nitrogen (TON) profiles exhibited a relatively noisy vertical structure from 130-150 m depth, with TOC concentrations fluctuating from 85 μM to 109 μM , and TON concentrations ranging from 6 μM to 11 μM (Fig S3d, e). At station 7, the top of anoxic zone, was denoted by a small turbidity signal at 145 m, and was accompanied also by increases in TOC (101-115 μM) and TON concentrations (6-10 μM ; Fig. S3m, n).

Overall, this study has presented a very nice set of data that in itself deserve publication, but the manuscript in its current state needs more work on its data interpretation and Discussion, to focus on what the data were actually showing, whilst avoiding overinterpretation and unnecessary speculations.

We thank the reviewer for their comments. We believe that the changes made to the manuscript according to the above discussions have greatly contributed to improving the manuscript, and have reduced unnecessary speculation.

Other specific comments:

Line 46: should bear in mind that your measured fluxes are not strictly gross rates though. Should correct for isotope dilution for the concurrent consumption/production processes.

We agree with the reviewer that the measured organic matter remineralization rates are not strictly gross rates, given the concurrent consumption and production processes of DIC and NH_4^+ . Although, given these inherent complexities and the various interacting processes, outlined on lines 244-267 and 304-310, it would be very difficult to accurately correct for all the concurrent isotope dilutions effects, which might also lead to greater error in the potential rates. Thus, after careful consideration, we have opted to keep the rates as is, but to clarify we have explicitly indicated the caveats associated with the absolute rate values. Moreover, we have clarified that the algal incubations provide important insight into the potential for heterotrophic activity across the water column.

Line 131: “multiple layers with high turbidity” this is an exaggeration considering the turbidity signals

We agree that in the original version of the manuscript “multiple layers with high turbidity” was ambiguous and vaguely worded. In our side-by-side comparison (included below), we find that at station 2 the turbidity signal was relatively noisy spanning from 50 to 120 m, with moderately sized peaks occurring at 55, 68, 87 and 105 m depth (Fig. 2c). In contrast, at station 7, we find a much smoother turbidity profile over the same depth range (Fig. 2k), apart from the turbidity signal associated with the oxic-anoxic transition zone. In the revised manuscript, we have re-worded our description of the vertical turbidity and N^2 profiles for clarification on lines 154-161.

Side-by-side comparison of vertical turbidity profiles for stations 2 (left) and 7 (right). Panels are also shown in Figure 2c, k.

Line 132: I can't see an enhanced 'density' gradient at 110m.

We agree with the reviewer that the deep N^2 layer is subtle, particularly with respect to the large density peak in the epilimnion. In our side-by-side comparison (included below), we find that at station 2, a small increase in the N^2 stability can be seen from 95-120 m (Fig. 2c; Peak A), which also coincided with increased turbidity at 105 m depth (see above panels). Whereas, at station 7, the N^2 profile revealed no large changes in density, and reported relatively minor noise from 65-140 m depth (Fig. 2k). In the revised manuscript, we have re-worded our description of the vertical turbidity and N^2 profiles for clarification on lines 154-161.

Side-by-side comparison of vertical N^2 profiles for stations 2 (left) and 7 (right). Panels are also shown in Figure 2c, k.

Lines 137-8: Have you measured (or previously) Fe/Mn oxide precipitates here or microbial biomass at this depth here to support this statement?

The reviewer is correct to highlight that we did not measure Fe/Mn precipitates and therefore have removed this from the discussion. In the revised manuscript, we now include bacterial cell densities (see Fig. S3b, j), which indicate a small cell density maximum in the ACM at station 2, and a small maximum at the top of the anoxic zone at station 7 near the turbidity peak (145 m). This information is now detailed on lines 183-187 and 197-200 for stations 2 and 7, respectively.

Line 139: What is the evidence for upwelling?

Please refer to our earlier discussion of the vertical temperature profiles. Briefly, at station 7, the strong temperature gradient, the shallower thermocline/primary chlorophyll max (Figs 1 and 2); in combination with higher reported wind velocities in the Southern basin and enhanced primary productivity (Verburg and Hecky 2003; O'Reilly et al. 2003; Bergamino et al. 2010), provide evidence to support enhanced upwelling in the Southern basin. In the revised manuscript, we have better clarified these points discussed above on lines 134-148.

Line 145: See main comments above on 16S sequencing results reporting

We have added additional information regarding the 16S rRNA gene survey. The recovered 16S rRNA gene abundances for key taxa identified in Lake Tanganyika are based on the total number of amplicon sequence variants (AVSs). The average number of raw reads generated per sample were 15525, while the mean filtered AVS reads were 5189, with the rarefaction curve shown in Figure. S2f, j. In the revised manuscript, the above information is now detailed in the Methods on lines 568-570. Moreover, we have provided more taxonomic information pertaining to key heterotrophic bacteria, ammonium- and nitrite-oxidizing prokaryotes and planctomycetes on lines 245-247, 325-329, 334-337, respectively.

Lines 165-7: This modelled results of apparent ammonium production seems to go against your measured fluxes (Fig. 3) which showed an overall net ammonium loss

Please see also the main comment above related to the predicted ammonium gradient discussion. After very careful consideration, we believe this calculation in the original manuscript was superfluous. We have since re-written this section to expand on the description of temperature, light, TOC, TON, and cell densities profiles, as well as a more thorough discussion of the phototrophic ecology at the ACM. These added discussion points can be found throughout the section now re-titled, "Characterization of the ACM in Lake Tanganyika".

Lines 178-8: again, these are not gross rates and should be corrected for isotope dilution

Please see refer to our earlier discussion of the isotope dilution.

Line 185: Sharp density gradient – I don't see it.

We have reworded this sentence for clarity in the revised manuscript. On lines 221-224 it now reads as, "At the 105 m turbidity maximum at station 2, a particularly significant $\delta^{13}\text{C}_{\text{DIC}}$ decrease of $\Delta 0.8\text{‰}$ and a peak in organic carbon remineralization up to 51 nM C d^{-1} (Fig. 2c, d; Peak A)."

Line 187-8: increased particle retention time – you only have shown turbidity or concentration, which is not equivalent to turnover time

The reviewer is correct to highlight that the turbidity peak does not equate to increased particle

retention time. As mentioned above, we have reworded this sentence for clarity in the revised manuscript. On lines 221-226 it now reads as, “At the 105 m turbidity maximum at station 2, a particularly significant $\delta^{13}\text{C}_{\text{DIC}}$ decrease of $\Delta 0.8\text{‰}$ and a peak in organic carbon remineralization up to 51 nM C d^{-1} (Fig. 2c, d; Peak A). The presence of this deep stratified oxic layer possibly enhanced organic matter remineralization and limited the export of labile organic matter to the underlying oxygen deficient zone.”

Line 191: the so-called gradual decline is overly subtle and only down to top of oxygen deficient zone

We thank the reviewer for pointing this out, the use of “gradual decline” in this statement was rather vague. We have subsequently reworded this sentence for clarity. On lines 226-228, it now reads as, “In contrast, at station 7, a similarly sized local decrease in the $\delta^{13}\text{C}_{\text{DIC}}$ profiles were not as apparent from 50-140 m (Fig. 2l). Moreover, the oxygenated water column exhibited relatively consistent rates of organic carbon remineralization of $\sim 40 \text{ nM C d}^{-1}$ (Fig. 2m).

Line 220: only Planctomycetes and not anammox bacteria?

We thank the reviewer for raising this point. A microbial community survey of 16S rRNA genes, in the oxygen deficient zone at stations 2 and 7, recovered sequences affiliated to *Planctomycetes* comprising of the non-anammox clades OM190/CL500-3 as well as the anammox phylum *Candidatus Brocadia* (Fig. S2e, k). Previous full length 16S rRNA analyses targeting *Planctomycetes* have also identified the anammox genus *Scalindua*, including anammox lipids in the anoxic zone (Schubert et al. 2006), which co-occurred with *Thaumarchaeota* (Schouten et al. 2012). In the revised manuscript, we have added this information above on lines 334-339.

Line 223: single peak or not doesn't matter, it's the magnitude that is of concern here

We thank the reviewer for highlighting this, we have re-written the sentence for clarity. In the revised manuscript, on lines 296-298 it now reads as, “Whereas, at station 7, rates of organic nitrogen remineralization (with amended allylthiourea) were mostly below the limit of detection in the oxygen deficient zone, with the exception of a local maximum occurring at the top of the anoxic zone up to 14.4 nM d^{-1} at 148 m (Fig. 2n).”

Line 230: the C remineralisation rates = $\sim 3\text{x}$ and 3.5x of NH_4 production, meaning that the former ‘exceeded’ the latter by a factor of 2 and 2.5 respectively!

We thank the reviewer for highlighting this, we have since revised this paragraph according to our earlier discussion concerning selective remineralization. In the revised manuscript, on lines 287-290 it now reads as, “Furthermore, the volumetric rates of organic nitrogen remineralization (45 nM N d^{-1}) exceeded the rates of organic carbon remineralization (24 nM C d^{-1}) in the ACM (Fig. 2e, f). Albeit, the DIC and NH_4^+ regeneration ratio of $\sim 0.5:1$, is likely exaggerated by the addition of more N-rich algal material (C:N ratio of 3.2) compared to the ambient organic matter pool (Fig. S3g, o).”

Line 232, 234: same as above on ‘exceeding’ by how many folds

We thank the reviewer for pointing this out, we have since revised this paragraph according to our discussion above, please see lines 287-290.

Lines 244-6: there were also a number of autotrophs as shown by your anammox and nitrification rates. As mentioned earlier, your CO_2 fixation rate measurements were not as

sensitive your remineralisation measurements.. See earlier major comments on the organic nitrogen remineralization

We thank the reviewer for raising this discussion, we have since revised this paragraph according to our earlier discussion concerning selective remineralization.

Lines 283-6: repetitive from before

We agree with the reviewer, and have therefore removed this intermediary summary as it was superfluous.

Lines 293-4: What algae were in your labelled substrates?

We thank the reviewer for highlighting this missing information. The labelled algal biomass was *Spirulina platensis* (Sigma-Aldrich). It is now indicated in the Methods on line 490.

Lines 306+: This section can be shortened as some is rather repetitive

We thank the reviewer for the suggestion, we have subsequently condensed the previous last two sections in the original manuscript into one section re-titled "The role of the ACM in nitrogen cycling and its fate in Lake Tanganyika" on lines 377 to 429.

Line 329: also autotrophs were present

We agree with the reviewer this is now indicated, on line 387-389 it now reads as, "We posit that this local production of organic matter by deep photo/litho-autotrophs, combined with the accumulation of sinking organic material from the photic zone at the ACM, likely sustained the active heterotrophic community identified by our algal addition experiments in the ACM."

Lines 334+: As mentioned in main comments above, the link to stratification is weak based on the evidence and arguments presented here, and so this section is largely speculative

We thank the reviewer for raising this discussion, we have subsequently re-written this section of the manuscript according to our points raised earlier concerning the link to stratification (see lines 399-429). In short, we have placed less focus on stratification as the main driver of ACM formation, and instead emphasize the factors that control the fate of epilimnetic primary productivity (PP), which indirectly regulates light penetration to deeper depths. Lake Tanganyika studies highlight two possible controlling mechanisms of PP. 1) thermal stratification, which can limit the vertical exchange of nutrients hindering PP in surface waters and decreasing light attenuation, and 2) remote wind-forcing, which drives upwelling that, in turn, enhances PP and increases light attenuation.

Lines 346: The percentage of cyanobacteria is very low in your sample! There were way more green sulfur bacteria

Indeed, *Synechococcus* contributed to only a small fraction of the 16S rRNA community survey, please see earlier comment concerning *Synechococcus* and *Chlorobium*. At the ACM, recovered 16S rRNA genes identified both *Synechococcales*, reaching up to 1.4% of the microbial community at 150 m depth; and *Chlorobium*, which attained up to 21.6% at a slightly deeper depth of 156 m (Fig. S2b, c). Although *Chlorobium* appears to dominate in the ACM, it is possible that the maximum *Synechococcus* peak was missed between 133-150 m. Previous studies have reported microstratification in two-layer phototrophic systems (Camacho et al. 2001). Thus, anoxygenic photosynthesis by green sulfur bacteria likely prevails in the ACM of Lake Tanganyika, however, we cannot preclude oxygenic photosynthesis by *Synechococcus*. If oxygenic photosynthesis is active, in situ oxygen production by cyanobacteria like *Synechococcus* may support microaerobic nitrification and organic matter remineralization in these anoxic waters as proposed elsewhere (Stolper et al.

2010; De Brabandere et al. 2012; Kalvelage et al. 2015; Brand et al. 2016; Garcia-Robledo et al. 2017). In the revised manuscript, we have placed less of an emphasis on *Synechococcus*, and rather, expanded on the idea of a two-layer phototrophic ecosystem comprising of anoxygenic/oxygenic photosynthesis on lines 378-385, and in the revised Figure 3.

Lines 355-364: Speculations aside, if you consider the overall biological production of the water column, should also consider the degree of production that has shifted to the SCM. Has it balanced the reduced production in the primary chlorophyll maximum compared to the water column without SCM but higher surface production like Station 7?

We thank the reviewer for raising this interesting discussion point. If we use the CO₂ fixation rates as indicators of primary productivity, then depth integrated values across the epilimnetic photic zone are similar, or slightly higher, at station 7 (25 mmol m⁻² d⁻¹) than compared to station 2 (24 mmol m⁻² d⁻¹). At station 2, the ACM contributes to 7 mmol m⁻² d⁻¹ of additional PP (roughly 31% of the epilimnetic PP), which amounts to a total water column PP of 32 mmol m⁻² d⁻¹. Whereas the anoxic zone of station 7 contributes to 0.5 mmol m⁻² d⁻¹ (or 2% of epilimnetic PP), amounting to a total water column PP of 26 mmol m⁻² d⁻¹. Hence, the presence of the ACM at station 2, provides a moderate increase of PP overall. Arguably, however, if the production is localized in the oxygen deficient zone, it may not stimulate higher biological production in terms of fish stocks in the more oxygenated surface waters. Again, we thank the reviewer for rising this interesting point. In the revised manuscript, we have briefly touched on this discussion above on lines 421-426, it reads as, "...the ACM in the Northern basin will strengthen and possibly expand southward to the central and southern basins. Under such a scenario, the ACM expansion could shift primary productivity to the oxygen deficient zone, increase the rate of organic matter remineralization, and in turn, accelerate the removal of fixed nitrogen by a nitrification-anammox coupling (Fig. 3a). The localization of primary productivity in the oxygen deficient zone and the enhanced nitrogen loss could further weaken the food web supporting fish productivity in the oxygenated surface waters."

Figure 3: Missing in this budget is the nitrite oxidation to nitrate, which would be important for the balance of nitrite as well as that of nitrate. Also, I think a table would suffice, and clearer for comparison purposes. The drawing doesn't add much.

The reviewer is correct to highlight that nitrite oxidation to nitrate is missing from our overall nitrogen budget. Nitrite oxidation to nitrate is essentially captured in our rates of ammonium oxidation to nitrate, which encompass ammonium oxidation to nitrite and subsequently nitrite oxidation to nitrate. We reason that the nitrite oxidation rate to nitrate would be similar or equivalent to the rate of ammonium oxidation to nitrate. For stations 2 and 7, the nitrite to nitrate oxidation rate would be 0.20 and 0.15 mmol m⁻² d⁻¹, respectively (these values are indicated in parentheses in the revised Fig. 3). In the revised version of the manuscript, we have updated this information in the category of nitrite sinks in Figure 3, and provided a brief note on this value based on the discussion above in the figure caption.

With respect to the conceptual model figure. We thank the reviewer for the suggestion, but would prefer to keep the conceptual model figure, as it provides a relatively simple summary of the complex interaction of the various nitrogen transformation processes in Lake Tanganyika. It is in some ways, similar to the conceptual models presented in other nitrogen cycling based manuscripts e.g. (Lam et al. 2009; Kalvelage et al. 2013; Stief et al. 2017).

The 'Synechococcus pool' is not justified as a separate pool from 'Org', as you showed that they only contributed little to the microbial community actually, and certainly DIC would go directly into 'Org' as well, as there are other autotrophs beside Synechococcus. Nitrification should be shown as a 2-step process for budget purposes, unless you found comammox. Colour scheme for NO₂ and NO₃ have been mixed up on the budget side.

We thank the reviewer for raising this point and have indicated more broadly in the revised Figure 3 that photo-/lithoautotrophic processes contribute to the pool of organic matter synthesis in the ACM. This includes both oxygenic and anoxygenic photosynthesis, as well as lithoautotrophic activity associated with the various nitrogen cycling processes.

Indeed, nitrification is a two-step process, and the reviewer is correct to highlight that traditionally it is shown as such (unless of course comammox is present). At the same time, we did measure rates of ammonium oxidation to nitrate, but agree that this could be misconstrued as comammox. Our only work around for this would be to make a clarification note in the figure and figure caption. It now reads as follows, "Note, that our $^{15}\text{NH}_4^+$ additions used to determine rates of ammonium oxidation to nitrate, are referred to as nitrification (indicated by the asterisk), these measurements do not specifically imply that comammox (i.e. complete nitrification by a single organism (van Kessel et al. 2015)) was involved."

We also appreciate the reviewer's attention to detail, we have corrected the color scheme for nitrite and nitrate.

References cited in the response to reviewer comments:

- Aldunate, M., C. Henríquez-Castillo, Q. Ji, J. Lueders-Dumont, M. R. Mulholland, B. B. Ward, P. Dassow, and O. Ulloa. 2019. Nitrogen assimilation in picocyanobacteria inhabiting the oxygen-deficient waters of the eastern tropical North and South Pacific. *Limnol. Oceanogr.* **113**: 11315. doi:10.1002/lno.11315
- Alonso-Saez, L., P. E. Galand, E. O. Casamayor, C. Pedros-Alio, and S. Bertilsson. 2010. High bicarbonate assimilation in the dark by Arctic bacteria. *ISME J* **4**: 1581–1590. doi:http://www.nature.com/ismej/journal/v4/n12/supinfo/ismej201069s1.html
- Bergamino, N., S. Horion, S. Stenuite, Y. Cornet, S. Loiselle, P.-D. Plisnier, and J.-P. Descy. 2010. Spatio-temporal dynamics of phytoplankton and primary production in Lake Tanganyika using a MODIS based bio-optical time series. *Remote Sens. Environ.* **114**: 772–780. doi:https://doi.org/10.1016/j.rse.2009.11.013
- De Brabandere, L., B. Thamdrup, N. P. Revsbech, and R. Foadi. 2012. A critical assessment of the occurrence and extend of oxygen contamination during anaerobic incubations utilizing commercially available vials. *J Microbiol Methods* **88**: 147–154. doi:10.1016/j.mimet.2011.11.001
- Brand, A., H. Bruderer, K. Oswald, C. Guggenheim, C. J. Schubert, and B. Wehrli. 2016. Oxygenic primary production below the oxycline and its importance for redox dynamics. *Aquat. Sci.* **78**: 727–741. doi:10.1007/s00027-016-0465-4
- Bristow, L. A., C. M. Callbeck, M. Larsen, and others. 2017. N₂ production rates limited by nitrite availability in the Bay of Bengal oxygen minimum zone. *Nat. Geosci.* **10**: 24–29. doi:10.1038/ngeo2847
- Bristow, L. A., T. Dalsgaard, L. Tiano, and others. 2016. Ammonium and nitrite oxidation at nanomolar oxygen concentrations in oxygen minimum zone waters. *Proc. Natl. Acad. Sci.* **113**: 10601–10606. doi:10.1073/pnas.1600359113
- Callbeck, C. M., G. Lavik, T. G. Ferdelman, and others. 2018. Oxygen minimum zone cryptic sulfur cycling sustained by offshore transport of key sulfur oxidizing bacteria. *Nat. Commun.* **9**: 1729. doi:10.1038/s41467-018-04041-x
- Callbeck, C. M., G. Lavik, L. Stramma, M. M. M. Kuypers, and L. A. Bristow. 2017. Enhanced Nitrogen Loss by Eddy-Induced Vertical Transport in the Offshore Peruvian Oxygen Minimum Zone Y. Hong [ed.]. *PLoS One* **12**: e0170059. doi:10.1371/journal.pone.0170059
- Callieri, C., V. Slabakova, N. Dzhembekova, and others. 2019. The mesopelagic anoxic Black Sea as an unexpected habitat for *Synechococcus* challenges our understanding of global “deep red fluorescence.” *ISME J.* **13**: 1676–1687. doi:10.1038/s41396-019-0378-z
- Camacho, A., J. Erez, A. Chicote, M. Florín, M. M. Squires, C. Lehmann, and R. Backofen. 2001. Microbial microstratification, inorganic carbon photoassimilation and dark carbon fixation at the chemocline of the meromictic Lake Cadagno (Switzerland) and its relevance to the food web. *Aquat. Sci.* **63**: 91–106. doi:10.1007/PL00001346
- Capone, D. G., D. A. Bronk, M. R. Mulholland, E. J. Carpenter, D. A. Bronk, and D. K. Steinberg. 2008. Nitrogen Regeneration. *Nitrogen Mar. Environ.* 385–467. doi:10.1016/B978-0-12-372522-6.00008-6
- Dalsgaard, T., B. Thamdrup, L. Farías, and N. P. Revsbech. 2012. Anammox and denitrification in the oxygen minimum zone of the eastern South Pacific. *Limnol. Oceanogr.* **57**: 1331–1346. doi:10.4319/lno.2012.57.5.1331
- Durisch-Kaiser, E., M. Schmid, F. Peeters, R. Kipfer, C. Dinkel, T. Diem, C. J. Schubert, and B. Wehrli. 2011. What prevents outgassing of methane to the atmosphere in Lake Tanganyika? *J. Geophys. Res.* **116**: G02022. doi:10.1029/2010JG001323
- Ehrenfels, B., M. Bartosiewicz, A. S. Mbonde, and others. 2020. Thermocline depth and euphotic zone thickness regulate the abundance of diazotrophic cyanobacteria in Lake Tanganyika. *Biogeosciences Discuss.* **2020**: 1–21. doi:10.5194/bg-2020-214
- Erb, T. J. 2011. Carboxylases in Natural and Synthetic Microbial Pathways. *Appl. Environ. Microbiol.* **77**: 8466–8477. doi:10.1128/aem.05702-11
- Franz, J., G. Krahnemann, G. Lavik, P. Grasse, T. Dittmar, and U. Riebesell. 2012. Dynamics and stoichiometry of nutrients and phytoplankton in waters influenced by the oxygen minimum zone in the eastern tropical Pacific. *Deep Sea Res. Part I Oceanogr. Res. Pap.* **62**: 20–31. doi:10.1016/J.DSR.2011.12.004
- Fuchsman, C. A., H. I. Palevsky, B. Widner, and others. 2019a. Cyanobacteria and cyanophage contributions to carbon and nitrogen cycling in an oligotrophic oxygen-deficient zone. *ISME J.* **13**: 1–13. doi:10.1038/s41396-019-0452-6

- Fuchsman, C. A., B. Paul, J. T. Staley, E. V. Yakushev, and J. W. Murray. 2019b. Detection of Transient Denitrification During a High Organic Matter Event in the Black Sea. *Global Biogeochem. Cycles* **33**: 143–162. doi:10.1029/2018GB006032
- Galán, A., J. Faúndez, B. Thamdrup, J. F. Santibáñez, and L. Fariás. 2014. Temporal dynamics of nitrogen loss in the coastal upwelling ecosystem off central Chile: Evidence of autotrophic denitrification through sulfide oxidation. *Limnol. Oceanogr.* **59**: 1865–1878. doi:10.4319/llo.2014.59.6.1865
- García-Robledo, E., C. C. Padilla, M. Aldunate, F. J. Stewart, O. Ulloa, A. Paulmier, G. Gregori, and N. P. Revsbech. 2017. Cryptic oxygen cycling in anoxic marine zones. *Proc. Natl. Acad. Sci. U. S. A.* **114**: 8319–8324. doi:10.1073/pnas.1619844114
- Goericke, R., R. Olson, and A. Shalapyonok. 2000. A novel niche for *Prochlorococcus* sp. in low-light suboxic environments in the Arabian Sea and the Eastern Tropical North Pacific. *Deep Sea Res. Part I Oceanogr. Res. Pap.* **47**: 1183–1205. doi:10.1016/S0967-0637(99)00108-9
- Halm, H., N. Musat, P. Lam, and others. 2009. Co-occurrence of denitrification and nitrogen fixation in a meromictic lake, Lake Cadagno (Switzerland). *Env. Microbiol.* **11**: 1945–1958. doi:10.1111/j.1462-2920.2009.01917.x
- Hammersley, M. R., G. Lavik, D. Woebken, and others. 2007. Anaerobic ammonium oxidation in the Peruvian oxygen minimum zone. *Limnol. Oceanogr.* **52**: 923–933. doi:10.4319/llo.2007.52.3.0923
- Hamilton, T. L., D. A. Bryant, and J. L. Macalady. 2016. The role of biology in planetary evolution: cyanobacterial primary production in low-oxygen Proterozoic oceans. *Environ. Microbiol.* **18**: 325–340. doi:10.1111/1462-2920.13118
- Holtappels, M., G. Lavik, M. M. Jensen, and M. M. M. Kuypers. 2011. ¹⁵N-labeling experiments to dissect the contributions of heterotrophic denitrification and anammox to nitrogen removal in the OMZ waters of the ocean, p. 223–251. *In* G.K. Martin [ed.], *Methods in Enzymology*. Academic Press.
- Johnson, Z., M. L. Landry, R. R. Bidigare, S. L. Brown, L. Campbell, J. Gunderson, J. Marra, and C. Trees. 1999. Energetics and growth kinetics of a deep *Prochlorococcus* spp. population in the Arabian Sea. *Deep Sea Res. Part II Top. Stud. Oceanogr.* **46**: 1719–1743. doi:10.1016/S0967-0645(99)00041-7
- Johnston, D. T., F. Wolfe-Simon, A. Pearson, and A. H. Knoll. 2009. Anoxygenic photosynthesis modulated Proterozoic oxygen and sustained Earth's middle age. *Proc. Natl. Acad. Sci.* **106**: 16925–16929. doi:10.1073/pnas.0909248106
- Kalvelage, T., G. Lavik, M. M. Jensen, and others. 2015. Aerobic Microbial Respiration In Oceanic Oxygen Minimum Zones. *PLoS One* **10**: e0133526. doi:10.1371/journal.pone.0133526
- Kalvelage, T., G. Lavik, P. Lam, and others. 2013. Nitrogen cycling driven by organic matter export in the South Pacific oxygen minimum zone. *Nat. Geosci.* **6**: 228–234. doi:10.1038/ngeo1739
- Kartal, B., M. M. M. Kuypers, G. Lavik, J. Schalk, H. J. M. Op den Camp, M. S. M. Jetten, and M. Strous. 2007. Anammox bacteria disguised as denitrifiers: nitrate reduction to dinitrogen gas via nitrite and ammonium. *Env. Microbiol.* **9**: 635–642. doi:10.1111/j.1462-2920.2006.01183.x
- van Kessel, M. A. H. J., D. R. Speth, M. Albertsen, P. H. Nielsen, H. J. M. Op den Camp, B. Kartal, M. S. M. Jetten, and S. Lücker. 2015. Complete nitrification by a single microorganism. *Nature* **528**: 555–559. doi:10.1038/nature16459
- Kuypers, M. M. M., G. Lavik, D. Woebken, M. Schmid, B. M. Fuchs, R. Amann, B. B. Jørgensen, and M. S. M. Jetten. 2005. Massive nitrogen loss from the Benguela upwelling system through anaerobic ammonium oxidation. *Proc Natl Acad Sci U S A* **102**: 6478–6483. doi:10.1073/pnas.0502088102
- Lam, P., M. M. Jensen, G. Lavik, and others. 2007. Linking crenarchaeal and bacterial nitrification to anammox in the Black Sea. *Proc. Natl. Acad. Sci.* **104**: 7104–7109. doi:10.1073/pnas.0611081104
- Lam, P., G. Lavik, M. M. Jensen, and others. 2009. Revising the nitrogen cycle in the Peruvian oxygen minimum zone. *Proc. Natl. Acad. Sci. U. S. A.* **106**: 4752–7. doi:10.1073/pnas.0812444106
- Lavik, G., T. Stührmann, V. Brüchert, and others. 2009. Detoxification of sulphidic African shelf waters by blooming chemolithotrophs. *Nature* **457**: 581–584. doi:10.1038/nature07588
- Lawson, C. E., S. Wu, A. S. Bhattacharjee, J. J. Hamilton, K. D. McMahon, R. Goel, and D. R. Noguera. 2017. Metabolic network analysis reveals microbial community interactions in anammox granules. *Nat. Commun.* **8**: 15416. doi:10.1038/ncomms15416
- Letscher, R. T., and J. K. Moore. 2015. Preferential remineralization of dissolved organic phosphorus and non-Redfield DOM dynamics in the global ocean: Impacts on marine productivity, nitrogen

- fixation, and carbon export. *Global Biogeochem. Cycles* **29**: 325–340.
doi:10.1002/2014GB004904
- Loginova, A. N., S. Thomsen, and A. Engel. 2016. Chromophoric and fluorescent dissolved organic matter in and above the oxygen minimum zone off Peru. *J. Geophys. Res. Ocean.* **121**: 7973–7990. doi:10.1002/2016jc011906
- Macomber, L., and J. A. Imlay. 2009. The iron-sulfur clusters of dehydratases are primary intracellular targets of copper toxicity. *Proc. Natl. Acad. Sci.* **106**: 8344–8349.
doi:10.1073/PNAS.0812808106
- Manske, A. K., J. Glaeser, M. M. M. Kuypers, and J. Overmann. 2005. Physiology and Phylogeny of Green Sulfur Bacteria Forming a Monospecific Phototrophic Assemblage at a Depth of 100 Meters in the Black Sea. *Appl Env. Microbiol* **71**: 8049–8060. doi:10.1128/aem.71.12.8049-8060.2005
- Márquez-Artavia, A., L. Sánchez-Velasco, E. D. Barton, A. Paulmier, E. Santamaría-Del-Ángel, and E. Beier. 2019. A suboxic chlorophyll-a maximum persists within the Pacific oxygen minimum zone off Mexico. *Deep Sea Res. Part II Top. Stud. Oceanogr.* **169–170**: 104686.
doi:https://doi.org/10.1016/j.dsr2.2019.104686
- Marschall, E., M. Jogler, U. Henßge, and J. Overmann. 2010. Large-scale distribution and activity patterns of an extremely low-light-adapted population of green sulfur bacteria in the Black Sea. *Environ. Microbiol.* **12**: 1348–1362. doi:10.1111/j.1462-2920.2010.02178.x
- Martens-Habbena, W., W. Qin, R. E. A. Horak, and others. 2015. The production of nitric oxide by marine ammonia-oxidizing archaea and inhibition of archaeal ammonia oxidation by a nitric oxide scavenger. *Environ. Microbiol.* **17**: 2261–2274. doi:10.1111/1462-2920.12677
- Milucka, J., M. Kirf, L. Lu, A. Krupke, P. Lam, S. Littmann, M. M. M. Kuypers, and C. J. Schubert. 2015. Methane oxidation coupled to oxygenic photosynthesis in anoxic waters. *ISME J.* **9**: 1991–2002. doi:10.1038/ismej.2015.12
- Van Mooy, B. A. ., R. G. Keil, and A. H. Devol. 2002. Impact of suboxia on sinking particulate organic carbon: Enhanced carbon flux and preferential degradation of amino acids via denitrification. *Geochim. Cosmochim. Acta* **66**: 457–465. doi:10.1016/S0016-7037(01)00787-6
- Müller, A. L., C. Pelikan, J. R. de Rezende, and others. 2018. Bacterial interactions during sequential degradation of cyanobacterial necromass in a sulfidic arctic marine sediment. *Environ. Microbiol.* **20**: 2927–2940. doi:10.1111/1462-2920.14297
- Naqvi, S. W. A., D. A. Jayakumar, P. V Narvekar, H. Naik, V. V. S. S. Sarma, W. D'Souza, S. Joseph, and M. D. George. 2000. Increased marine production of N₂O due to intensifying anoxia on the Indian continental shelf. *Nature* **408**: 346–349. doi:10.1038/35042551
- O'Reilly, C. M., S. R. Alin, P.-D. Plisnier, A. S. Cohen, and B. A. McKee. 2003. Climate change decreases aquatic ecosystem productivity of Lake Tanganyika, Africa. *Nature* **424**: 766–768. doi:10.1038/nature01833
- Oswald, K., J. S. Graf, S. Littmann, and others. 2017. Crenothrix are major methane consumers in stratified lakes. *ISME J.* **11**: 2124–2140. doi:10.1038/ismej.2017.77
- Oswald, K., C. Jegge, J. Tischer, and others. 2016. Methanotrophy under Versatile Conditions in the Water Column of the Ferruginous Meromictic Lake La Cruz (Spain). *Front. Microbiol.* **7**: 1762. doi:10.3389/fmicb.2016.01762
- Repeta, D. J., D. J. Simpson, B. B. Jorgensen, and H. W. Jannasch. 1989. Evidence for anoxygenic photosynthesis from the distribution of bacterio-chlorophylls in the Black Sea. *Nature* **342**: 69–72. doi:10.1038/342069a0
- Schouten, S., W. I. C. Rijpstra, E. Durisch-Kaiser, C. J. Schubert, and J. S. Sinninghe Damsté. 2012. Distribution of glycerol dialkyl glycerol tetraether lipids in the water column of Lake Tanganyika. *Org. Geochem.* **53**: 34–37. doi:https://doi.org/10.1016/j.orggeochem.2012.01.009
- Schubert, C. J., E. Durisch-Kaiser, B. Wehrli, B. Thamdrup, P. Lam, and M. M. M. Kuypers. 2006. Anaerobic ammonium oxidation in a tropical freshwater system (Lake Tanganyika). *Environ. Microbiol.* **8**: 1857–1863. doi:10.1111/j.1462-2920.2006.01074.x
- Schunck, H., G. Lavik, D. K. Desai, and others. 2013. Giant Hydrogen Sulfide Plume in the Oxygen Minimum Zone off Peru Supports Chemolithoautotrophy L.J. Stal [ed.]. *PLoS One* **8**: e68661. doi:10.1371/journal.pone.0068661
- Seitzinger, S., J. A. Harrison, J. K. Böhlke, A. F. Bouwman, R. Lowrance, B. Peterson, C. Tobias, and G. Van Drecht. 2006. Denitrification across landscapes and waterscapes: A synthesis. *Ecol. Appl.* **16**: 2064–2090. doi:10.1890/1051-0761(2006)016[2064:dalawa]2.0.co;2
- Stief, P., A. S. B. Lundgaard, Á. Morales-Ramírez, B. Thamdrup, and R. N. Glud. 2017. Fixed-Nitrogen Loss Associated with Sinking Zooplankton Carcasses in a Coastal Oxygen Minimum Zone (Golfo Dulce, Costa Rica) . *Front. Mar. Sci.* **4**: 152.

- Stolper, D. A., N. P. Revsbech, and D. E. Canfield. 2010. Aerobic growth at nanomolar oxygen concentrations. *Proc. Natl. Acad. Sci. U. S. A.* **107**: 18755–60. doi:10.1073/pnas.1013435107
- Thamdrup, B., and T. Dalsgaard. 2002. Production of N₂ through anaerobic ammonium oxidation coupled to nitrate reduction in marine sediments. *Appl. Environ. Microbiol.* **68**: 1312–1318. doi:10.1128/aem.68.3.1312-1318.2002
- Thiel, V., M. Tank, and D. A. Bryant. 2018. Diversity of Chlorophototrophic Bacteria Revealed in the Omics Era. *Annu. Rev. Plant Biol.* **69**: 21–49. doi:10.1146/annurev-arplant-042817-040500
- Ulloa, O., D. E. Canfield, E. F. DeLong, R. M. Letelier, and F. J. Stewart. 2012. Microbial oceanography of anoxic oxygen minimum zones. *Proc Natl Acad Sci U S A* **109**: 15996–16003. doi:10.1073/pnas.1205009109
- Verburg, P., and R. E. Hecky. 2003. Wind Patterns, Evaporation, and Related Physical Variables in Lake Tanganyika, East Africa. *J. Great Lakes Res.* **29**: 48–61. doi:https://doi.org/10.1016/S0380-1330(03)70538-3
- Verburg, P., R. E. Hecky, and H. Kling. 2003. Ecological Consequences of a Century of Warming in Lake Tanganyika. *Science (80-.)*. **301**: 505 LP – 507. doi:10.1126/science.1084846
- Verburg, P., and R. E. Hecky. 2009. The physics of the warming of Lake Tanganyika by climate change. *Limnol. Oceanogr.* **54**: 2418–2430. doi:10.4319/lo.2009.54.6_part_2.2418
- De Wever, A., K. Muylaert, K. Van der Gucht, S. Pirlot, C. Cocquyt, J.-P. Descy, P.-D. Plisnier, and W. Vyverman. 2005. Bacterial community composition in Lake Tanganyika: vertical and horizontal heterogeneity. *Appl. Environ. Microbiol.* **71**: 5029–37. doi:10.1128/AEM.71.9.5029-5037.2005
- Zakem, E. J., and N. M. Levine. 2019. Systematic Variation in Marine Dissolved Organic Matter Stoichiometry and Remineralization Ratios as a Function of Lability. *Global Biogeochem. Cycles* **33**: 1389–1407. doi:10.1029/2019GB006375

REVIEWER COMMENTS

Reviewer #1 (Remarks to the Author):

The authors undertook a conscientious revision of the manuscript, largely following the comments from the different reviewers. The title and the discussion of the results are now more focused on the data measured and shown in the manuscript. The over-interpretation of the results have decreased substantially, although I still consider some sections are speculative.

Although I believe it is a nice manuscript with interesting data to be published in Nature Communications, I still have some comments and suggestions:

- Title: it is more accurate than the previous one. However, I think that the ACM at Tanganyika Lake does not "accelerate" organic matter remineralization. ACM might increase the local production and enhance microbial processes but do not accelerate the decomposition of the OM.
- The ACM of the lake was mainly composed by *Chlorobium* (ca. 22% of community), being *Synechococcus* (max. 1.5%) about 15 times less abundant. In addition, light profiles show virtually no light reaching the ACM. As discussed in the manuscript, *Synechococcus* was most likely growing in a heterotrophic way as shown in the Black Sea and the carbon fixation was most likely coming from anoxygenic photosynthesis. Considering the evidence stated in the manuscript, local oxygen production in the ACM is not proved either, being different from the ACM in the OMZs and being more similar to the ACM of the Black Sea. In the Abstract, Introduction and Discussion there is still a tendency to enhance, even inflate, the relevance of *Synechococcus* and resemble the ACM to the ones found in the OMZs. In my opinion, there are clear differences between ACM of OMZs and this lake. Contrary to the OMZ ACM, the dominance of anoxygenic photosynthesis is clear in this ACM, being probably supported by H₂S (although no data are shown).
- Abstract: although I did not provide any comments in the previous version, it is quite vague and provides little information of the results obtained but only general information. I believe more specific data could be included in the Abstract, including quantitative data of the community composition and some of the relevant rates measured.
- L116-118: although sporadic plumes of sulfide were measured in the OMZs, euxinic bottom waters are not characteristic of marine OMZs.
- L155-160: Considering the noise of the profiles, I do not see the relevance of stressing these "peaks".
- L167-169 and Figure S1: The light reaching depths of the ACM seems to be the noise of the sensor. Indeed, the signal gets stable at values of ca. 0.05, being probably the zero signal of the sensor. In consequence, the statement of light reaching the ACM is highly doubtful. In addition, light was attenuated in the upper layers but probably extinguished at 100 m depth.
- L174-176: These sentences are speculative and try to introduce an idea without the support of the measured data.
- L176-end of the paragraph: Considering the light profiles provided in S1 and the data presented in the manuscript, the persistence of *Synechococcus* might only be due to the heterotrophic mode of growth. The extra discussion seems to over-interpret the results.
- L184: The "small maximum" is based on one single low value, otherwise the profile seems quite linear to me.
- L201: total organic matter? Do you mean TOC? In addition, I do not really understand the meaning of this sentence.
- L244-end of the paragraph: I do not clearly see the meaning of this paragraph. What do you mean with over-represented? What is the reference? The comparison with a sediment community does not seem to be adequate, considering that it is a quite different system. Why is it relevant that some groups catabolize peptidases? And, do the authors mean photoautotrophic community instead of processes?
- L257-end of the paragraph: I do not believe that the dilution of the isotopes during the incubation period could cause any impact on the measured rates. Maximum rates of 1 $\mu\text{M d}^{-1}$ were measured (only for CO₂ fixation), while the potential DIC production was measured to be less than 75 nM d⁻¹. Considering the isotopic additions, it seems unlikely to produce significant changes in the isotopic

pools to modify to a large extent the measured rates. Considering the lack of control (or measurements) of the light during the incubations, could the C-fixation rates be overestimated instead of the DIC production rates underestimated?

- L279-282: Please add references to the figures.

- L291-end of the paragraph: It seems to be clear some error or limitation in the method and the justification of the measured rates are not convincing.

- L306-308: Similar to the previous comment, the low rates measured are not enough to modify the added pools.

- L332-end of the paragraph: This paragraph seems to be misplaced and probably could be merged with other paragraphs of the same section.

- L382: The abundance of green sulfur bacteria was much higher than *Synechococcus* at any depth, then they were not situated at a lower depth in the ACM.

- L470-474: No data are provided of the PAR intensity of the light incubations. The use of filters and aluminum foil could produce or not the desired light level. Considering the effective use of light by green sulfur bacteria, the light intensity during the incubation is a relevant data.

- Figure 3: this manuscript does not show the presence of active oxygenic photosynthesis in the ACM of the Tanganyika Lake. Presented data does not allow to prove this scheme with enhanced aerobic processes several meters below the oxycline.

- Figure S1: The graphs showing the surface light irradiance profile measured over time are not easy to understand. In addition, I do not really see the need of such graphs. As stated above, the line indicating the anoxic layer should be 1 μM , as used elsewhere, and not 10 μM .

Reviewer #2 (Remarks to the Author):

This version of the manuscript is much improved over the original. Now the title clearly reflects the content of the text. Nearly all of the confusing sentences and grammar have been cleaned up. The inclusion of expanded discussion sulfur oxidizing bacteria contributes greatly and alleviates many of my concerns. I think the manuscript is ready for publication in *Nature*. There are some minor grammatical things and a couple of minor omissions/corrections that they should think about, but I don't think they should stop publication.

Line lines. "ACM ,....., is considered a prevalent of all major marine upwelling systems". This is not true. Examples of major upwelling systems that don't have an ACM the entire California current upwelling system does not, most of the European west coast does not and most notably the entire equatorial upwelling system does not. Most if not all oxygen deficient zone upwelling systems do have one.

Line 118. "coastal marine upwelling systems". I would say "many" or "some". As examples of coastal upwelling systems that don't: the entire California current upwelling system does not. Most of the European west coast does not and ditto for much of West Africa.

Line 236. "eluded to" replace with "eluded" or better still "suggest".

Lines 347-349. "30N2 production was detected.....only at the top of the anoxic zone" then next sentence "In addition, 30N2 gas production was reported..." Second sentence contradicts first. Try "30N2 production was reported only at the upper and lower edges of the anoxic zone" or something like that. Also, they use "reported" is this the proper use of the word? To me "showed" is more straightforward.

Line 381. Sentence starting with "While green sulfur bacteria". This sentence this sentence is just a series of clauses; it seems to be lacking a subject and a verb.

Line 396. I think they overlooked to changing "oxygen minimum" to "oxygen deficient".

Line 399. Change "What governs the fate of" to "Factors that control" or something like that.

Line 427 delete "of"

Reviewer #3 (Remarks to the Author):

First of all, my sincere apologies to both the editor and the authors for the delay in my review – It has been challenging to find time this past summer to do all we have been tasked, and thanks for your patience and understanding.

In this revision, Callbeck and co-authors have done a very good job in addressing the questions and comments previously raised by myself and other reviewers. Having trimmed the unnecessary speculations and side points, while adding the previously missing data and information, has certainly made this manuscript a lot clearer, more focussed and scientifically robust. As mentioned before, this is an impressive and comprehensive dataset and I highly commended the excellent efforts the authors have put into this study. Lake Tanganyika is both biogeochemically and socio-economically important that findings from this study would be of much appeal to many. I look forward to seeing this study in its final published form.

I only have a few minor comments below, which I hope would help further improve the accuracy and clarity of the manuscript; and I think the manuscript would benefit from another round of proofread to weed out the grammatical errors and awkward/incomplete sentences – though I don't think that they should be a hindrance to its publication.

Specific comments:

Line 160 (and often throughout) – 'whereas' – word choice: inappropriate use.

Lines 162-4 – This would imply that it was nitracline has moved deeper at station 2 but in reality it occurred at the same depth while oxycline has shoaled... I think the important point to make is that nitrate was present within anoxic zone. Suggest rephrase.

Line 166 – it seems that the fluorescent cells would account for almost 50% DAPI cell counts? It's substantial and a good point to make also.

Line 169 – incomplete sentence

Line 201 – "...vertical depth profiles... converged...". I suppose you meant the 'maxima' of the various variables converged instead.

Line 208 – the cited De Wever et al (2005) study didn't report Chlorobium. They found instead green 'non'-sulfur bacteria and Synechococcus in the anoxic zone

Lines 219-20 – "a shift to heterotrophy" is misleading in my opinion as there was certainly still some autotrophic activities occurring e.g. by nitrifiers, perhaps better to say a shift to dominance by heterotrophy

Line 231 – Although the overall C:N of TOC/TON was higher at ~60-150 m, a closer scrutiny sees actually a dip in C:N at ~130-150 where there was active anammox and nitrification associated with OrgN from Fig S3 g and h. This in fact is more in agreement with your observed chemolithoautotrophic activities in the latter depth range, as C:N dropped closer to Redfield.

Line 246 – Bacteroidetes abundance results were not shown in Fig S2d,j. And why you particularly picked these few taxa were shown amongst others should probably be explained briefly.

Lines 248-50 – while I agree that Bacteroidetes and Firmicutes initially hydrolyze the particulate organics to smaller MW organic compounds like acetate, whether or not the hydrolysed products were subsequently utilised by sulfate- and nitrate- reducing bacteria are likely dependent on the environmental settings and local communities. Hence, unless you are showing evidence of the presence of these other functional groups in your samples, it shouldn't be stated as if it is going to be a similar case.

Lines 248-256 – While I could see that these info could be relevant to your results, you should be more explicit in how these may help explain your findings rather than leaving it to the readers to connect the dots themselves. Some rewriting needed.

Line 296 – “whereas” word choice

Line 304 – “although” inappropriate use of conjunctive, incomplete sentence

Lines 305-8 – and NH₄⁺ use by anammox and nitrification

Lines 345-350 – I don't think you can fully rule out denitrification here. The lack of 30N₂ production from 15NO₃ incubations could be due to the rapid turnovers of nitrite by nitrate reduction and nitrite oxidation, such that the amount of 15N-nitrite available for further denitrification was likely limited. It's easier to check with 15NO₂ (+14NH₄) incubation instead. And indeed, you indicated there was some 30N₂ produced in the 15NO₂- + 14NH₄ incubations. Because you have included 14NH₄ in incubations, NH₄⁺ is not limiting that it is unlikely that anammox bacteria would take up the produced 15NH₄⁺ from DNRA to produce 30N₂. This is different from the anammox-DNRA coupling observed in the Arabian Sea (Jensen et al, 2011) in which 30N₂ was produced from '15NO₂ only' incubations. Meanwhile, you actually cited the Kartal et al (2007), which in fact was not an anammox-DNRA 'coupling' but DNRA activity by the anammox bacteria themselves. In that study, 30N₂ was produced from 15NO₃ incubations in the presence of simple organics, such that anammox bacteria exhibit nitrate reducing activity and produce 15NO₂, two molecules of which were subsequently converted to N₂ via NO. Hence, unless you can show that stoichiometrically all 30N₂ production can be accounted for by your observed anammox activity plus DNRA from your incubations, and/or there were parallel evidence from e.g. molecular biological analyses on the lack of active denitrifiers, you cannot really rule out denitrifying activities. In fact, some of the Bacteroidetes and Firmicutes, both present in your samples, are known to have denitrifying capability.

Lines 366-7 – the processes themselves don't “converge”, perhaps the 'high rates of' or 'rate maxima of'?

Line 385 – I can't see why ref 29 is needed for this statement. Isn't this commonly known? Instead, why not show some of your own results to substantiate this statement, as you have calculated in your reply to reviewers' comments.

Line 408 – 'deep' density gradients?

Line 413 – 'deeper nitrate penetration' – see earlier comment

Lines 415-7 – awkward sentence

Line 418 – missing pronoun

Line 505 – He-purging for only 5 minutes? This seems short. Normally I see 15 min.

Line 512 – 'headspace' is not a verb

Fig 2 – unit missing for d13C-DIC

Fig 3 – I am still not convinced that this large figure is necessary – all the flows between pools are the same, except that all arrows are thicker in panel (a) bar DNRA and NH₄⁺ released. Hence, a side by side table for comparison would be more effective and informative in my opinion.

We would again like to thank the three reviewers for their support of this work, their attention to detail and their constructive criticism. Their due diligence has greatly contributed to improving the manuscript.

Below is a point-by-point response to the reviewers' comments, indicated in the red-type face. The grey highlighted line numbers refer to changes made in the revised manuscript (e.g. lines XX-XX). A complete list of references is included at the end of the response to reviewers' comments.

REVIEWER COMMENTS

Reviewer #1 (Remarks to the Author):

The authors undertook a conscientious revision of the manuscript, largely following the comments from the different reviewers. The title and the discussion of the results are now more focused on the data measured and shown in the manuscript. The over-interpretation of the results have decreased substantially, although I still consider some sections are speculative.

Although I believe it is a nice manuscript with interesting data to be published in Nature Communications, I still have some comments and suggestions:

We thank Reviewer 1 for their critical feedback. In short, reviewer 1 still has open questions regarding the relevance/role of *Synechococcus* in Lake Tanganyika, which we have further clarified in the revised manuscript (detailed below).

- Title: it is more accurate than the previous one. However, I think that the ACM at Tanganyika Lake does not “accelerate” organic matter remineralization. ACM might increase the local production and enhance microbial processes but do not accelerate the decomposition of the OM.

We thank the reviewer for the suggestion, and agree with the reviewer that it would be more accurate to replace “accelerates” to “enhances local...” in the title of the revised manuscript (line 1).

- The ACM of the lake was mainly composed by *Chlorobium* (ca. 22% of community), being *Synechococcus* (max. 1.5%) about 15 times less abundant. In addition, light profiles show virtually no light reaching the ACM. As discussed in the manuscript, *Synechococcus* was most likely growing in a heterotrophic way as shown in the Black Sea and the carbon fixation was most likely coming from anoxygenic photosynthesis. Considering the evidence stated in the manuscript, local oxygen production in the ACM is not proved either, being different from the ACM in the OMZs and being more similar to the ACM of the Black Sea. In the Abstract, Introduction and Discussion there is still a tendency to enhance, even inflate, the relevance of *Synechococcus* and resemble the ACM to the ones found in the OMZs. In my opinion, there are clear differences between ACM of OMZs and this lake. Contrary to the OMZ ACM, the dominance of anoxygenic photosynthesis is clear in this ACM, being probably supported by H₂S (although no data are shown).

We understand the point raised by the reviewer and their open question regarding the relevance/role of *Synechococcus*. We will address the point regarding its relevance first. Indeed, even though it is not as abundant as *Chlorobium* as the reviewer correctly stated above. The fact is, it is still identified in the ACM at a very distinct peak at 150 m depth at station 2, and indeed, depths above and below 150 m found much lower *Synechococcus* abundances (see Figure below). In contrast, at station 7, which has no ACM, 16S rRNA gene abundances of *Synechococcus* were even lower than at station

2. The abstract simply states that we find both *Chlorobium* and *Synechococcus*, which is true according to Figure S2. Unfortunately, the abstract word limit constrains us from providing additional values. Nevertheless, we have made clear in the revised manuscript that anoxygenic photosynthesis likely prevails in the ACM on lines 184-185, and have further reiterated this point in our discussion summary (lines 416-418), and again in the caption of figure 3 (lines 710-712). We have therefore highlighted that *Chlorobium* dominates in the ACM over *Synechococcus*, on multiple lines throughout the text in the revised manuscript.

Recovered 16S rRNA gene abundances for key taxa identified in Lake Tanganyika. The full figure is presented in the revised manuscript as Figure S2.

Regarding the specific role of *Synechococcus* (i.e. growth via oxygenic photosynthesis or heterotrophy). We have explained the two possible modes of growth by *Synechococcus* on lines 179-184 and this was made further clear that (lines 185-187), “the ACM likely sustains active anoxygenic photosynthesis by green sulfur bacteria. However, it is less clear if the low-light availability could support oxygenic photosynthesis by ACM *Synechococcus*, alternatively they may grow via chemoorganoheterotrophy.” We therefore clearly presented both modes of growth as open possibilities in Lake Tanganyika in the revised manuscript, as we cannot say definitely that one dominates over the other as this point.

What is also important to keep in mind, is that, we do observe the presence of an active nitrification community in the ACM at station 2 (Fig. 2g). The oxygen supporting nitrification could be supplied via vertical transport processes, alternatively a previous seminal study on this topic has invoked possible in situ oxygen production as another explanation for its presence in the anoxic zone (Garcia-Robledo et al. 2017).

Indeed, we stated in the previous revised version, that *Chlorobium* is “ostensibly” supported by the reduced sulfur flux from underlying waters. No sulfide was measured in this campaign, but it is well

established that the bottom waters of Lake Tanganyika are sulfidic (Durisch-Kaiser et al. 2011). It is therefore reasonable to propose that sulfide is supporting the presence of green sulfur bacteria at the ACM at station 2.

We agree that marine and freshwater oxygen deficient zones, with all their nuances, are different in many respects. But, as reviewer 3 had also eluded to in their opening remarks in the previous round, these highly stratified water bodies do share some similarities. For instance, the water column of coastal upwelling regions often observe large ammonium fluxes, the presence of dissolved sulfide, and in some cases reports of an ACM, along with active nitrogen cycling related to anammox, denitrification, nitrate reduction to nitrite and nitrification (Naqvi et al. 2000; Lavik et al. 2009; Ulloa et al. 2012; Schunck et al. 2013; Galán et al. 2014; Callbeck et al. 2018), in a similar metabolic model (Kalvelage et al. 2013), as presented here in Figure 3 (albeit without the detailed analysis of the ACM on nitrogen cycling). Lake Tanganyika, also exhibits very similar chemical zonation to other well studied euxinic basins, such as the Black Sea, Cariaco basin, and the Saanich inlet, which have also been compared to coastal marine oxygen deficient zones (Lam et al. 2009; Ulloa et al. 2012), and are akin to conditions that would be found in the ancient euxinic ocean (Johnston et al. 2009; Ulloa et al. 2012). Thus, we generally agree with the reviewer, and have been careful to make these distinctions clear in the manuscript, while still highlighting the potential to transfer knowledge between these stratified anoxic environments.

- Abstract: although I did not provide any comments in the previous version, it is quite vague and provides little information of the results obtained but only general information. I believe more specific data could be included in the Abstract, including quantitative data of the community composition and some of the relevant rates measured.

We carefully considered the reviewers suggestion, but the 150-word limit for the abstract makes this difficult to implement specific values. Moreover, we are following the abstract guidelines laid out by Nature Communications, which state that “The abstract — which should be no more than 150 words long and contain no references — should serve both as a general introduction to the topic and as a brief, non-technical summary of the main results and their implications.”

We thank the reviewer for the suggestion, but we don't consider adding community composition values from Figure S2, as a “main result” of the manuscript. This level of detail is made explicitly clear throughout the revised version as outlined above. We still believe our concluding statement, that “the ACM was a local hotspot of organic matter remineralization that controlled an important supply of ammonium driving a nitrification-anammox coupling, and thereby played a key role in regulating nitrogen loss in the oxygen deficient zone.” is a sufficiently clear main result.

- L116-118: although sporadic plumes of sulfide were measured in the OMZs, euxinic bottom waters are not characteristic of marine OMZs.

We partly agree with the reviewer that euxinic conditions are not characteristic of the entire OMZ. But, euxinic conditions are characteristic of shallow coastal OMZ. We would like to refer the reviewer to the recent studies tracking the remote sensing of sulfur plumes off the coast of Peru and Namibia by (Ohde 2018) and (Ohde and Dadou 2018). Briefly, these two studies suggest that multiple sulfur/sulfide can develop in OMZ shelf waters over prolonged periods of anoxia 180-200 days of the year (with the exception of the El Niño phase). This is further corroborated by the countless other remote sensing and ship-based studies that have detected sulfide/sulfur plumes in oxygen minimum zones. Some of these studies we have indicated by respective region: Eastern tropical South Pacific (Schunck et al. 2013; Sommer et al. 2016; Callbeck et al. 2018; Ohde 2018), Eastern Tropical South Atlantic (Weeks et al. 2004; Ohde et al. 2007; Lavik et al. 2009), and Arabian Sea (Naqvi et al. 2000, 2006; Shenoy et al. 2012; Shirodkar et al. 2018).

Nevertheless, we have slightly reworded this statement to read as (lines 115-118), “Located along the East African Rift, Lake Tanganyika (maximum depth of 1,470 m) supports an active nitrogen

cycling community at the top of anoxic zone with euxinic bottom waters at deeper depths (De Wever et al. 2005; Schubert et al. 2006; Durisch-Kaiser et al. 2011; Schouten et al. 2012), similar to the Black Sea and to some coastal marine oxygen deficient zones (Ulloa et al. 2012).”

- L155-160: Considering the noise of the profiles, I do not see the relevance of stressing these “peaks”.

We thank the reviewer for pointing this out. In the original manuscript we referred to the signal as “noisy”, but this was not properly worded. At station 2, the vertical turbidity signal from 50 to 120 m exhibited a high degree of vertical structure (Fig. 2c), particularly in contrast to station 7, where we find a much smoother turbidity profile over the same depth range (Fig. 2k).” We have re-worded our description of the vertical turbidity for clarification on lines 155-157.

Side-by-side comparison of vertical turbidity profiles for stations 2 (left) and 7 (right). Panels are also shown in Figure 2c, k.

- L167-169 and Figure S1: The light reaching depths of the ACM seems to be the noise of the sensor. Indeed, the signal gets stable at values of ca. 0.05, being probably the zero signal of the sensor. In consequence, the statement of light reaching the ACM is highly doubtful. In addition, light was attenuated in the upper layers but probably extinguished at 100 m depth.

We thank the reviewer for the discussion. Indeed, the reviewer is correct to highlight that PAR values are below the limit of detection for the sensor at 100 m depth. However, we have little doubt that some light must reach the ACM, otherwise the ACM would be non-existent. In other words, if light did not reach this depth at all, then station 2 would appear more akin to station 7 (i.e. void of an anoxic chlorophyll maximum). We would like to further reiterate that this specific PAR cast at station 2, was done in the morning when surface irradiance was $\sim 400 \mu\text{mol photons m}^{-2} \text{ s}^{-1}$, the afternoon sun is nearly twice that intensity ($\sim 750 \mu\text{mol photons m}^{-2} \text{ s}^{-1}$; see the revised figure S1). We have reworded this statement to further clarify this point, it now reads as (lines 167-172), “The photosynthetically active radiation (PAR) at the ACM was below the limit of detection at less than $<0.05 \mu\text{mol photons m}^{-2} \text{ s}^{-1}$ as the light irradiance was strongly attenuated at $\sim 100 \text{ m}$ depth (Fig. S1). However, because the PAR measurements were performed in the morning, the transition from morning to mid-day light, which would be a nearly 2-fold increase in surface light irradiance (Fig. S1), would have likely resulted in deeper light penetration at station 2. Albeit, the exact amount PAR that would reach the ACM at mid-day remains unclear.”

- L174-176: These sentences are speculative and try to introduce an idea without the support of the measured data.

We are simply being honest with the data. Since we have not taken 16S rRNA gene abundance samples from 133-150 m depth, it could certainly be possible that we have missed the maximum *Synechococcus* peak between 133-150 m (Fig. S2). The reason why we mention this, is because it has been established in previous studies that microstratification does exist in phototrophic systems (Camacho et al. 2001). These couple of sentences merely present this possibility, that one needs to consider when viewing the 16S rRNA diversity data, and the distribution of *Synechococcus*.

- L176-end of the paragraph: Considering the light profiles provided in S1 and the data presented in the manuscript, the persistence of *Synechococcus* might only be due to the heterotrophic mode of growth. The extra discussion seems to over-interpret the results.

Regarding the specific role of *Synechococcus*. We have explained the two possible modes of growth by *Synechococcus* on lines 179-184 and have explicitly indicated (lines 184-187), that, "...the ACM likely sustains active anoxygenic photosynthesis by green sulfur bacteria. However, it is less clear if the low-light availability could support oxygenic photosynthesis by ACM *Synechococcus*, alternatively they may grow via chemoorganoheterotrophy." We therefore have clearly presented both modes of growth as open possibilities in Lake Tanganyika in the revised manuscript, as we cannot say definitely that one dominates over the other as this point.

- L184: The "small maximum" is based on one single low value, otherwise the profile seems quite linear to me.

We thank the reviewer for raising this discussion. Both DAPI curves for stations 2 and 7 when compared side-by-side exhibit a striking similarity. Indeed, cell densities are strongly attenuated with depth at both stations as the reviewer indicates. Interestingly, however, the area around the oxic-anoxic transition zone observes a minimum at the top of the profile (~120 m) followed by a maximum (~120 m), which is followed again by a well-defined minimum at the bottom of the vertical profile at around (~165 m). Most importantly, the maximums in the DAPI cell counts coincided with the turbidity peaks at each respective station, suggesting that these trends are indeed associated. It is therefore important to view the DAPI cell counts in context with the turbidity profiles, which were included in side-by-side panels in Figure S3 for this reason. Moreover, we have indicated in-text that the turbidity peaks coincided with a small maximum in the DAPI cell counts at both stations in the previous revised version.

- L201: total organic matter? Do you mean TOC? In addition, I do not really understand the meaning of this sentence.

We agree with the reviewer that this was not clearly written, it now reads as (on lines 206-208), "Overall, at station 7, the maxima of total organic carbon/nitrogen, bacterial abundances, and turbidity, observed a small peak at the top of the anoxic zone, where ammonium and nitrate gradients also intersected (Fig. 2j)."

- L244-end of the paragraph: I do not clearly see the meaning of this paragraph. What do you mean with over-represented? What is the reference? The comparison with a sediment community does not seem to be adequate, considering that it is a quite different system. Why is it relevant that some groups catabolize peptidases? And, do the authors mean photoautotrophic community instead of processes?

We thank the reviewer for the highlighting these points and we realize our discussion of the 16S rRNA heterotrophic community data required further elaboration, also in accordance to a comment made by reviewer 3.

The use of “over-represented” was inaccurate and we have therefore changed this in the revised manuscript to read as (lines 270-276), “...we identified an abundance of Gammaproteobacteria, mostly within the order Betaproteobacteriales, which averaged 36% of the 16S rRNA microbial community at stations 2 and 7 (Fig. S2 d, j). In addition, we identified a moderate fraction of *Bacteroidetes* that consisted mainly of the orders Sphingobacteriales (8%), Chlorobiales (5%) and Ignavibacteriales (2%), as well as the phylum *Actinobacteria* (8.6%) (Fig. S2d, j). Alphaproteobacteria, Deltaproteobacteria and Firmicutes were also identified, but in lower relative abundances (1-3%).”

The marine sediment study that we referred to also used ^{13}C -algal biomass as a tracer for organic matter remineralization, like we did in this study. Muller et al., find that the added ^{13}C -algal material is degraded by *Bacteroidetes* and Firmicutes. We also identified *Bacteroidetes* in the water column in Lake Tanganyika and also used ^{13}C algal additions to quantify the remineralization of organic matter to DIC. The similarities between the studies are relevant, but realize this was not clearly stated. Hence, we have rephrased this in the revised manuscript, it now reads as (lines 278-282), “*Bacteroidetes* and Firmicutes were further shown, using similar ^{13}C -labelled algal additions in a marine sediment study, to initially catalyze the degradation of the algal tracer to smaller molecular weight compounds, such as acetate³⁹. We suspect that *Bacteroidetes* and Gammaproteobacteria could serve an analogous role in degrading organic matter in Lake Tanganyika, and in our own algal addition experiments.”

We agree with the reviewer that the connection to peptidases in this sentence was unclear. The intent of this sentence was to relay the idea that Ignavibacteria (which was also moderately abundant in the ACM) was another potentially relevant heterotrophic microbe that could contribute to the remineralization of proteins. We have since revised this part, it now reads as (lines 287-290), “*Ignavibacteria*, are non-pigmented, heterotrophic bacteria that are distantly related to Chlorobi; and have been previously shown in anaerobic anammox reactors to facilitate the remineralization of proteins, possibly providing substrates for anammox in a close metabolic coupling(Lawson et al. 2017).”

Overall, we have clarified and slightly expanded our description of the heterotrophic community in the anoxic waters of Lake Tanganyika, based on our discussion above, and according to a later comment by reviewer 3. The entire revised paragraph can be found on lines 264-295.

- L257-end of the paragraph: I do not believe that the dilution of the isotopes during the incubation period could cause any impact on the measured rates. Maximum rates of 1 $\mu\text{M d}^{-1}$ were measured (only for CO_2 fixation), while the potential DIC production was measured to be less than 75 nM d^{-1} . Considering the isotopic additions, it seems unlikely to produce significant changes in the isotopic pools to modify to a large extent the measured rates. Considering the lack of control (or measurements) of the light during the incubations, could the C-fixation rates be overestimated instead of the DIC production rates underestimated?

We thank the reviewer for the discussion, but we respectfully disagree. Isotope dilution effects are certainly possible and this is best exemplified in bacterial heterotrophic pure culture experiments. When $^{13}\text{CO}_2$ is given to heterotrophic cultures (which is originally intended to quantify rates of heterotrophic CO_2 fixation), Roslev et al. 2004, find that the $^{13}\text{C}/^{12}\text{C}$ ratio is progressively diluted with time as a result of the production of $^{12}\text{CO}_2$ from the organism's metabolism of unlabeled organic material. Roslev et al., state that, “dilution of CO_2 is factor that should be considered in studies involving heterotrophs”. In our ^{13}C -algal experiments the production of ^{13}C -DIC is also susceptible to

concurrent ^{12}C -DIC dilution as a result of the metabolism of unlabeled ambient organic material. Thus, we generally believe this statement to be true in the manuscript.

It is also important to reiterate that, "Apart from DIC production, a fraction of the remineralized algal biomass might also accumulate in the dissolved organic carbon pool (i.e. as acetate), which is missed in our quantification of the rates of organic matter remineralization." These caveats possibly contribute to an underestimation of the DIC production rates, which were much lower relative to the rates of carbon fixation.

The carbon fixation rates were performed according to previous studies applying various light filters to mimic the light irradiances at depths (which were determined by in situ sensors). Our measured rates of carbon fixation in the ACM of $0.42 \mu\text{M C d}^{-1}$ (Fig. 2d), were in close agreement with values reported in other green sulfur bacteria containing chemoclines of $0.55\text{-}2.04 \mu\text{M C d}^{-1}$ ^{19,66}. Thus, our measured rates of carbon fixation do not appear to be anomalously high compared to other chemoclines carbon fixation studies. We have now more clearly indicated this on lines 515-517 and on lines 528-529 in the revised manuscript.

- L279-282: Please add references to the figures.

We thank the reviewer for pointing this out, we have added the reference to the figure in the revised version on line 309.

- L291-end of the paragraph: It seems to be clear some error or limitation in the method and the justification of the measured rates are not convincing.

We thank the reviewer for the discussion. The reviewer seems to be critiquing the section discussing selective organic matter remineralization, and whether the limitations associated with our measurements from the $^{13}\text{C}/^{15}\text{N}$ -algal additions can yield convincing rate values.

The limitations of the experiments as previously indicated in-text, have a degree of uncertainty in the absolute values. Indeed, we have been upfront in the manuscript that our rates represent "potential" activity, as we are adding a tracer in a slightly higher concentration than the ambient background (50%), with a C:N make-up that is more N-rich.

Even with these limitations (i.e. the addition of N-rich substrate), our rate values are still well within reason. Our measured organic nitrogen remineralization rates, which reached up to $50 \text{ nM N L}^{-1} \text{ d}^{-1}$, are within the range of values predicted in other oxygen deficient zones ($21\text{-}333 \text{ nmol N L}^{-1} \text{ d}^{-1}$) (Kalvelage et al. 2015). Congruently, the rates of organic nitrogen remineralization from our algal additions are, similar to the other various coupled nitrogen transformation processes (presented in Fig. 3). If we found that the rates of organic nitrogen remineralization were much higher compared to the rates of other nitrogen cycling processes, then one could perhaps argue that the addition of the N-rich algal substrate caused superficially enhanced remineralization activity. However, this is clearly not the case (Figure 3). Thus, generally speaking the absolute values are not significantly anomalous, and thus don't appear to be artifact induced. Rather the organic remineralization rates are in line with previous studies, and with our own various N-transformation processes. We therefore have a high degree of confidence in the values.

Coming back to the specific topic of selective organic matter remineralization. We made the simple observation that, "the volumetric rates of organic nitrogen remineralization (45 nM N d^{-1}) exceeded the rates of organic carbon remineralization (24 nM C d^{-1}) in the ACM (Fig. 2e, f)." It is possible that the DIC and NH_4^+ regeneration ratio of $\sim 0.5:1$, is exaggerated as a result of the addition of more N-rich algal material (C:N ratio of 3.2) compared to the ambient organic matter pool (Fig. S3g, o). However,

this does not fully explain why we don't observe the same phenomena in the oxygenated surface waters. In fact, the same algal additions when applied in the oxygenated water column at both stations 2 and 7, find the opposite pattern, whereby the rates of organic carbon remineralization often exceed the rates of organic nitrogen remineralization (we have now clarified this on lines 317-323). Alternatively, previous water column studies have reported selective remineralization of nitrogen-rich organic matter, such as amino acids, coupled to denitrification in in-situ experiments (Van Mooy et al. 2002). In globally compiled marine datasets, selective remineralization of nitrogen over carbon was shown to increase with depth (from the thermocline down to intermediate waters), with models also reproducing this pattern of preferential heterotrophic remineralization (Letscher and Moore 2015; Zakem and Levine 2019). We have further clarified (lines 327-329) that while, "...it's tempting to speculate that the ACM contributes to selective organic nitrogen remineralization, this requires further investigation in the future using various labeled organic matter substrate additions."

We strongly believe that the topic of selective remineralization in the ACM does warrant some form of discussion in the manuscript, since we used double labelled algal material. Keep in mind that we have also greatly downplayed this discussion from the previous version of the manuscript. Moreover, we have ended this section on a hypothesis driven note for future research.

- L306-308: Similar to the previous comment, the low rates measured are not enough to modify the added pools.

We thank the reviewer for the discussion, but we respectfully disagree. Similar to the ^{13}C -DIC example provided earlier, dilution by concurrent processes also effects the ^{15}N - NH_4^+ pool. Indeed, previous studies have even used the dilution of ^{15}N labelled NH_4^+ in incubation experiments as a method to quantify rates of ammonium regeneration in the water column (Gilbert et al. 1982; Bronk and Steinberg 2008). It occurs by the same principles as outlined for ^{13}C -DIC, whereby unlabeled ambient organic matter is remineralized to produce $^{14}\text{NH}_4^+$, thereby diluting the $^{15}\text{NH}_4^+$ tracer produced from ^{15}N -algae remineralization over time.

Again, we would like to point out that our measured rates of organic nitrogen remineralization, which reached a maximum of 50 nM d^{-1} , were within the range of ammonium regeneration estimates for the water column of other oxygen deficient zones at $21\text{-}333 \text{ nmol N L}^{-1} \text{ d}^{-1}$ (Kalvelage et al. 2015). Congruently, the rates of organic nitrogen remineralization from our algal additions are, similar to the other various coupled nitrogen transformation processes (summarized in Fig. 3). We therefore have a high degree of confidence in the general values.

- L332-end of the paragraph: This paragraph seems to be misplaced and probably could be merged with other paragraphs of the same section.

We have carefully considered the reviewers suggestion, but we find it would be difficult to merge this with the other paragraphs, given that this section explores in detail the evidence for anammox vs denitrification as the main nitrogen removal pathway. Nevertheless, we have tried to clarify some sections of this paragraph. Most notably, we have moved the 16S rRNA work, previously mentioned at the top of the section, towards the end of the paragraph to lines 382-385, where we think it improves the overall readability.

- L382: The abundance of green sulfur bacteria was much higher than *Synechococcus* at any depth, then they were not situated at a lower depth in the ACM.

We agree with the reviewer, and have reworded this paragraph to better highlight that the phototrophic community was dominated by *Chlorobium* and to a lesser extent by *Synechococcus*. It

now reads as follows in the revised manuscript (lines 416-418), “The community of phototrophs in the ACM at station 2, which was comprised mainly of *Chlorobium* (21.6%; Fig. S2c) and to a lesser extent by *Synechococcus* (1.4%; Fig. S2b), were positioned between the gradients of nitrate and ammonium.”

- L470-474: No data are provided of the PAR intensity of the light incubations. The use of filters and aluminum foil could produce or not the desired light level. Considering the effective use of light by green sulfur bacteria, the light intensity during the incubation is a relevant data.

We thank the reviewer for raising this discussion. We applied various light filters to mimic the light irradiance at depth, according to the in situ PAR data collected at each station (Fig. S1). Even though the light levels were not monitored over the course of the incubation period, our measured rate at the ACM, of $0.42 \mu\text{M C d}^{-1}$, was in close agreement with values reported in other green sulfur bacteria containing chemoclines of $0.55\text{-}2.04 \mu\text{M C d}^{-1}$ (Gorlenko et al. 2005; Halm et al. 2009). We have added this statement above to the revised version on lines 515-517. We therefore have no reason to suspect that these potential rates are anomalous (also indicated on line 528-529). Furthermore, our ^{13}C rate measurements combined with the C:N ratio (Fig. S3g), and the $\delta^{13}\text{C}$ -DIC profile (Fig. 2d), provide strong support for active carbon fixation in the ACM at station 2, which directly contrasts with the ACM void station 7.

- Figure 3: this manuscript does not show the presence of active oxygenic photosynthesis in the ACM of the Tanganyika Lake. Presented data does not allow to prove this scheme with enhanced aerobic processes several meters below the oxycline.

The reviewer is correct to highlight that we do not show active oxygenic photosynthesis in the ACM. We have therefore changed the Figure 3 caption to read “The phototrophic community in the ACM comprised mainly of green sulfur bacteria (21.6%; Fig. S2c) and to a lesser extent by *Synechococcus* (1.4%; Fig. S2b).” We would like to further emphasize that in the revised version, the figure schematic and corresponding caption nowhere indicate that oxygenic photosynthesis is active.

What is also important to reiterate is that we do observe the presence of an active nitrification community in the ACM at station 2 (Fig. 2g). The oxygen supporting nitrification could be supplied via vertical transport processes, alternatively a previous seminal study on this topic has invoked possible in situ oxygen production as another explanation for its presence (Garcia-Robledo et al. 2017). We have only suggested this possibility on lines 432-435, “Moreover, if in situ oxygen production by ACM cyanobacteria like *Synechococcus* is active, this may further support microaerobic nitrification and organic matter remineralization in these anoxic waters as proposed elsewhere (Kalvelage et al. 2015; Brand et al. 2016; Garcia-Robledo et al. 2017). Alternatively, nitrification could be sustained by the vertical mixing of oxygen into the anoxic zone.”

- Figure S1: The graphs showing the surface light irradiance profile measured over time are not easy to understand. In addition, I do not really see the need of such graphs. As stated above, the line indicating the anoxic layer should be $1 \mu\text{M}$, as used elsewhere, and not $10 \mu\text{M}$.

We agree with the reviewer that the surface irradiance profiles were overly complicated. To simplify this figure, we have integrated the average surface irradiance values, previously shown in panels b and d, into the corresponding vertical irradiance profiles in panels a and c. Generally speaking, the surface irradiance values are relevant to our discussion of light penetration at station 2, see the revised version lines 169-172.

Also, for consistency sake we have delineated the oxygen deficit zone at an O₂ cutoff of 1 μM indicated by the dotted line, according to the reviewer suggestion. We greatly appreciate the reviewer's close attention to detail.

Reviewer #2 (Remarks to the Author):

This version of the manuscript is much improved over the original. Now the title clearly reflects the content of the text. Nearly all of the confusing sentences and grammar have been cleaned up. The inclusion of expanded discussion sulfur oxidizing bacteria contributes greatly and alleviates many of my concerns. I think the manuscript is ready for publication in Nature. There are some minor grammatical things and a couple of minor omissions/corrections that they should think about, but I don't think they should stop publication.

We greatly appreciate the reviewer's constructive criticism as well as their support of this work throughout the review process.

Line lines. "ACM ,....., is considered a prevalent of all major marine upwelling systems". This is not true. Examples of major upwelling systems that don't have an ACM the entire California current upwelling system does not, most of the European west coast does not and most notably the entire equatorial upwelling system does not. Most if not all oxygen deficient zone upwelling systems do have one.

We agree with the reviewer that the ACM is not found in all marine upwelling regions. In the revised manuscript, it now reads as (on line 98), "is considered a prevalent feature of most marine oxygen minimum zones (Goericke et al. 2000; Franz et al. 2012; Garcia-Robledo et al. 2017; Fuchsman et al. 2019a; Aldunate et al. 2019)".

Line 118. "coastal marine upwelling systems". I would say "many" or "some". As examples of coastal upwelling systems that don't: the entire California current upwelling system does not. Most of the European west coast does not and ditto for much of West Africa.

We agree with the reviewer. We have made this change in the revised manuscript, it now reads as (on lines 117-118), "similar to the Black Sea and to some coastal marine oxygen deficient zones (Ulloa et al. 2012)."

Line 236. "eluded to" replace with "eluded" or better still "suggest".

We agree with the reviewer. This is now changed in the revised manuscript according to the reviewer's suggestion on line 244.

Lines 347-349. "30N₂ production was detected.....only at the top of the anoxic zone" then next sentence "In addition, 30N₂ gas production was reported..." Second sentence contradicts first. Try "30N₂ production was reported only at the upper and lower edges of the anoxic zone" or something like that. Also, they use "reported" is this the proper use of the

word? To me “showed” is more straightforward.

We thank the reviewer for highlighting this contradiction. We have therefore reworded these two sentences according to the reviewer’s suggestion, it now reads as (on lines 376-378), “Interestingly, however, $^{30}\text{N}_2$ production was detected in $^{15}\text{NH}_4^+$ and $^{15}\text{NO}_2^-$ incubations, but only at the upper and lower edges of the anoxic zone where nitrification and DNRA activity occurred, respectively (Fig. 2g, h, o, p).”

Line 381. Sentence starting with “While green sulfur bacteria”. This sentence this sentence is just a series of clauses; it seems to be lacking a subject and a verb.

We agree with the reviewer that this sentence was a little awkward. In the revised manuscript, we have structured this part slightly differently now. It subsequently reads as (on lines 418-423), “Previous studies have demonstrated that *Chlorobium* and *Synechococcus* serve as important sinks of dissolved inorganic nitrogen (e.g. NO_3^- and NH_4^+)(Halm et al. 2009; Giardina et al. 2018) as well as organic nitrogen compounds (e.g. urea and amino acids)(Wawrik et al. 2009). In addition, *Chlorobium*, has also been shown to fix N_2 into biomass...”

Line 396. I think they overlooked to changing “oxygen minimum” to “oxygen deficient”.

We agree with the reviewer. This is now corrected in the revised manuscript on line 436.

Line 399. Change “What governs the fate of” to “Factors that control” or something like that.

We have made this change recommended by the reviewer on line 438 in the revised manuscript.

Line 427 delete “of”

Done.

Reviewer #3 (Remarks to the Author):

First of all, my sincere apologies to both the editor and the authors for the delay in my review – It has been challenging to find time this past summer to do all we have been tasked, and thanks for your patience and understanding.

In this revision, Callbeck and co-authors have done a very good job in addressing the questions and comments previously raised by myself and other reviewers. Having trimmed the unnecessary speculations and side points, while adding the previously missing data and information, has certainly made this manuscript a lot clearer, more focussed and scientifically robust. As mentioned before, this is an impressive and comprehensive dataset and I highly commended the excellent efforts the authors have put into this study. Lake Tanganyika is both biogeochemically and socio-economically important that findings from this study would be of much appeal to many. I look forward to seeing this study in its final published form.

We sincerely thank the reviewer for their efforts, and providing important feedback that has greatly contributed to improving the manuscript.

I only have a few minor comments below, which I hope would help further improve the accuracy and clarity of the manuscript; and I think the manuscript would benefit from another round of proofread to weed out the grammatical errors and awkward/incomplete sentences – though I don't think that they should be a hindrance to its publication.

We have given the manuscript a thorough proofread to tidy-up any awkward sentences, and other confusing wording.

Specific comments:

Line 160 (and often throughout) – ‘whereas’ – word choice: inappropriate use.

We agree that “whereas” was an inappropriate word choice. We thank the reviewer for point this out. We have therefore dropped the word “whereas” throughout the manuscript.

Lines 162-4 – This would imply that it was nitracline has moved deeper at station 2 but in reality it occurred at the same depth while oxycline has shoaled... I think the important point to make is that nitrate was present within anoxic zone. Suggest rephrase.

We agree with the reviewer that the wording here needed rephrasing. In the revised manuscript it now reads as (lines 162-163), “Furthermore, at station 2, the oxycline was positioned at a shallower depth and the nitracline was present well into the anoxic zone than compared to station 7 (Fig. 2a, b, i, j).”

Line 166 – it seems that the fluorescent cells would account for almost 50% DAPI cell counts? It's substantial and a good point to make also.

We thank the reviewer raising this point. We have subsequently added this information to the revised manuscript, it now reads as (lines 165-167), “Congruently, fluorescent cells, as detected by epifluorescence microscopy comprised nearly 40% of the total cell counts, or up to 4.6×10^5 cells ml^{-1} in the ACM.”

Line 169 – incomplete sentence

We agree with the reviewer. In the revised manuscript, we have subsequently reworded this sentence for clarity (lines 169-171), also in response to a comment made by Reviewer 1.

Line 201 – “...vertical depth profiles... converged...”. I suppose you meant the ‘maxima’ of the various variables converged instead.

We agree with the reviewer, and have therefore reworded this as follows (lines 206-208), “Overall, at station 7, the maxima of total organic carbon/nitrogen, bacterial abundances, and turbidity, observed a small peak at the top of the anoxic zone, where ammonium and nitrate gradients also intersected (Fig. 2j).”

Line 208 – the cited De Wever et al (2005) study didn't report Chlorobium. They found instead green ‘non’-sulfur bacteria and Synechococcus in the anoxic zone

We thank the reviewer for highlighting this. We have subsequently changed this to read as (line 214), “...and green non-sulfur bacteria with similarities to *Chloroflexi*,...”.

Lines 219-20 – “a shift to heterotrophy” is misleading in my opinion as there was certainly still some autotrophic activities occurring e.g. by nitrifiers, perhaps better to say a shift to dominance by heterotrophy

We have made the change suggested by the reviewer on line 227.

Line 231 – Although the overall C:N of TOC/TON was higher at ~60-150 m, a closer scrutiny sees actually a dip in C:N at ~130-150 where there was active anammox and nitrification associated with OrgN from Fig S3 g and h. This in fact is more in agreement with your observed chemolithoautotrophic activities in the latter depth range, as C:N dropped closer to Redfield.

We agree with the reviewer that there is a perceivable drop in the C:N ratio closer to Redfield, which is in line with where we have active anammox and nitrification. This is something that we had also noticed. Another possibility is that N₂ fixation might also contribute to such a drop in the C:N ratio, which could be due to N₂-fixation by *Chlorobium* (demonstrated in previous studies). Generally speaking we find the discussion of the C:N ratio can become a complicated topic of discussion and although we agree with the reviewer, we have not further expanded on our interpretation of the C:N ratio in the revised manuscript. We anyways would like to thank the reviewer for raising this point.

Line 246 – Bacteroidetes abundance results were not shown in Fig S2d,j. And why you particularly picked these few taxa were shown amongst others should probably be explained briefly.

We agree with the reviewer that in the previous version it seemed rather arbitrary why we picked certain taxa in our 16S rRNA analysis presented in Figure S2.

Indeed, certain phyla play an important role in the remineralization of organic matter, and the seminal water column study by Bergauer et al.,³⁸, indicate that phyla such as Alphaproteobacteria, Gammaproteobacteria, Deltaproteobacteria, Bacteroidetes and Actinobacteria, are key contributors to water column organic matter uptake and remineralization. These same major phyla were also identified in our 16S rRNA gene analysis in the anoxic waters at stations 2 and 7 (Fig. S2d). Specifically, we identified an abundance of Gammaproteobacteria, mostly within the order Betaproteobacteriales, which averaged 36% of the 16S rRNA microbial community at stations 2 and 7 (Fig. S2 d, j). In addition, we identified a moderate fraction of *Bacteroidetes* that consisted mainly of the orders Sphingobacteriales (8%), Chlorobiales (5%) and Ignavibacteriales (2%), as well as the phylum *Actinobacteria* (8.6%) (Fig. S2d, j). Alphaproteobacteria, Deltaproteobacteria and Firmicutes were also identified, but in lower relative abundances (1-3%).

We have further amended our discussion of the 16S rRNA heterotrophic community with the above information on lines 264-295. Moreover, we have added the 16S rRNA genes abundances of these aforementioned phyla in a revised Figure S2d and j, which also includes Bacteroidetes.

Lines 248-50 – while I agree that Bacteroidetes and Firmicutes initially hydrolyze the particulate organics to smaller MW organic compounds like acetate, whether or not the hydrolysed products were subsequently utilised by sulfate- and nitrate- reducing bacteria are likely dependent on the environmental settings and local communities. Hence, unless you are

showing evidence of the presence of these other functional groups in your samples, it shouldn't be stated as if it is going to be a similar case.

The point of the sentence was to highlight that Bacteroidetes and Firmicutes are linked to organic matter degradation based on previous studies. And as the reviewer suggested, whether organic matter is further degraded by "sulfate and nitrate-reducing bacteria" depends on the environment. In fact, the latter part of the sentence was generally superfluous to our point and has since been dropped in the revised manuscript, it subsequently reads now as (line 278-282), "Bacteroidetes and Firmicutes were further shown, using similar ¹³C-labelled algal additions in a marine sediment study, to initially catalyze the degradation of the algal tracer to smaller molecular weight compounds, such as acetate³⁹. We suspect that Bacteroidetes and Gammaproteobacteria could serve an analogous role in degrading organic matter in Lake Tanganyika, and in our algal addition experiments."

Lines 248-256 – While I could see that these info could be relevant to your results, you should be more explicit in how these may help explain your findings rather than leaving it to the readers to connect the dots themselves. Some rewriting needed.

We agree with the reviewer that the study cited and its relevance to this work was not made explicitly clear. Indeed, this section required some rewriting. In the revised manuscript, we have now clarified the 16S rRNA heterotrophic community data on lines 265-295.

Line 296 – "whereas" word choice

We have dropped "whereas" throughout the manuscript.

Line 304 – "although" inappropriate use of conjunctive, incomplete sentence

We have dropped the word "although".

Lines 305-8 – and NH₄⁺ use by anammox and nitrification

In the revised manuscript, we have made this change recommended by the reviewer on lines 342-343.

Lines 345-350 – I don't think you can fully rule out denitrification here. The lack of ³⁰N₂ production from ¹⁵N₂O₃ incubations could be due to the rapid turnovers of nitrite by nitrate reduction and nitrite oxidation, such that the amount of ¹⁵N-nitrite available for further denitrification was likely limited. It's easier to check with ¹⁵N₂O₂ (+¹⁴NH₄⁺) incubation instead. And indeed, you indicated there was some ³⁰N₂ produced in the ¹⁵N₂O₂ + ¹⁴NH₄⁺ incubations. Because you have included ¹⁴NH₄⁺ in incubations, NH₄⁺ is not limiting that it is unlikely that anammox bacteria would take up the produced ¹⁵NH₄⁺ from DNRA to produce ³⁰N₂. This is different from the anammox-DNRA coupling observed in the Arabian Sea (Jensen et al, 2011) in which ³⁰N₂ was produced from '¹⁵N₂O₂ only' incubations. Meanwhile, you actually cited the Kartal et al (2007), which in fact was not an anammox-DNRA 'coupling' but DNRA activity by the anammox bacteria themselves. In that study, ³⁰N₂ was

produced from $^{15}\text{NO}_3^-$ incubations in the presence of simple organics, such that anammox bacteria exhibit nitrate reducing activity and produce $^{15}\text{NO}_2^-$, two molecules of which were subsequently converted to N_2 via NO . Hence, unless you can show that stoichiometrically all $^{30}\text{N}_2$ production can be accounted for by your observed anammox activity plus DNRA from your incubations, and/or there were parallel evidence from e.g. molecular biological analyses on the lack of active denitrifiers, you cannot really rule out denitrifying activities. In fact, some of the *Bacteroidetes* and *Firmicutes*, both present in your samples, are known to have denitrifying capability.

We thank the reviewer for raising this discussion. But it's not entirely clear to us why $^{30}\text{N}_2$ production is not possible in $^{15}\text{NO}_3^-$ additions when ambient nitrite is limiting, or not available. If an organism is capable of denitrification and has the complete array of denitrification genes (*Nar*, *Nap*, *Nir*, *Nor*, and *Nos*), then the reaction presumably occurs inside the cell where two molecules of $^{15}\text{NO}_3^-$ are reduced to form $^{30}\text{N}_2$ (via nitrite). Because everything occurs inside the cell, $^{30}\text{N}_2$ is generated even if ambient dissolved nitrite concentrations were limiting. If denitrification is performed by a series of microbes (which can certainly be the case), then for example, nitrate is reduced to nitrite which is then exchanged into the environment where it is utilized by the next microbe. In this case, available ambient nitrite could be outcompeted for by anammox and nitrifiers. But of course, then one has to assume that denitrifiers are somehow less competitive at consuming the available nitrite compared to other nitrite-utilizing microbes (i.e. anammox, nitrifiers). If we assume the point made by the reviewer that nitrite is outcompeted for by other N-cycling processes, then the permanently nitrite-depleted waters of Lake Tanganyika could never support denitrification under in situ conditions.

Moreover, we still have to reconcile with the fact that past $^{15}\text{NO}_3^-$ additions performed by Schubert et al., 2007 (under nitrite limiting conditions) reported the production of $^{30}\text{N}_2$ gas. These experiments demonstrate that even under nitrite limiting conditions $^{30}\text{N}_2$ gas is, still, a possible product in Lake Tanganyika. In our $^{15}\text{NO}_3^-$ addition, since we did not see the same $^{30}\text{N}_2$ production (only $^{29}\text{N}_2$ production), we must presume that there was a general absence of active denitrification at the time of sampling. Otherwise we have to assume that the added $^{15}\text{NO}_3^-$ was limited by nitrite availability preventing further $^{30}\text{N}_2$ gas production, but this would not be consistent with the previous findings that observed $^{30}\text{N}_2$ production from $^{15}\text{NO}_3^-$ additions. We think that the easier explanation is that denitrification occurs sporadically as a result of the sudden inputs of organic matter (Dalsgaard et al. 2012; Fuchsman et al. 2019b), possibly stimulating heterotrophic denitrifiers such as *Bacteroidetes* and *Firmicutes*.

With respect to the DNRA-anammox activity. The reviewer states, "Because you have included $^{14}\text{NH}_4^+$ in incubations, NH_4^+ is not limiting that it is unlikely that anammox bacteria would take up the produced $^{15}\text{NH}_4^+$ from DNRA to produce $^{30}\text{N}_2$." Kartal et al., 2007 (which assumes DNRA by anammox themselves), find that even when anammox were supplied extremely high backgrounds of $^{14}\text{NH}_4^+$ (10 mM), labelled $^{30}\text{N}_2$ gas was still detected by anammox in $^{15}\text{NO}_3^-$ addition experiments. It is therefore possible that the $^{30}\text{N}_2$ production in our $^{15}\text{NO}_2^- + ^{14}\text{NH}_4^+$ (10 μM of each) experiment could be attributed to anammox, without invoking nitrite reduction to N_2 . We have clarified that this could involve either a DNRA-anammox coupling or that anammox could facilitate DNRA activity themselves citing both Kartal et al., 2007 and Jensen et al., 2011 on lines 380-381 in the revised manuscript.

We would like to avoid adding an even more convoluted discussion of all the various possible ^{15}N stable isotope interpretations. We also see the point raised by the reviewer, and hence agree that we would like to be careful not to completely attribute all N_2 production to anammox. For simplicity sake, we still support anammox activity as an important process producing N_2 , but have appended our discussion of denitrification vs anammox with the caveat that (on lines 389-391), "Even though anammox appeared to dominate at stations 2 and 7, we do not rule out the possibility that

denitrification could play an important role in Lake Tanganyika, especially following the development of large algal blooms⁵³.”

Lines 366-7 – the processes themselves don’t “converge”, perhaps the ‘high rates of’ or ‘rate maxima of’?

We have made this change suggested by the reviewer on line 404 in the revised manuscript.

Line 385 – I can’t see why ref 29 is needed for this statement. Isn’t this commonly known? Instead, why not show some of your own results to substantiate this statement, as you have calculated in your reply to reviewers’ comments.

We agree with the reviewer that this is relatively common knowledge. We therefore dropped ref 29 in the revised manuscript.

Line 408 – ‘deep’ density gradients?

For clarification we have reworded this sentence on lines 447-449 to read as, “This could be further compounded by the presence of density anomalies in the stratified water column (e.g. Fig. 2d; peak A) that physically isolate phototrophic communities at the base of photic zone.”

Line 413 – ‘deeper nitrate penetration’ – see earlier comment

Based on the earlier comment, this now reads as (on lines 451-453), “Furthermore, the shallowing of the oxycline, in combination with active nitrification in the ACM could have caused the nitracline to persist well into the anoxic zone compared to the southern basin (Fig. 2b, j).”

Lines 415-7 – awkward sentence

We thank the reviewer for pointing this out. We have rephrased this sentence, and have broken it into two parts in the revised manuscript. It now reads as (on lines 454-457), “In the southern basin, remote wind-forcing weakens the warming rate in the South⁶¹, and it also deepens the oxycline and enhances nutrient upwelling fuelling epilimnetic primary productivity³⁶. Higher primary production in surface waters, in turn, results in stronger water column light attenuation compared to the Northern basin, which likely limits the formation of an ACM (e.g. station 7).”

Line 418 – missing pronoun

We thank the reviewer for pointing this out, we have slightly rewritten this sentence in the revised version. It now reads as (lines 458-460), “Furthermore, the strong vertical mixing in the water column may prevent the formation of a stable phototrophic community in deep waters. In marine oxygen deficient zones, for example, phototrophic communities have been shown to occupy a relatively stable water mass²³.”

Line 505 – He-purging for only 5 minutes? This seems short. Normally I see 15 min.

We apologize, but this was a typo that went unnoticed in the previous manuscript. We thank the reviewer for highlighting this, but it should read as 15 minutes, instead of 5 minutes (line 550).

Line 512 – ‘headspace’ is not a verb

We agree with the reviewer - a verb was missing. It now reads as (on lines 557-558), “At land-based facilities, we added a 2 mL helium headspace to the exetainer samples. From the headspace, the...”

Fig 2 – unit missing for d13C-DIC

We have now added the missing unit in figure 2, it now reads as “d13C-DIC (‰)”.

Fig 3 – I am still not convinced that this large figure is necessary – all the flows between pools are the same, except that all arrows are thicker in panel (a) bar DNRA and NH₄⁺ released. Hence, a side by side table for comparison would be more effective and informative in my opinion.

We have carefully considered Figure 3 and what it provides to the manuscript. The reviewer is correct in pointing out that it’s not absolutely necessary. The figure could be dropped from the manuscript; however, we still firmly believe that it provides a relatively simple summary of the complex interaction of the various carbon and nitrogen transformation processes in Lake Tanganyika. Previous nitrogen cycling studies have also presented similar conceptual models highlighting their findings on the topic e.g. (Lam et al. 2009; Kalvelage et al. 2013; Stief et al. 2017). We thank the reviewer for raising this discussion.

References

- Aldunate, M., C. Henríquez-Castillo, Q. Ji, J. Lueders-Dumont, M. R. Mulholland, B. B. Ward, P. Dassow, and O. Ulloa. 2019. Nitrogen assimilation in picocyanobacteria inhabiting the oxygen-deficient waters of the eastern tropical North and South Pacific. *Limnol. Oceanogr.* **113**:15. doi:10.1002/lno.11315
- Bergamino, N., S. Horion, S. Stenuite, Y. Cornet, S. Loiselle, P.-D. Plisnier, and J.-P. Descy. 2010. Spatio-temporal dynamics of phytoplankton and primary production in Lake Tanganyika using a MODIS based bio-optical time series. *Remote Sens. Environ.* **114**: 772–780. doi:https://doi.org/10.1016/j.rse.2009.11.013
- Bergauer, K., A. Fernandez-Guerra, J. A. L. Garcia, R. R. Sprenger, R. Stepanauskas, M. G. Pachiadaki, O. N. Jensen, and G. J. Herndl. 2018. Organic matter processing by microbial communities throughout the Atlantic water column as revealed by metaproteomics. *Proc. Natl. Acad. Sci. U. S. A.* **115**: E400–E408. doi:10.1073/pnas.1708779115
- Brand, A., H. Bruderer, K. Oswald, C. Guggenheim, C. J. Schubert, and B. Wehrli. 2016. Oxygenic primary production below the oxycline and its importance for redox dynamics. *Aquat. Sci.* **78**: 727–741. doi:10.1007/s00027-016-0465-4
- Bronk, D. A., and D. K. Steinberg. 2008. Nitrogen Regeneration, p. 385–467. *In* D.G. Capone, D.A. Bronk, M. Mulholland, and E.J. Carpenter [eds.], *Nitrogen in the Marine Environment*. Academic Press.
- Callbeck, C. M., G. Lavik, T. G. Ferdelman, and others. 2018. Oxygen minimum zone cryptic sulfur cycling sustained by offshore transport of key sulfur oxidizing bacteria. *Nat. Commun.* **9**: 1729. doi:10.1038/s41467-018-04041-x
- Camacho, A., J. Erez, A. Chicote, M. Florín, M. M. Squires, C. Lehmann, and R. Backofen. 2001. Microbial microstratification, inorganic carbon photoassimilation and dark carbon

- fixation at the chemocline of the meromictic Lake Cadagno (Switzerland) and its relevance to the food web. *Aquat. Sci.* **63**: 91–106. doi:10.1007/PL00001346
- Dalsgaard, T., B. Thamdrup, L. Farías, and N. P. Revsbech. 2012. Anammox and denitrification in the oxygen minimum zone of the eastern South Pacific. *Limnol. Oceanogr.* **57**: 1331–1346. doi:10.4319/lo.2012.57.5.1331
- Durisch-Kaiser, E., M. Schmid, F. Peeters, R. Kipfer, C. Dinkel, T. Diem, C. J. Schubert, and B. Wehrli. 2011. What prevents outgassing of methane to the atmosphere in Lake Tanganyika? *J. Geophys. Res.* **116**: G02022. doi:10.1029/2010JG001323
- Franz, J., G. Krahnemann, G. Lavik, P. Grasse, T. Dittmar, and U. Riebesell. 2012. Dynamics and stoichiometry of nutrients and phytoplankton in waters influenced by the oxygen minimum zone in the eastern tropical Pacific. *Deep Sea Res. Part I Oceanogr. Res. Pap.* **62**: 20–31. doi:10.1016/J.DSR.2011.12.004
- Fuchsman, C. A., H. I. Palevsky, B. Widner, and others. 2019a. Cyanobacteria and cyanophage contributions to carbon and nitrogen cycling in an oligotrophic oxygen-deficient zone. *ISME J.* 1–13. doi:10.1038/s41396-019-0452-6
- Fuchsman, C. A., B. Paul, J. T. Staley, E. V Yakushev, and J. W. Murray. 2019b. Detection of Transient Denitrification During a High Organic Matter Event in the Black Sea. *Global Biogeochem. Cycles* **33**: 143–162. doi:10.1029/2018GB006032
- Galán, A., J. Faúndez, B. Thamdrup, J. F. Santibáñez, and L. Farías. 2014. Temporal dynamics of nitrogen loss in the coastal upwelling ecosystem off central Chile: Evidence of autotrophic denitrification through sulfide oxidation. *Limnol. Oceanogr.* **59**: 1865–1878. doi:10.4319/lo.2014.59.6.1865
- García-Robledo, E., C. C. Padilla, M. Aldunate, F. J. Stewart, O. Ulloa, A. Paulmier, G. Gregori, and N. P. Revsbech. 2017. Cryptic oxygen cycling in anoxic marine zones. *Proc. Natl. Acad. Sci. U. S. A.* **114**: 8319–8324. doi:10.1073/pnas.1619844114
- Giardina, M., S. Cheong, C. E. Marjo, and others. 2018. Quantifying Inorganic Nitrogen Assimilation by *Synechococcus* Using Bulk and Single-Cell Mass Spectrometry: A Comparative Study. *Front. Microbiol.* **9**: 2847. doi:10.3389/fmicb.2018.02847
- Gilbert, P. M., F. Lipschultz, J. J. McCarthy, and M. A. Altabet. 1982. Isotope dilution models of uptake and remineralization of ammonium by marine plankton. *Limnol. Oceanogr.* **27**: 639–650. doi:10.4319/lo.1982.27.4.0639
- Goericke, R., R. Olson, and A. Shalapyonok. 2000. A novel niche for *Prochlorococcus* sp. in low-light suboxic environments in the Arabian Sea and the Eastern Tropical North Pacific. *Deep Sea Res. Part I Oceanogr. Res. Pap.* **47**: 1183–1205. doi:10.1016/S0967-0637(99)00108-9
- Gorlenko, V. M., P. V Mikheev, I. I. Rusanov, N. V Pimenov, and M. V Ivanov. 2005. Ecophysiological properties of photosynthetic bacteria from the Black Sea chemocline zone. *Microbiology* **74**: 201–209. doi:10.1007/s11021-005-0052-5
- Halm, H., N. Musat, P. Lam, and others. 2009. Co-occurrence of denitrification and nitrogen fixation in a meromictic lake, Lake Cadagno (Switzerland). *Env. Microbiol* **11**: 1945–1958. doi:10.1111/j.1462-2920.2009.01917.x
- Johnston, D. T., F. Wolfe-Simon, A. Pearson, and A. H. Knoll. 2009. Anoxygenic photosynthesis modulated Proterozoic oxygen and sustained Earth's middle age. *Proc. Natl. Acad. Sci.* **106**: 16925–16929. doi:10.1073/pnas.0909248106
- Kalvelage, T., G. Lavik, M. M. Jensen, and others. 2015. Aerobic Microbial Respiration In Oceanic Oxygen Minimum Zones. *PLoS One* **10**: e0133526. doi:10.1371/journal.pone.0133526
- Kalvelage, T., G. Lavik, P. Lam, and others. 2013. Nitrogen cycling driven by organic matter export in the South Pacific oxygen minimum zone. *Nat. Geosci* **6**: 228–234. doi:10.1038/ngeo1739

- Kraemer, B. M., S. Hook, T. Huttula, and others. 2015. Century-Long Warming Trends in the Upper Water Column of Lake Tanganyika G. Bohrer [ed.]. *PLoS One* **10**: e0132490. doi:10.1371/journal.pone.0132490
- Lam, P., G. Lavik, M. M. Jensen, and others. 2009. Revising the nitrogen cycle in the Peruvian oxygen minimum zone. *Proc. Natl. Acad. Sci. U. S. A.* **106**: 4752–7. doi:10.1073/pnas.0812444106
- Lavik, G., T. Stührmann, V. Brüchert, and others. 2009. Detoxification of sulphidic African shelf waters by blooming chemolithotrophs. *Nature* **457**: 581–584. doi:10.1038/nature07588
- Lawson, C. E., S. Wu, A. S. Bhattacharjee, J. J. Hamilton, K. D. McMahon, R. Goel, and D. R. Noguera. 2017. Metabolic network analysis reveals microbial community interactions in anammox granules. *Nat. Commun.* **8**: 15416. doi:10.1038/ncomms15416
- Letscher, R. T., and J. K. Moore. 2015. Preferential remineralization of dissolved organic phosphorus and non-Redfield DOM dynamics in the global ocean: Impacts on marine productivity, nitrogen fixation, and carbon export. *Global Biogeochem. Cycles* **29**: 325–340. doi:10.1002/2014GB004904
- Márquez-Artavia, A., L. Sánchez-Velasco, E. D. Barton, A. Paulmier, E. Santamaría-Del-Ángel, and E. Beier. 2019. A suboxic chlorophyll-a maximum persists within the Pacific oxygen minimum zone off Mexico. *Deep Sea Res. Part II Top. Stud. Oceanogr.* **169–170**: 104686. doi:https://doi.org/10.1016/j.dsr2.2019.104686
- Van Mooy, B. A. ., R. G. Keil, and A. H. Devol. 2002. Impact of suboxia on sinking particulate organic carbon: Enhanced carbon flux and preferential degradation of amino acids via denitrification. *Geochim. Cosmochim. Acta* **66**: 457–465. doi:10.1016/S0016-7037(01)00787-6
- Müller, A. L., C. Pelikan, J. R. de Rezende, and others. 2018. Bacterial interactions during sequential degradation of cyanobacterial necromass in a sulfidic arctic marine sediment. *Environ. Microbiol.* **20**: 2927–2940. doi:10.1111/1462-2920.14297
- Naqvi, S. A. W., H. Naik, D. A. Jayakumar, M. S. Shailaja, and P. V Narvekar. 2006. Seasonal oxygen deficiency over the western continental shelf of India , p. 195–224. *In* N.L. Neretin [ed.], *Past and Present Water Column Anoxia*. Springer Netherlands.
- Naqvi, S. W. A., D. A. Jayakumar, P. V Narvekar, H. Naik, V. V. S. S. Sarma, W. D'Souza, S. Joseph, and M. D. George. 2000. Increased marine production of N₂O due to intensifying anoxia on the Indian continental shelf. *Nature* **408**: 346–349. doi:10.1038/35042551
- Ohde, T. 2018. Coastal Sulfur Plumes off Peru During El Niño, La Niña, and Neutral Phases. *Geophys. Res. Lett.* **45**: 7075–7083. doi:10.1029/2018GL077618
- Ohde, T., and I. Dadou. 2018. Seasonal and annual variability of coastal sulphur plumes in the northern Benguela upwelling system. *PLoS One* **13**: e0192140.
- Ohde, T., H. Siegel, J. Reißmann, and M. Gerth. 2007. Identification and investigation of sulphur plumes along the Namibian coast using the MERIS sensor. *Cont. Shelf Res.* **27**: 744–756. doi:10.1016/j.csr.2006.11.016
- Roslev, P., M. B. Larsen, D. Jørgensen, and M. Hesselsoe. 2004. Use of heterotrophic CO₂ assimilation as a measure of metabolic activity in planktonic and sessile bacteria. *J. Microbiol. Methods* **59**: 381–393. doi:10.1016/j.mimet.2004.08.002
- Schouten, S., W. I. C. Rijpstra, E. Durisch-Kaiser, C. J. Schubert, and J. S. Sinninghe Damsté. 2012. Distribution of glycerol dialkyl glycerol tetraether lipids in the water column of Lake Tanganyika. *Org. Geochem.* **53**: 34–37. doi:https://doi.org/10.1016/j.orggeochem.2012.01.009
- Schubert, C. J., E. Durisch-Kaiser, B. Wehrli, B. Thamdrup, P. Lam, and M. M. M. Kuypers. 2006. Anaerobic ammonium oxidation in a tropical freshwater system (Lake

- Tanganyika). *Environ. Microbiol.* **8**: 1857–1863. doi:10.1111/j.1462-2920.2006.01074.x
- Schunck, H., G. Lavik, D. K. Desai, and others. 2013. Giant Hydrogen Sulfide Plume in the Oxygen Minimum Zone off Peru Supports Chemolithoautotrophy L.J. Stal [ed.]. *PLoS One* **8**: e68661. doi:10.1371/journal.pone.0068661
- Shenoy, D. M., K. B. Sujith, M. U. Gauns, S. Patil, A. Sarkar, H. Naik, P. V Narvekar, and S. W. A. Naqvi. 2012. Production of dimethylsulphide during the seasonal anoxia off Goa. *Biogeochemistry* **110**: 47–55. doi:10.1007/s10533-012-9720-5
- Shirodkar, G., S. W. A. Naqvi, H. Naik, A. K. Pratihary, S. Kurian, and D. M. Shenoy. 2018. Methane dynamics in the shelf waters of the West coast of India during seasonal anoxia. *Mar. Chem.* **203**: 55–63. doi:10.1016/J.MARCHEM.2018.05.001
- Sommer, S., J. Gier, T. Treude, U. Lomnitz, M. Dengler, J. Cardich, and A. W. Dale. 2016. Depletion of oxygen, nitrate and nitrite in the Peruvian oxygen minimum zone cause an imbalance of benthic nitrogen fluxes. *Deep Sea Res. Part I Oceanogr. Res. Pap.* **112**: 113–122. doi:10.1016/J.DSR.2016.03.001
- Stief, P., A. S. B. Lundgaard, Á. Morales-Ramírez, B. Thamdrup, and R. N. Glud. 2017. Fixed-Nitrogen Loss Associated with Sinking Zooplankton Carcasses in a Coastal Oxygen Minimum Zone (Golfo Dulce, Costa Rica) . *Front. Mar. Sci.* **4**: 152.
- Ulloa, O., D. E. Canfield, E. F. DeLong, R. M. Letelier, and F. J. Stewart. 2012. Microbial oceanography of anoxic oxygen minimum zones. *Proc Natl Acad Sci U S A* **109**: 15996–16003. doi:10.1073/pnas.1205009109
- Wawrik, B., A. V Callaghan, and D. A. Bronk. 2009. Use of inorganic and organic nitrogen by *Synechococcus* spp. and diatoms on the west Florida shelf as measured using stable isotope probing. *Appl. Environ. Microbiol.* **75**: 6662–70. doi:10.1128/AEM.01002-09
- Weeks, S. J., B. Currie, A. Bakun, and K. R. Peard. 2004. Hydrogen sulphide eruptions in the Atlantic Ocean off southern Africa: implications of a new view based on SeaWiFS satellite imagery. *Deep Sea Res. Part I Oceanogr. Res. Pap.* **51**: 153–172. doi:10.1016/j.dsr.2003.10.004
- De Wever, A., K. Muylaert, K. Van der Gucht, S. Pirlot, C. Cocquyt, J.-P. Descy, P.-D. Plisnier, and W. Vyverman. 2005. Bacterial community composition in Lake Tanganyika: vertical and horizontal heterogeneity. *Appl. Environ. Microbiol.* **71**: 5029–37. doi:10.1128/AEM.71.9.5029-5037.2005
- Zakem, E. J., and N. M. Levine. 2019. Systematic Variation in Marine Dissolved Organic Matter Stoichiometry and Remineralization Ratios as a Function of Lability. *Global Biogeochem. Cycles* **33**: 1389–1407. doi:10.1029/2019GB006375

REVIEWERS' COMMENTS

Reviewer #1 (Remarks to the Author):

The authors have revised the manuscript one more according to several comments and suggestions. I believe the authors have replied correctly to my comments and modified the text accordingly. Although there are still some minor discrepancies, I believe the manuscript is ready for publication. Thanks for the willingness to improve the manuscript.

Below is a point-by-point response to the reviewers' comments, indicated in the red-type face.

REVIEWER COMMENTS

Reviewer #1 (Remarks to the Author):

The authors have revised the manuscript one more according to several comments and suggestions. I believe the authors have replied correctly to my comments and modified the text accordingly. Although there are still some minor discrepancies, I believe the manuscript is ready for publication. Thanks for the willingness to improve the manuscript.

We would again like to thank the reviewer for their support of this work, their attention to detail and their constructive criticism. Their due diligence has greatly contributed to improving the manuscript.